# Decentralized Noncooperative Games with Coupled Decision-Dependent Distributions

**Wenjing Yan**    **Xuanyu Cao** *
Department of Electronic and Computer Engineering
The Hong Kong University of Science and Technology
`wj.yan@connect.ust.hk, eexcao@ust.hk`

## Abstract

Distribution variations in machine learning, driven by the dynamic nature of deployment environments, significantly impact the performance of learning models. This paper explores endogenous distribution shifts in learning systems, where deployed models influence environments, which in turn alters the data distributions that the learning models rely on. This phenomenon is formulated by a decision-dependent distribution mapping within the recently introduced framework of performative prediction (PP) (Perdomo et al., 2020). Our study investigates the performative effect in a decentralized noncooperative game, where players aim to minimize private cost functions while simultaneously managing coupled inequality constraints. In this context, we examine two equilibrium concepts for the studied game: performative stable equilibrium (PSE) and Nash equilibrium (NE), and establish sufficient conditions for their existence and uniqueness. Notably, we provide the first upper bound on the distance between the PSE and NE in the literature, which is challenging to evaluate due to the absence of strong convexity on the joint cost function. Furthermore, we develop a decentralized stochastic primal-dual algorithm for efficiently computing the PSE point. By rigorously bounding the performative effect, we prove that the proposed algorithm achieves sublinear convergence rates for both performative regret and constraint violations and maintains the same order of convergence rate as the case without performativity. Numerical experiments further confirm the effectiveness of our algorithm and theoretical results.

## 1 Introduction

Machine learning aims to generalize models trained on given datasets to make accurate predictions or decisions on new, unseen data (El Naqa and Murphy, 2015). The effectiveness of those models depends on the alignment between the training datasets and deployment environments (Quinonero-Candela et al., 2008). However, real-world environments are seldom static and often exhibit fluctuations that can severely degrade model performance (Zhou, 2022). In particular, shifts in data-generating distributions, driven by the dynamic nature of real-world conditions, present significant challenges for model deployment.

Distribution shifts in machine learning can occur exogenously or endogenously. Exogenous distribution shifts are driven by external factors beyond the control of the learning platforms, such as environmental changes (Chan et al., 2020) or policy amendments (Wu et al., 2021). In contrast, endogenous shifts arise from the system's inherent dynamics and interactions, where the deployed models affect environments, which in turn alters the data distributions that the learning models rely on (Dong et al., 2018). For instance, an increase in commodity prices may decrease user interest, thereby impacting sales. The key distinction lies in the controllability of endogenous shifts, providing

---

*Corresponding Author.

38th Conference on Neural Information Processing Systems (NeurIPS 2024).

an opportunity for designers to either exploit these shifts for improved performance or mitigate unintended consequences (Dean et al., 2023).

While substantial efforts have been made to address exogenous distribution changes, such as covariate shift (Chan et al., 2020), label shift (Wu et al., 2021), and concept drift (Lu et al., 2018), relatively little attention has been paid to the challenges posed by endogenous distribution shifts. Tackling these endogenous shifts is particularly challenging as data distributions are intrinsically linked to the decisions made by the learning model itself (Perdomo et al., 2020). As a result, addressing endogenous shifts may require the explicit modeling of feedback loops, consideration of causal relationships, and the adaptation of models to dynamic environments.

A notable advancement in this area is the recently proposed framework of "performative prediction (PP)" (Perdomo et al., 2020), also referred to as "decision-dependent learning" (Drusvyatskiy and Xiao, 2023). This framework elegantly captures the dynamic interplay between decisions and data distributions through a decision-dependent mapping, denoted by $\mathcal{D}(\boldsymbol{\theta})$ where $\boldsymbol{\theta}$ represents the decision variable. By linking $\boldsymbol{\theta}$ to the data distribution, this formulation bridges the gap between model deployment and parameter optimization. Following the seminal work of (Perdomo et al., 2020), a growing body of research has emerged, focusing on stability and optimality analysis (Piliouras and Yu, 2023; Miller et al., 2021), as well as algorithmic design for various settings, including reinforcement learning (Mandal et al., 2023), online learning (Wood et al., 2021), bandit problems (Jagadeesan et al., 2022), and bilevel optimization (Lu, 2023).

This paper investigates endogenous distribution shifts in a decentralized noncooperative game, where players aim to minimize private cost functions while simultaneously managing coupled inequality constraints. To contextualize this setting, consider scenarios where strategic responses exhibit in learning environments and competitive interactions occur among players. For example, in autonomous vehicular networks, multiple vehicles compete to select their routes under constraints such as road capacities, traffic congestion, and travel costs. The route choices of each vehicle influence traffic patterns and consequently affect the travel times experienced by other vehicles (Mori et al., 2015). Similarly, in finance, traders compete to maximize profits under constraints like market capacities and inventory levels. The trading strategies of these participants impact market volatility and the distribution of asset prices, creating a dynamic pricing landscape (Fattouh and Mahadeva, 2014). These dynamics extend to other domains, such as electricity market competition (Moshari et al., 2010), ride-sharing platforms (Narang et al., 2023), natural resource extraction (Cust and Poelhekke, 2015), and online advertising auctions (Varian, 2009).

Despite its pervasiveness, this performative phenomenon has largely been overlooked in the studies of decentralized noncooperative games. This paper addresses the problem by formulating performativity using coupled decision-dependent distributions, following the PP framework of (Perdomo et al., 2020). However, the intricate interplay between decentralized players and endogenous distribution shifts presents challenging theoretical and algorithmic questions: *How do strategic responses in learning environments influence the game's equilibrium? How can players adapt their strategies effectively when confronted with coupled decision-dependent distributions? How can we design algorithms to exploit these dynamics for optimal decision-making?* These questions form the core of our investigation, guiding us toward more resilient, adaptive, and efficient learning outcomes in decentralized games, especially in environments characterized by continuously evolving data and decision-making processes. Our main contributions are summarized below:

- We initially formulate the problem of decentralized noncooperative games with data performativity, where selfish players seek to minimize individual costs while managing coupled inequality constraints. Under this setting, we examine two equilibrium concepts: performative stable equilibrium (PSE) and Nash equilibrium (NE), and establish sufficient conditions for their existence and uniqueness. Compared to conventional games, this examination is more complicated due to the interplay between decision-making and distribution changes. Notably, we make a significant contribution by providing the first upper bound on the distance between the PSE and NE in the literature. Computing this distance in PP games is challenging due to the absence of strong convexity on the joint cost function, an essential property for determining the optimality gap of performative stable points in previous work. Instead, we characterize the distance by leveraging relations from strong duality and derive a result comparable to the findings of the prior work (Perdomo et al., 2020; Lu, 2023).

- To compute the PSE point of the PP-game, we propose a decentralized stochastic primal-dual algorithm based on repeated risk minimization (RRM). The development and convergence analysis of this algorithm face two primary challenges. First, there is a complex interaction between decentralized competition and endogenous distribution shifts. Second, players only have partial observation, as they communicate solely with neighbors, despite their private cost functions being influenced by the strategies of all players. We evaluate the performance of our algorithm by two commonly used metrics: performative regret, which measures the suboptimality of the strategy sequence generated by RRM relative to the PSE point, and constraint violation. By rigorously bounding the performative effect, we prove that the proposed algorithm achieves sublinear convergence rates for both metrics. Furthermore, our results show that while the performative effect slows down convergence, it does not degrade the order of performative regret compared to the case without performativity (Lu et al., 2020).

Finally, we conduct numerical experiments on a networked Cournot game and a ride-share market. The simulation results confirm the sublinear convergence of our algorithm. Furthermore, the results demonstrate that while greater performative strength leads to a wider gap between the PSE and NE, the discrepancy between these two equilibria remains marginal. This verifies both the effectiveness of the PSE solutions and the accuracy of our distance analysis between the PSE and NE.

**Related Work:** Among the numerous existing studies, two closely related works (Narang et al., 2023) and (Wang et al., 2023) have considered performative behaviors in games. A key distinction in our work is that our model requires all players' collective strategies to adhere to the constraints of the learning system, whereas both (Narang et al., 2023) and (Wang et al., 2023) address unconstrained settings. This difference results in fundamentally distinct algorithmic designs and convergence analyses. Our approach employs a primal-dual technique and requires consensus, whereas their methods only rely on local stochastic gradient descent. Additionally, we consider a mathematically richer model compared to (Wang et al., 2023), whose framework is structured in a specific form involving local costs dependent solely on individual strategies and a regularizer quantifying similarity among neighboring strategies. Furthermore, our algorithm design accounts for practical constraints where players can only communicate with their immediate neighbors, while (Narang et al., 2023) assumes full accessibility to all players' strategies across the entire network. Importantly, our work makes a significant contribution by providing the first upper bound on the distance between the performative stable equilibrium (PSE) and Nash equilibrium (NE)—a gap not previously addressed. Other related works such as (Li et al., 2022) and (Piliouras and Yu, 2023), have studied performative prediction in decentralized multi-agent optimization. The former focuses on consensus-seeking agents, while the latter is restricted to location-scale families. Finally, (Yan and Cao, 2024b) considers the constrained performative prediction problem in a single-agent setting, whereas our paper addresses decentralized noncooperative games. A more comprehensive literature review is provided in Appendix A.

## 2 Problem Formulation

Consider a decentralized noncooperative game with $n$ players. Each player $i$ selects a strategy (or, interchangeably, decision, action), denoted as $\boldsymbol{\theta}_i$, from its feasible set $\boldsymbol{\Omega}_i \subseteq \mathbb{R}^d$. Let the collective decisions of all players be denoted as $\boldsymbol{\theta} := \mathrm{col}\,(\boldsymbol{\theta}_1, \cdots, \boldsymbol{\theta}_n)$, and the collective decisions of all players except player $i$ be represented as $\boldsymbol{\theta}_{-i} := \mathrm{col}\,(\boldsymbol{\theta}_1, \cdots, \boldsymbol{\theta}_{i-1}, \boldsymbol{\theta}_{i+1}, \cdots, \boldsymbol{\theta}_n)$, for any $i \in [n]$, where $[n]$ denotes the set of integers $\{1, 2, \ldots, n\}$. Each player $i$ has a private cost function $J_i(\boldsymbol{\xi}_i; \boldsymbol{\theta}_i, \boldsymbol{\theta}_{-i})$, which depends on the random variable $\boldsymbol{\xi}_i \in \boldsymbol{\Xi}_i$, the player's private decision $\boldsymbol{\theta}_i$, and the decisions of all other players $\boldsymbol{\theta}_{-i}$. This paper considers a scenario where the underlying populations strategically respond to the players' decisions, causing shifts in data distributions. This interplay is modeled by a decision-dependent distribution mapping $\boldsymbol{\xi}_i \sim \mathcal{D}_i\,(\boldsymbol{\theta}_i, \boldsymbol{\theta}_{-i})$ for all $i \in [n]$. The objective of each player $i$ is to selfishly minimize its *performative risk* $\mathbb{E}_{\boldsymbol{\xi}_i \sim \mathcal{D}_i(\boldsymbol{\theta}_i, \boldsymbol{\theta}_{-i})} J_i(\boldsymbol{\xi}_i; \boldsymbol{\theta}_i, \boldsymbol{\theta}_{-i})$ (abbreviated as $\mathrm{PR}_i(\boldsymbol{\theta}_i, \boldsymbol{\theta}_{-i})$), subject to a coupled constraint $\sum_{i=1}^n \boldsymbol{g}_i(\boldsymbol{\theta}_i) \preceq \boldsymbol{0}$, i.e.,

$$\min_{\boldsymbol{\theta}_i \in \boldsymbol{\Omega}_i} \quad \mathbb{E}_{\boldsymbol{\xi}_i \sim \mathcal{D}_i(\boldsymbol{\theta}_i, \boldsymbol{\theta}_{-i})} J_i\,(\boldsymbol{\xi}_i; \boldsymbol{\theta}_i, \boldsymbol{\theta}_{-i})$$
$$\text{subject to} \quad \boldsymbol{g}_i(\boldsymbol{\theta}_i) + \textstyle\sum_{j \neq i} \boldsymbol{g}_j(\boldsymbol{\theta}_j) \preceq \boldsymbol{0}. \tag{1}$$

Both $J_i(\cdot)$ and $\boldsymbol{g}_i(\cdot)$ are only locally accessible to player $i$ for all $i \in [n]$. In the game (1), each player solves its private optimization problem to determine the best strategy, given the current strategies

of all the other players. An equilibrium of the game (1) corresponds to a set of strategies where no player can improve its performance by deviating unilaterally from its strategy.

Denote by $\boldsymbol{\xi} := \mathrm{col}\left(\boldsymbol{\xi}_1, \cdots, \boldsymbol{\xi}_n\right)$ the concatenation of the variables $\boldsymbol{\xi}_i$ and by $J\left(\boldsymbol{\xi}; \boldsymbol{\theta}\right) := \mathrm{col}\left(J_1\left(\boldsymbol{\xi}_1; \boldsymbol{\theta}\right), \cdots, J_n\left(\boldsymbol{\xi}_n; \boldsymbol{\theta}\right)\right)$ the concatenation of the cost functions $J_i(\cdot)$ for all $i \in [n]$. A stochastic pseudogradient mapping of $J\left(\boldsymbol{\xi}; \boldsymbol{\theta}\right)$ is defined as $\nabla J\left(\boldsymbol{\xi}; \boldsymbol{\theta}\right) := \mathrm{col}\left(\nabla_{\boldsymbol{\theta}_1} J_1\left(\boldsymbol{\xi}_1; \boldsymbol{\theta}\right), \cdots, \nabla_{\boldsymbol{\theta}_n} J_n\left(\boldsymbol{\xi}_n; \boldsymbol{\theta}\right)\right)$. We have the following assumption on $\nabla J\left(\boldsymbol{\xi}; \boldsymbol{\theta}\right)$.

**Assumption 2.1.** There exists a constant $\mu > 0$ such that the stochastic gradient mapping $\nabla J\left(\boldsymbol{\xi}; \boldsymbol{\theta}\right)$ is $\mu$-strongly monotone, i.e., $\left\langle \nabla J\left(\boldsymbol{\xi}; \boldsymbol{\theta}\right) - \nabla J\left(\boldsymbol{\xi}; \boldsymbol{\theta}'\right), \boldsymbol{\theta} - \boldsymbol{\theta}' \right\rangle \geq \mu \|\boldsymbol{\theta} - \boldsymbol{\theta}'\|_2^2, \forall \boldsymbol{\xi} \in \boldsymbol{\Xi}, \boldsymbol{\theta}, \boldsymbol{\theta}' \in \boldsymbol{\Omega}$, where $\boldsymbol{\Xi} := \boldsymbol{\Xi}_1 \times \cdots \times \boldsymbol{\Xi}_n$ and $\boldsymbol{\Omega} := \boldsymbol{\Omega}_1 \times \cdots \times \boldsymbol{\Omega}_n$.

Assumption 2.1 is commonly made in the literature of game theory. It suffices to guarantee the existence of Nash equilibrium for a stochastic game with fixed data distributions (Facchinei and Pang, 2003, Theorem 2.3.3(b)). However, in our paper, since the data distributions are decision-dependent, Assumption 2.1 does not imply the monotonicity of the gradient mapping of the joint performative risk, denoted by $\mathrm{PR}(\cdot) := \mathrm{col}\left(\mathrm{PR}_1(\cdot), \cdots, \mathrm{PR}_n(\cdot)\right)$. Therefore, the existence and uniqueness (E&U) conditions for the Nash equilibrium of the game (1) need further investigation.

We define a graph $\mathcal{G}(\mathbf{P})$ to represent the impact of players' decisions on the data distributions of different players. In $\mathcal{G}(\mathbf{P})$, the weight $p_{ij} > 0$ if player $j$'s decision affects player $i$'s data distribution, and $p_{ij} = 0$ otherwise. Particularly, $p_{ii}$ represents the weight of self-influence. These weights are normalized as $\sum_{j=1}^{n} p_{ij} = 1$, for all $i \in [n]$. Clearly, the larger the weight $p_{ij}$, the stronger the effect of player $j$'s decision on the data distribution of player $i$.

Let $\mathcal{W}_1\left(\mathcal{D}, \mathcal{D}'\right)$ represent the *Wasserstein*-1 distance between two probability measures $\mathcal{D}$ and $\mathcal{D}'$. Following (Wang et al., 2023), we impose the following assumption on the distributions $\{\mathcal{D}_i\}_{i \in [n]}$.

**Assumption 2.2.** For any $i \in [n]$, there exists a constant $\varepsilon_i \geq 0$ such that, $\forall \boldsymbol{\theta}, \boldsymbol{\theta}' \in \boldsymbol{\Omega}$, the distribution mapping $\mathcal{D}_i$ is constrained by $\mathcal{W}_1\left(\mathcal{D}_i\left(\boldsymbol{\theta}\right), \mathcal{D}_i\left(\boldsymbol{\theta}'\right)\right) \leq \varepsilon_i \sqrt{\sum_{j=1}^{n} p_{ij} \left\|\boldsymbol{\theta}_j - \boldsymbol{\theta}'_j\right\|_2^2}$.

For any $i \in [n]$, the parameter $\varepsilon_i$ bounds the sensitivity of player $i$'s distribution with respect to (w.r.t.) the decision variations of all players. This $\varepsilon$-sensitivity property of distributions is conceptually akin to the Lipschitz continuity of functions that quantifies the variation of function values w.r.t argument changes. We also require the following assumptions.

**Assumption 2.3.** For any $i \in [n]$, the non-empty feasible set $\boldsymbol{\Omega}_i$ is closed, convex, and bounded, i.e., there exists a constant $C \geq 0$ such that, $\forall \boldsymbol{\theta}_i \in \boldsymbol{\Omega}_i, \|\boldsymbol{\theta}_i\|_2 \leq C$.

**Assumption 2.4.** For any $i \in [n]$ and $\boldsymbol{\theta}_i \in \boldsymbol{\Omega}_i$, the cost function $J_i(\boldsymbol{\xi}_i; \boldsymbol{\theta}_i, \boldsymbol{\theta}_{-i})$ is convex w.r.t. $\boldsymbol{\theta}_i$. Moreover, there exists a constant $L_i \geq 0$ such that $J_i\left(\boldsymbol{\xi}_i; \boldsymbol{\theta}\right)$ is $L_i$-smooth, i.e, $\left\|\nabla J_i\left(\boldsymbol{\xi}_i; \boldsymbol{\theta}\right) - \nabla J_i\left(\boldsymbol{\xi}'_i; \boldsymbol{\theta}'\right)\right\|_2 \leq L_i \left(\left\|\boldsymbol{\xi}_i - \boldsymbol{\xi}'_i\right\|_2 + \left\|\boldsymbol{\theta} - \boldsymbol{\theta}'\right\|_2\right), \forall \boldsymbol{\xi}_i, \boldsymbol{\xi}'_i \in \boldsymbol{\Xi}_i, \boldsymbol{\theta}, \boldsymbol{\theta}' \in \boldsymbol{\Omega}$.

**Assumption 2.5.** For any $i \in [n]$ and $\boldsymbol{\theta}_i \in \boldsymbol{\Omega}_i$, the constraint function $\boldsymbol{g}_i(\boldsymbol{\theta}_i)$ is convex w.r.t. $\boldsymbol{\theta}_i$. Moreover, there exist a constant $G_g \geq 0$ such that $\boldsymbol{g}_i(\cdot)$ is $G_g$-Lipschitz, i.e., $\left\|\boldsymbol{g}_i(\boldsymbol{\theta}_i) - \boldsymbol{g}_i(\boldsymbol{\theta}'_i)\right\|_2 \leq G_g \|\boldsymbol{\theta}_i - \boldsymbol{\theta}'_i\|_2, \forall \boldsymbol{\theta}_i, \boldsymbol{\theta}'_i \in \boldsymbol{\Omega}_i$.

Assumptions 2.3 and 2.5 are widely used in constrained optimization (Bertsekas, 2014; Yan and Cao, 2024a), and Assumption 2.4 is standard in the PP literature. From Yan and Cao (2024a, Proposition 1), under Assumptions 2.3 and 2.4, the cost function $J_i(\boldsymbol{\xi}_i; \boldsymbol{\theta}), \forall i \in [n]$ is Lipschitz continuous, i.e., there exist a constant $G_i \geq 0$ such that $|J_i(\boldsymbol{\xi}_i; \boldsymbol{\theta}) - J_i(\boldsymbol{\xi}'_i; \boldsymbol{\theta}')| \leq G_i \left(\left\|\boldsymbol{\xi}_i - \boldsymbol{\xi}'_i\right\|_2 + \left\|\boldsymbol{\theta} - \boldsymbol{\theta}'\right\|_2\right), \forall \boldsymbol{\xi}_i, \boldsymbol{\xi}'_i \in \boldsymbol{\Xi}_i, \boldsymbol{\theta}, \boldsymbol{\theta}' \in \boldsymbol{\Omega}$. Moreover, Assumptions 2.3 and 2.5 imply the boundedness of $\|\boldsymbol{g}_i(\boldsymbol{\theta}_i)\|_2$, i.e., there exists a constant $B \geq 0$ such that $\|\boldsymbol{g}_i(\boldsymbol{\theta}_i)\|_2 \leq B, \forall \boldsymbol{\theta}_i \in \boldsymbol{\Omega}_i, i \in [n]$.

## 3   Equilibrium of the PP-Game

This section examines two fundamental equilibrium concepts of the performative game (1): Nash equilibrium (NE) and performative stable equilibrium (PSE), as defined below.

**Definition 3.1** (Nash Equilibrium). A vector $\boldsymbol{\theta}^{\mathrm{ne}} := \mathrm{col}\left(\boldsymbol{\theta}_1^{\mathrm{ne}}, \ldots, \boldsymbol{\theta}_n^{\mathrm{ne}}\right)$ achieves an NE of the game (1) if it holds for any $i \in [n]$ that

$$\boldsymbol{\theta}_i^{\mathrm{ne}} \in \arg\min_{\boldsymbol{\theta}_i \in \boldsymbol{\Omega}_i} \quad \mathbb{E}_{\boldsymbol{\xi}_i \sim \mathcal{D}_i\left(\boldsymbol{\theta}_i, \boldsymbol{\theta}_{-i}^{\mathrm{ne}}\right)} J_i\left(\boldsymbol{\xi}_i; \boldsymbol{\theta}_i, \boldsymbol{\theta}_{-i}^{\mathrm{ne}}\right)$$

$$\text{subject to} \quad \boldsymbol{g}_i(\boldsymbol{\theta}_i) + \sum_{j \neq i} \boldsymbol{g}_j(\boldsymbol{\theta}_j^{\mathrm{ne}}) \preceq \mathbf{0}.$$

**Definition 3.2** (Performative Stable Equilibrium). A vector $\boldsymbol{\theta}^{\mathrm{pse}} := \mathrm{col}\,(\boldsymbol{\theta}_1^{\mathrm{pse}}, \ldots, \boldsymbol{\theta}_n^{\mathrm{pse}})$ achieves a PSE of the game (1) if it holds for any $i \in [n]$ that

$$\boldsymbol{\theta}_i^{\mathrm{pse}} \in \underset{\boldsymbol{\theta}_i \in \boldsymbol{\Omega}_i}{\arg\min} \quad \mathbb{E}_{\boldsymbol{\xi}_i \sim \mathcal{D}_i(\boldsymbol{\theta}^{\mathrm{pse}})} J_i\left(\boldsymbol{\xi}_i; \boldsymbol{\theta}_i, \boldsymbol{\theta}_{-i}^{\mathrm{pse}}\right)$$

$$\text{subject to} \quad \boldsymbol{g}_i(\boldsymbol{\theta}_i) + \sum_{j \neq i} \boldsymbol{g}_j(\boldsymbol{\theta}_j^{\mathrm{pse}}) \preceq \mathbf{0}.$$

NE is a fundamental concept in game theory. At NE, each player's strategy optimally aligns with its own interest, given the strategies of other players. Hence, no player has an incentive to deviate from its strategy unilaterally. In the case of performative games, the computation of NE needs to take into account the data distributions $\mathcal{D}_i(\cdot)$ for all $i \in [n]$, as they are parameterized by the optimization variable $\boldsymbol{\theta}$. However, this information is often unavailable in practice. Instead, at PSE, the data distribution of each player $i \in [n]$ is fixed at $\mathcal{D}_i\left(\boldsymbol{\theta}^{\mathrm{pse}}\right)$ and the PSE point achieves an NE of the game (1) under the fixed data distribution of its own deployment. This formulation draws benign properties akin to problems with fixed data distributions, facilitating the adaptation of existing algorithms. Therefore, PSE is more frequently chosen as a performance metric in the literature of PP.

### 3.1 Existence and Uniqueness of PSE

We first establish the condition for the E&U of the PSE of the game (1). Our approach relies on repeated risk minimization (RRM) for closed-loop retraining. First, we define a mapping $\mathcal{T}(\boldsymbol{\theta}) := \{\mathcal{T}_i(\boldsymbol{\theta})\}_{i \in [n]}$ that, for any $i \in [n]$,

$$\boldsymbol{\theta}_i' = \mathcal{T}_i(\boldsymbol{\theta}) := \underset{\boldsymbol{u}_i \in \boldsymbol{\Omega}_i}{\arg\min} \quad \mathbb{E}_{\boldsymbol{\xi}_i \sim \mathcal{D}_i(\boldsymbol{\theta}_i, \boldsymbol{\theta}_{-i})} J_i\left(\boldsymbol{\xi}_i; \boldsymbol{u}_i, \boldsymbol{\theta}_{-i}'\right)$$

$$\text{subject to} \quad \boldsymbol{g}_i(\boldsymbol{u}_i) + \sum_{j \neq i} \boldsymbol{g}_j\left(\boldsymbol{\theta}_j'\right) \preceq \mathbf{0}.$$

The mapping $\mathcal{T}(\boldsymbol{\theta})$ outputs the NE of the game (1) under the fixed data distributions $\mathcal{D}_i(\boldsymbol{\theta}_i, \boldsymbol{\theta}_{-i})$ for all $i \in [n]$. With Assumption 2.1, the E&U of this NE is guaranteed, thereby ensuring the validity of the mapping $\mathcal{T}(\boldsymbol{\theta})$. Based on $\mathcal{T}(\boldsymbol{\theta})$, the RRM updates $\boldsymbol{\theta}_i^t$ at each iteration $t$ by

$$\boldsymbol{\theta}_i^{t+1} = \mathcal{T}_i(\boldsymbol{\theta}^t), \forall i \in [n]. \tag{2}$$

Clearly, $\boldsymbol{\theta}^{t+1}$ is an NE of the game (1) under the deployment of $\boldsymbol{\theta}^t$. Additionally, we have that any fixed point of (2) achieves an PSE for the game (1), i.e., $\boldsymbol{\theta}^{\mathrm{pse}} = \mathcal{T}(\boldsymbol{\theta}^{\mathrm{pse}})$. By investigating the convergence the iterative equation (2), we have the following sufficient condition for the E&U of the PSE of the game (1).

**Theorem 3.3.** *Suppose that Assumptions 2.1-2.5 hold. Then, for any $\boldsymbol{\theta}, \boldsymbol{\delta} \in \boldsymbol{\Omega}$, the mapping $\mathcal{T}(\boldsymbol{\theta})$ satisfies*

$$\|\mathcal{T}(\boldsymbol{\theta}) - \mathcal{T}(\boldsymbol{\delta})\|_2 \leq \tfrac{1}{\mu} \sqrt{\sum_{i=1}^n L_i^2 \varepsilon_i^2 \max_{j \in [n]} p_{ij}} \, \|\boldsymbol{\theta} - \boldsymbol{\delta}\|_2 \,.$$

*Thus, if it is satisfied that*

$$\tfrac{1}{\mu} \sqrt{\sum_{i=1}^n L_i^2 \varepsilon_i^2 \max_{j \in [n]} p_{ij}} < 1, \tag{3}$$

*the sequence generated by the RRM* (2) *converges to a unique PSE point $\boldsymbol{\theta}^{\mathrm{pse}}$ at a linear rate that*

$$\|\boldsymbol{\theta}^{t+1} - \boldsymbol{\theta}^{\mathrm{pse}}\|_2 \leq \left(\tfrac{1}{\mu} \sqrt{\sum_{i=1}^n L_i^2 \varepsilon_i^2 \max_{j \in [n]} p_{ij}}\right)^t \|\boldsymbol{\theta}^1 - \boldsymbol{\theta}^{\mathrm{pse}}\|_2 \,.$$

The proof of Theorem 3.3 is provided in Appendix B. According to Theorem 3.3, under Assumptions 2.1-2.5, when condition (3) holds, we have that: (i) the game (1) admits a unique PSE, and (ii) the RRM method (2) converges linearly to the PSE.

Since the influence weights $\{p_{ij}\}_{j \in [n]}$ are normalized, with $\sum_{j=1}^n p_{ij} = 1$ for all $i \in [n]$, we generally have that $p_{ij} = \mathcal{O}(\frac{1}{n})$. Therefore, the contraction condition (3) exhibits good scalability w.r.t. the number of players. Moreover, according to the proof in Appendix B, if for any player $i \in [n]$, its distribution $\mathcal{D}_i(\cdot)$ depends only on its own decision $\boldsymbol{\theta}_i$, i.e., $p_{ij} = 0$ for all $j \neq i$, then we have

$$\|\mathcal{T}(\boldsymbol{\theta}) - \mathcal{T}(\boldsymbol{\delta})\|_2 \leq \tfrac{1}{\mu} \max_{i \in [n]} L_i \varepsilon_i \, \|\boldsymbol{\delta} - \boldsymbol{\theta}\|_2 \,.$$

The contraction of the above iterative equation only requires that $\frac{1}{\mu} \max_{i\in[n]} L_i\varepsilon_i < 1$. Furthermore, if all players exhibit equivalent model parameters that $L_1 = \cdots = L_n = L$ and $\varepsilon_1 = \cdots = \varepsilon_n = \varepsilon$ and $p_{ij} = \frac{1}{n}$ for all $i, j \in [n]$, condition (3) reduces to $\frac{L\varepsilon}{\mu} < 1$, recovering the contraction requirement of (Perdomo et al., 2020) for a single-agent PP case.

## 3.2 Existence and Uniqueness of NE

First, we define a gradient mapping $G_{\boldsymbol{\theta}}^{(i)}(\boldsymbol{\delta}_i, \boldsymbol{\delta}_{-i}) := \mathbb{E}_{\boldsymbol{\xi}_i \sim \mathcal{D}_i(\boldsymbol{\theta})} \nabla_{\boldsymbol{\delta}_i} J_i(\boldsymbol{\xi}_i; \boldsymbol{\delta}_i, \boldsymbol{\delta}_{-i})$ for any $i \in [n]$, and $G_{\boldsymbol{\theta}}(\boldsymbol{\delta}) := \mathrm{col}\left(G_{\boldsymbol{\theta}}^{(1)}(\boldsymbol{\delta}), \cdots, G_{\boldsymbol{\theta}}^{(n)}(\boldsymbol{\delta})\right)$. Moreover, for any $i \in [n]$, define

$$H_{\boldsymbol{\theta}_i, \boldsymbol{\theta}_{-i}}^{(i)}(\boldsymbol{\delta}) := \nabla_{\boldsymbol{u}_i} \mathbb{E}_{\boldsymbol{\xi}_i \sim \mathcal{D}_i(\boldsymbol{u}_i, \boldsymbol{\theta}_{-i})} \left[J_i\left(\boldsymbol{\xi}_i; \boldsymbol{\delta}\right)\right]\big|_{\boldsymbol{u}_i = \boldsymbol{\theta}_i}$$

and $H_{\boldsymbol{\theta}}(\boldsymbol{\delta}) := \mathrm{col}\left(H_{\boldsymbol{\theta}_1, \boldsymbol{\theta}_{-1}}^{(1)}(\boldsymbol{\delta}), \cdots, H_{\boldsymbol{\theta}_n, \boldsymbol{\theta}_{-n}}^{(n)}(\boldsymbol{\delta})\right)$. Then, for any $i \in [n]$, the gradient of the performative risk $\mathrm{PR}_i(\boldsymbol{\theta}_i, \boldsymbol{\theta}_{-i})$ w.r.t. $\boldsymbol{\theta}_i$ is given by

$$\nabla_{\boldsymbol{\theta}_i} \mathrm{PR}_i(\boldsymbol{\theta}_i, \boldsymbol{\theta}_{-i}) = G_{\boldsymbol{\theta}_i, \boldsymbol{\theta}_{-i}}^{(i)}(\boldsymbol{\theta}_i, \boldsymbol{\theta}_{-i}) + H_{\boldsymbol{\theta}_i, \boldsymbol{\theta}_{-i}}^{(i)}(\boldsymbol{\theta}_i, \boldsymbol{\theta}_{-i}).$$

Define $\nabla\mathrm{PR}(\boldsymbol{\theta}) := \mathrm{col}\left(\nabla_{\boldsymbol{\theta}_1} \mathrm{PR}_i(\boldsymbol{\theta}), \cdots, \nabla_{\boldsymbol{\theta}_n} \mathrm{PR}_n(\boldsymbol{\theta})\right)$, we further have

$$\nabla\mathrm{PR}(\boldsymbol{\theta}) = G_{\boldsymbol{\theta}}(\boldsymbol{\theta}) + H_{\boldsymbol{\theta}}(\boldsymbol{\theta}).$$

From Facchinei and Pang (2003, Theorem 2.3.3(b)), to prove the E&U of the NE of the (1), we require the strongly monotonivity of the gradient mapping $\nabla\mathrm{PR}(\boldsymbol{\theta})$. Therefore, we have the following sufficient condition for the E&U of the NE of the game (1).

**Theorem 3.4.** *Suppose that Assumptions 2.1-2.5 hold. If it is satisfied that*

$$\mu - \sum_{i=1}^{n} L_i\varepsilon_i \max_{j\in[n]} \sqrt{p_{ij}} - \sqrt{\sum_{i=1}^{n} L_i^2\varepsilon_i^2 p_{ii}} > 0, \tag{4}$$

*then, the PP-game (1) is strongly monotone and admits a unique NE.*

The proof of Theorem 3.4 is presented in Appendix C. Since $p_{ij}$ characterizes the influence of player $j$'s decision on the data distribution of player $i$, we typically have $p_{ij} \le p_{ii}$ for $j \ne i$ and thus $\max_{j\in[n]} p_{ij} = p_{ii}$ for all $i \in [n]$. Then, the condition (4) reduces to $\mu - \sum_{i=1}^{n} L_i\varepsilon_i p_{ii} - \sqrt{\sum_{i=1}^{n} L_i^2\varepsilon_i^2 p_{ii}} > 0$. Similarly, when $L_1 = \cdots = L_n = L$, $\varepsilon_1 = \cdots = \varepsilon_n = \varepsilon$, and $p_{ij} = \frac{1}{n}$ for all $i, j \in [n]$, we require that $\mu - 2L\varepsilon > 0$, i.e., $\varepsilon \le \frac{\mu}{2L}$, which recovers the condition to guarantee the convexity of the performative risk $\mathrm{PR}(\cdot)$, and thereby the E&U of the performative optimal point of (Miller et al., 2021) for single-agent PP.

## 3.3 Distance Between PSE and NE

**Theorem 3.5.** *Define* $\widetilde{\mu} := \mu - \sum_{i=1}^{n} L_i\varepsilon_i \max_{j\in[n]} \sqrt{p_{ij}}$ *and* $\alpha := \sum_{i=1}^{n} G_i\left(1 + \varepsilon_i \max_{j\in[n]} \sqrt{p_{ij}}\right)$. *Suppose that Assumptions 2.1-2.5 hold and* $\widetilde{\mu} > 0$. *Then, for every PSE point and NE point, we have the following relations:*

$$\|\boldsymbol{\theta}^{\mathrm{pse}} - \boldsymbol{\theta}^{\mathrm{ne}}\|_2 \le \frac{1}{\widetilde{\mu}} \sqrt{\sum_{i=1}^{n} G_i^2\varepsilon_i^2 p_{ii}} \quad and \quad |\mathrm{PR}(\boldsymbol{\theta}^{\mathrm{pse}}) - \mathrm{PR}(\boldsymbol{\theta}^{\mathrm{ne}})| \le \frac{\alpha}{\widetilde{\mu}} \sqrt{\sum_{i=1}^{n} G_i^2\varepsilon_i^2 p_{ii}}.$$

The proof of Theorem 3.5 is presented in Appendix D. According to Theorem 3.5, the distance between the PSE and NE of the game (1) depends on the cost functions' parameters $\mu$, $\{G_i\}$, $\{L_i\}$, as well as the sensitivity of the data distributions $\{\varepsilon_i\}$. Larger sensitivity parameters widen the gap between the PSE and NE, while a bigger monotonicity parameter $\mu$ reduces it. Notably, when the sensitivity parameter $\varepsilon_i = 0$ for all $i \in [n]$, the game (1) reduces to a conventional stochastic game with fixed data distributions, and as a result, the PSE and NE converge to the same point.

To the best of our knowledge, this is the first result on the distance between PSE and NE of PP-games. Characterizing this distance is challenging in games due to the lack of strong convexity on the joint cost function $J(\cdot)$, which is an essential property for determining the optimality gap of performative stable points in previous work (Perdomo et al., 2020; Lu, 2023). In this paper, we characterize this gap by leveraging relations from strong duality (Boyd and Vandenberghe, 2004; Facchinei and Pang, 2010). Our result is comparable to the findings in (Perdomo et al., 2020) for single-agent PP problems wherein this optimality gap is bounded by $\frac{2L\varepsilon}{\mu}$. In our case, when $G_1 = \cdots = G_n = G$, $\varepsilon_1 = \cdots = \varepsilon_n = \varepsilon$ and $p_{ij} = \frac{1}{n}$ for all $i, j \in [n]$, we have $\|\boldsymbol{\theta}^{\mathrm{pse}} - \boldsymbol{\theta}^{\mathrm{ne}}\|_2 \le \frac{G\varepsilon}{\mu - L\varepsilon}$.

**Algorithm 1** Decentralized Stochastic Primal-Dual Algorithm: The Procedures at Player $i$, $\forall i \in [n]$:

1: Initialize $\boldsymbol{\theta}_i^1 \in \Xi_i$ arbitrarily. Set $\boldsymbol{\lambda}_i^1 = \mathbf{0}$ and $\widehat{\boldsymbol{\theta}}_{ih}^1 = \mathbf{0}$ for all $h \neq i$.
2: **for** $t = 1$ to $T$ **do**
3:     Exchange $\boldsymbol{\theta}_i^t$, $\widehat{\boldsymbol{\theta}}_i^t$, and $\boldsymbol{\lambda}_i^t$ with all neighbors;
4:     Update the estimate $\widehat{\boldsymbol{\theta}}_{ih}^t$ for all $h \neq i$ by: $\widehat{\boldsymbol{\theta}}_{ih}^{t+1} = \sum_{k \neq h} a_{ik} \widehat{\boldsymbol{\theta}}_{kh}^t + a_{ih} \boldsymbol{\theta}_h^t$;
5:     Deploy the model $\boldsymbol{\theta}_i^t$ and sample $\boldsymbol{\xi}_i^t \sim \mathcal{D}_i(\boldsymbol{\theta}_i^t, \boldsymbol{\theta}_{-i}^t)$;
6:     Update the primal variable by: $\boldsymbol{\theta}_i^{t+1} = P_{\boldsymbol{\Omega}_i} \left[ \boldsymbol{\theta}_i^t - \gamma_t \left( \nabla_{\boldsymbol{\theta}_i} J_i \left( \boldsymbol{\xi}_i^t; \boldsymbol{\theta}_i^t, \widehat{\boldsymbol{\theta}}_i^t \right) + \gamma_t \nabla \boldsymbol{g}_i(\boldsymbol{\theta}_i^t)^\top \boldsymbol{\lambda}_i^t \right) \right]$;
7:     Update the dual variable by: $\boldsymbol{\lambda}_i^{t+1} = \left[ \left(1 - \gamma_t^2\right) \sum_{j \in \mathcal{N}_i} a_{ij} \boldsymbol{\lambda}_j^t + \gamma_t \boldsymbol{g}_i \left( \boldsymbol{\theta}_i^t \right) \right]_+$.
8: **end for**

## 4 Computation of the PSE

Although RRM theoretically has the capability to find a PSE point, how to perform risk minimization at its each update remains unknown. Moreover, RRM requires the computation of an NE for each deployment, which is computationally intensive. In this section, we present a decentralized stochastic primal-dual algorithm for efficiently computing the PSE of the game (1). Theoretical analysis is also provided on the convergence of the proposed algorithm.

### 4.1 Algorithm Development

For each player $i \in [n]$, define a regularized Lagrangian as

$$\mathcal{L}_{\boldsymbol{\delta}}^{(i)}(\boldsymbol{\theta}_i, \boldsymbol{\theta}_{-i}, \boldsymbol{\lambda}) = \mathbb{E}_{\boldsymbol{\xi}_i \sim \mathcal{D}_i(\boldsymbol{\delta})} J_i \left( \boldsymbol{\xi}_i; \boldsymbol{\theta}_i, \boldsymbol{\theta}_{-i} \right) + \left\langle \boldsymbol{\lambda}, \boldsymbol{g}_i(\boldsymbol{\theta}_i) + \sum_{j \neq i} \boldsymbol{g}_j \left( \boldsymbol{\theta}_j \right) \right\rangle,$$

where $\boldsymbol{\lambda} \in \mathbb{R}_+^m$ is the dual variable. Denote by $\nabla \boldsymbol{g}_i(\cdot)$ the Jacobian matrix of $\boldsymbol{g}_i(\cdot)$. From the primal-dual theory (Boyd and Vandenberghe, 2004; Facchinei and Pang, 2010), for any $\gamma > 0$, there exists a bounded Lagrangian multiplier $\boldsymbol{\lambda}^{\text{pse}}$ such that the following condition holds:

$$\boldsymbol{\theta}_i^{\text{pse}} = P_{\boldsymbol{\Omega}_i} \left[ \boldsymbol{\theta}_i^{\text{pse}} - \gamma \left( G_{\boldsymbol{\theta}^{\text{pse}}}^{(i)} (\boldsymbol{\theta}^{\text{pse}}, \boldsymbol{\lambda}^{\text{pse}}) + \gamma \nabla \boldsymbol{g}_i(\boldsymbol{\theta}_i^{\text{pse}})^\top \boldsymbol{\lambda}^{\text{pse}} \right) \right], \quad \forall i \in [n],$$

$$\boldsymbol{\lambda}^{\text{pse}} = \left[ \boldsymbol{\lambda}^{\text{pse}} + \gamma \left( \boldsymbol{g}_i(\boldsymbol{\theta}_i^{\text{pse}}) + \sum_{j \neq i} \boldsymbol{g}_j \left( \boldsymbol{\theta}_j^{\text{pse}} \right) \right) \right]_+,$$

where $\gamma$ is a control parameter. Thus, given $\boldsymbol{\theta}_{-i}^{\text{pse}}$ and under $\boldsymbol{\xi}_i \sim \mathcal{D}_i(\boldsymbol{\theta}^{\text{pse}})$, $(\boldsymbol{\theta}_i^{\text{pse}}, \boldsymbol{\lambda}^{\text{pse}})$ is a saddle point of the Lagrangian $\mathcal{L}_{\boldsymbol{\theta}^{\text{pse}}}^{(i)}(\boldsymbol{\theta}_i, \boldsymbol{\theta}_{-i}^{\text{pse}}, \boldsymbol{\lambda})$ for any $i \in [n]$. The joint saddle point $(\boldsymbol{\theta}^{\text{pse}}, \boldsymbol{\lambda}^{\text{pse}})$ achieve the PSE of the game (1) under strong duality (Boyd and Vandenberghe, 2004).

In the decentralized noncooperative game (1), each player can only communicate with its neighbors. We use $\mathcal{G}(\mathbf{A})$ to denote the communication graph of the network, where $\mathbf{A} = (a_{ij})_{n \times n}$ represents a weight matrix. In $\mathcal{G}(\mathbf{A})$, $a_{ij} = a_{ji} > 0$ if there is a communication link between player $i$ and play $j$, and $a_{ij} = a_{ji} = 0$ otherwise. Let $\mathcal{N}_i$ be the set containing player $i$ and all its neighbors such that $j \in \mathcal{N}_i$ if $a_{ij} > 0$. We assume that the communication graph $\mathcal{G}(\mathbf{A})$ is connected and the weight matrix $\mathbf{A}$ is doubly stochastic.

To find the saddle point $(\boldsymbol{\theta}^{\text{pse}}, \boldsymbol{\lambda}^{\text{pse}})$, we develop a decentralized stochastic primal-dual algorithm, as presented in Algorithm 1. The basic idea of Algorithm 1 is to perform gradient update on the primal variables $\boldsymbol{\theta}_i$ for all $i \in [n]$ and the dual variable $\boldsymbol{\lambda}$. In the decentralized noncooperative game, each player $i \in [n]$ only observes information from its neighbors. However, its private cost funtion $J_i(\boldsymbol{\xi}_i; \boldsymbol{\theta}_i, \boldsymbol{\theta}_{-i})$ involves all players' strategies. To solve this problem, we let each player $i$ store an estimate for the strategies of all the other players, denoted by $\widehat{\boldsymbol{\theta}}_{ih}$, for all $h \neq i$. Define a vector $\widehat{\boldsymbol{\theta}}_i$ that concatenates all the estimates $\widehat{\boldsymbol{\theta}}_{ih}$. In each iteration $t$, neighbors exchange strategy $\boldsymbol{\theta}_i^t$, estimate $\widehat{\boldsymbol{\theta}}_i^t$, and dual varible $\boldsymbol{\lambda}_i^t$ with each other. Then, each player $i$ updates the estimates $\widehat{\boldsymbol{\theta}}_{ih}$, for all $h \neq i$ by weighted average in Step 4. The primal variable $\boldsymbol{\theta}_i^t$ is updated by gradient descent by Step 6, and the dual variable $\boldsymbol{\lambda}_i^t$ is updated by gradient ascent by Step 7. The coefficient $\gamma_t$ is the stepsize at the $t$th iteration for all $t \in [T]$.

## 4.2 Performance Analysis

Before analyzing the performance of Algorithm 1, we define the performance metrics adopted in this paper. The first metric is performative regret. For any player $i \in [n]$, its performative regret over $T$ iterations is defined as

$$\mathcal{R}_i(T) := \sum_{t=1}^{T} \left( \mathbb{E}_{\boldsymbol{\xi}_i \sim \mathcal{D}_i(\boldsymbol{\theta}^{\mathrm{pse}})} J_i \left( \boldsymbol{\xi}_i; \boldsymbol{\theta}_i^t, \boldsymbol{\theta}_{-i}^{\mathrm{pse}} \right) - \mathrm{PR}_i \left( \boldsymbol{\theta}^{\mathrm{pse}} \right) \right).$$

The regret $\mathcal{R}_i(T)$ measures the suboptimality of the sequence of decisions $\{\boldsymbol{\theta}_i^1, \cdots, \boldsymbol{\theta}_i^T\}$ taken by play $i$ relative to $\boldsymbol{\theta}_i^{\mathrm{pse}}$. Besides, since the decisions of all players are subject to constraints, another performance metric of constraint violation, denoted by $\mathcal{R}_g(T)$, is required, defined as

$$\mathcal{R}_g(T) = \left\| \left[ \sum_{t=1}^{T} \sum_{i=1}^{n} \boldsymbol{g}_i \left( \boldsymbol{\theta}_i^t \right) \right]_+ \right\|_2.$$

Any online or learning algorithm is regarded as "good" if both the time-average regret and the time-average constraint violation are sublinear, i.e., $\lim_{T \to \infty} \mathcal{R}_i(T)/T \leq o(1)$ for any $i \in [n]$ and $\lim_{T \to \infty} \mathcal{R}_g(T)/T \leq o(1)$.

For analysis, we make the following assumption on the variance of the stochastic gradient $\nabla_{\boldsymbol{\theta}_i} J_i (\boldsymbol{\xi}_i; \boldsymbol{\delta})$, $\forall i \in [n]$.

**Assumption 4.1.** The stochastic gradient $\nabla_{\boldsymbol{\delta}_i} J_i (\boldsymbol{\xi}_i; \boldsymbol{\delta}_i, \boldsymbol{\delta}_{-i})$ is unbiased that $\mathbb{E}_{\boldsymbol{\xi}_i \sim \mathcal{D}_i(\boldsymbol{\theta})} \nabla_{\boldsymbol{\delta}_i} J_i (\boldsymbol{\xi}_i; \boldsymbol{\delta}_i, \boldsymbol{\delta}_{-i}) = G_{\boldsymbol{\theta}}^{(i)} (\boldsymbol{\delta}_i, \boldsymbol{\delta}_{-i})$ and there exist constants $\sigma_0, \sigma_1 \geq 0$ such that $\sum_{i=1}^{n} \mathbb{E}_{\boldsymbol{\xi}_i \sim \mathcal{D}_i(\boldsymbol{\theta})} \left\| \nabla_{\boldsymbol{\delta}_i} J_i (\boldsymbol{\xi}_i; \boldsymbol{\delta}_i, \boldsymbol{\delta}_{-i}) - G_{\boldsymbol{\theta}}^{(i)} (\boldsymbol{\delta}_i, \boldsymbol{\delta}_{-i}) \right\|_2^2 \leq \sigma_0^2 + \sigma_1^2 \|\boldsymbol{\theta} - \boldsymbol{\theta}^{\mathrm{pse}}\|_2^2, \forall \boldsymbol{\theta}, \boldsymbol{\delta} \in \boldsymbol{\Omega}$.

**Theorem 4.2.** *Define* $\widetilde{\mu} := \mu - \sum_{i=1}^{n} L_i \varepsilon_i \max_{j \in [n]} \sqrt{p_{ij}}$ *and* $\nu := 3 \left( \sigma_1^2 + 3 \sum_{i=1}^{n} L_i^2 \left( 1 + \varepsilon_i^2 \max_{j \in [n]} p_{ij} \right) \right)$. *Suppose that Assumptions 2.1-2.5 and 4.1 hold and* $\widetilde{\mu} > 0$. *By Algorithm 1, if the stepsize satisfies* $\sup_{t \in [T]} \gamma_t \leq \frac{\widetilde{\mu}}{\nu}$, *then, the performative regret of the game* (1) *is bounded by*

$$\mathcal{R}_i(T) \leq \mathcal{O}\left( \sqrt{\frac{T}{\widetilde{\mu}} \left( \frac{1}{\gamma_T} + \sum_{t=1}^{T} \gamma_t \right)} \right), \forall i \in [n].$$

*Further, the constraint violation is bounded by*

$$\mathcal{R}_g(T) \leq \mathcal{O}\left( \frac{1}{\gamma_T} \sqrt{\left( \frac{1}{\gamma_T} + \sum_{t=1}^{T} \gamma_t \right) \left( 1 + \sum_{t=1}^{T} \gamma_t^2 \right)} \right).$$

For a sequence of diminishing stepsize $\gamma_t = \tau_1^\eta (\tau_2 t + \tau_1)^{-\eta}$, where $\tau_1, \tau_2 > 0$ and $0 < \eta < 1$, we have that: 1) $\sum_{t=1}^{T} \gamma_t \leq \mathcal{O}\left( T^{1-\eta} \right)$; 2) $\sum_{t=1}^{T} \gamma^2(t) \leq \mathcal{O}\left( T^{1-2\eta} \right)$. Plugging the above results into Theorem 4.2 yields

$$\mathcal{R}_i(T) \leq \mathcal{O}\left( T^{\frac{1+\eta}{2}} + T^{1-\frac{\eta}{2}} \right), i \in [n] \quad \text{and} \quad \mathcal{R}_g(T) \leq \mathcal{O}\left( T^{\frac{3}{2}\eta} + T^{\frac{1+\eta}{2}} + T^{1-\frac{\eta}{2}} \right).$$

Based on the above two inequalities, the best choice of $\eta$ is $\frac{1}{2}$ such that $\mathcal{R}_i(T) \leq \mathcal{O}(T^{\frac{3}{4}}), \forall i \in [n]$ and $\mathcal{R}_g(T) \leq \mathcal{O}(T^{\frac{3}{4}})$. This convergence speed matches that of the decentralized noncooperative game without performativity (Lu et al., 2020).

The proof of Theorem 4.2 is provided in Appendix E. According to Theorem 4.2, the performative effect reduces the convergence rate by amplifying the coefficient $\frac{1}{\widetilde{\mu}}$ in the regret bounds. Specifically, as the sensitivity parameters $\varepsilon_i$ increase, the coefficient $\widetilde{\mu}$ decreases, leading to a slower convergence rate of $\mathcal{R}_i(T)$ for all $i \in [n]$. This occurs because a larger $\varepsilon_i$ indicates a stronger performative influence, which more significantly impacts the algorithm's convergence. Nevertheless, the performative effect does not degrade the convergence order of Algorithm 1 compared to the case without performativity (Lu et al., 2020).

## 5 Numerical Experiments

In this section, we evaluate the effectiveness of our algorithm and theoretical results by conducting numerical experiments on a networked Cournot game (Abolhassani et al., 2014), which is a foundational model in economic theory (Allaz and Vila, 1993) for analyzing oligopolistic competitions. We

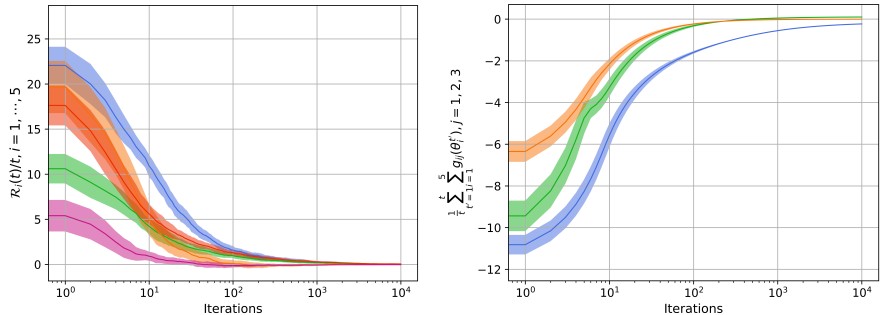

Figure 1: Convergence of time-average regrets and time-average constraint violations.

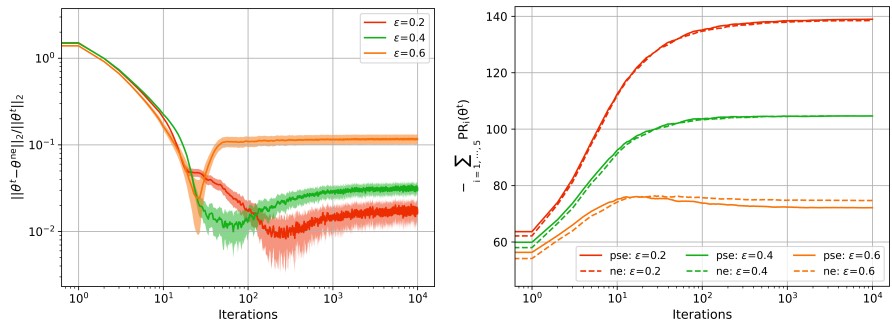

Figure 2: (a). Normalized distance between $\boldsymbol{\theta}^t$ and $\boldsymbol{\theta}^{\mathrm{ne}}$. (b). Total revenue at PSE and NE.

consider a networked Cournot game with five firms selling a single commodity across three markets. Each firm aims to maximize its profit by determining the quantities it serves in all markets. The total accommodated quantity in each market is limited by its market capacity. The simulation details and additional numerical results are presented in Appendix F.1. We also provide an additional experiment on a ride-share market in Appendix F.2.

Fig. 1 illustrates the convergence of the time-average regrets of five firms, denoted by $\mathcal{R}_i(t)/t$, $\forall i \in [5]$, and the convergence of the time-average constraint violations of three markets, denoted by $\frac{1}{t}\sum_{t'=1}^{t}\sum_{i=1}^{n}g_{ij}(\boldsymbol{\theta}_i^{t'})$, $\forall j \in [3]$. The results demonstrate that both $\mathcal{R}_i(t)/t$ and $\frac{1}{t}\sum_{t'=1}^{t}\sum_{i=1}^{n}g_{ij}(\boldsymbol{\theta}_i^{t'})$ approach zero as the iterations increase. This verifies the sublinear convergence of the regrets and constraint violations in Theorem 4.2.

Fig. 2 (a) compares the normalized distance between $\boldsymbol{\theta}^t$, generated by Algorithm 1, and the NE point $\boldsymbol{\theta}^{\mathrm{ne}}$, denoted as $\|\boldsymbol{\theta}^t - \boldsymbol{\theta}^{\mathrm{ne}}\|_2/\|\boldsymbol{\theta}^t\|_2$. The NE point is computed based on perfect knowledge of $\{\mathcal{D}_i\}_{i\in[n]}$. We consider three different performative strengths: $\varepsilon = 0.2$, $0.4$, and $0.6$. It is observed that $\|\boldsymbol{\theta}^t - \boldsymbol{\theta}^{\mathrm{ne}}\|_2/\|\boldsymbol{\theta}^t\|_2$ stabilizes at values approximately equal to or smaller than $10^{-1}$ with iterations, varifying the effectiveness of Algorithm 1. Additionally, a larger performative strength leads to a wider normalized distance between the convergent point of $\boldsymbol{\theta}^t$ and $\boldsymbol{\theta}^{\mathrm{ne}}$. In Fig. 2 (b), we compare the total revenues, denoted by $-\sum_{i=1}^{5}\mathrm{PR}_i(\boldsymbol{\theta}^t)$ under the same three $\varepsilon$ settings. We consider two scenarios: 1). "pse", where $\boldsymbol{\theta}^t$ is generated by Algorithm 1; 2). "ne", where $\boldsymbol{\theta}^t$ is generated by performing the same procedures as Algorithm 1 but with perfect information on the distributions $\{\mathcal{D}_i(\boldsymbol{\theta})\}_{i\in[n]}$. The result demonstrates the close performance of the "pse" approach and the "ne" approach. More numerical results can be found in Appendix F.

**Conclusions:** We have studied the performative phenomenon in a decentralized noncooperative game where selfish players seek to maximize their individual profits while adhering to coupled inequality constraints. We have derived sufficient conditions for the E&U of both PSE and NE and provided the first upper bound on the distance between these two equilibria. Furthermore, we have developed a decentralized stochastic primal-dual algorithm for efficiently computing of the PSE point. Theoretical analysis has demonstrated the same order of convergence speed of our algorithm as the case without performativity. Finally, numerical simulations have been provided to verify the effectiveness of our algorithm and theoretical results.

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

# A   Related Work

In recent years, the exploration of distribution shifts in machine learning systems has been extended beyond traditional exogenous shifts (Quinonero-Candela et al., 2008), such as covariate (Chan et al., 2020), label (Wu et al., 2021), and concept (Lu et al., 2018) drifts, to include endogenous shifts resulting from strategic behaviors within the learning platforms themselves. Perdomo et al. (2020) introduced the framework of performative prediction, which captures the platform's strategic responses using decision-dependent distribution mappings. Following this seminal work, significant research effort has been dedicated to investigating the phenomenon of performativity in various scenarios. In particular, (Shan et al., 2023) studied the endogenous distribution change in open environments, where data are obtained from a corrupted decision-dependent distribution. They proposed an effective algorithm with theoretical guarantees by decoupling the two sources of effects. Lu (2023) investigated the presence of performativity in bilevel optimization. They first established sufficient conditions for the existence of performatively stable solutions and then developed a stochastic algorithm to find the PS point. In (Mandal et al., 2023), the authors examined the performative effect in a regularized reinforcement learning problem and showed that repeatedly optimizing this objective converges to a performatively stable policy under reasonable assumptions on the transition dynamics. It is demonstrated in (Drusvyatskiy and Xiao, 2023) that typical gradient-based stochastic algorithms can be applied to find performative stable equilibria with a biased gradient oracle.

While most existing work focused on finding performative stable points, there are studies aimed at identifying the optimal solutions for performative prediction problems (Miller et al., 2021; Izzo et al., 2021; Jagadeesan et al., 2022). The optimality gap of performative stable points was first presented in (Perdomo et al., 2020), where their bound is proportional to the strong convexity parameter and inversely proportional to the smoothness parameter of cost functions and the sensitivity parameter of the decision-dependent distributions. The primary challenges in computing optimal points in performative prediction problems lie in the unknown decision-dependent data distributions. To address this challenge, a commonly used method is to make parametric assumptions on the data distributions and then design algorithms to estimate them. For instance, (Miller et al., 2021) proposed a two-stage algorithm to find the performative optima for distribution maps in the location family. Izzo et al. (2021) proposed a PerfGD algorithm by exploiting the exponential structure of the underlying distribution maps.

Among the numerous existing studies, (Narang et al., 2023) and (Wang et al., 2023) are, at a conceptual level, the closest papers to our own since they have considered performative behaviors in games. On a technical level, however, these two works are quite distinct from ours since we study completely different problem settings. One defining distinction is that, in our model, the collective strategies of all players must adhere to the learning system's constraints, whereas both (Narang et al., 2023) and (Wang et al., 2023) are unconstrained. Constraints are unavoidable in certain game scenarios, such as safety and cost constraints in transportation, relevance and diversity constraints in advertising, and risk tolerance and portfolio constraints in financial trading. The constrained problem in our work results in a fundamentally different algorithm design and convergence analysis from these two papers. Our work utilizes the primal-dual technique and necessitates consensus, whereas their approach only requires local stochastic gradient descent. Additionally, there are distinctions in the problem settings. In (Wang et al., 2023), the private cost function of each player is structured in a specific form, involving a local cost depending solely on its own strategy and a regularizer quantifying the similarity of strategies among neighbors. In contrast, we consider a mathematically richer setting where each player's private cost function depends on the strategies of all players in the game, thus encompassing the model in (Wang et al., 2023). Moreover, our algorithm design takes into account the practical implementation where players can only communicate with their neighbors, while (Narang et al., 2023) assumes that the strategies of all players are publicly accessible across the entire network. This more practical setting poses challenges for each player in observing the entire network. More importantly, although (Narang et al., 2023) and (Wang et al., 2023) demonstrated the existence and uniqueness of the PSE and NE for their respective game settings, neither of them offers insights into the distance between these two equilibria. This paper makes a significant contribution by presenting the first upper bound on this distance.

Furthermore, there are works on decentralized optimization of multiagent performative prediction (Li et al., 2022; Piliouras and Yu, 2023). Specifically, (Li et al., 2022) focused on decentralized

optimization with consensus-seeking agents, where the data distribution of each agent depends only on its own decision. Although (Piliouras and Yu, 2023) considers multiagent, their study is in a centralized fashion and their data distributions are restricted to location-scale families. Lastly, it is worth mentioning that one paper (Yan and Cao, 2024b) has considered constrained optimization in the context of performative prediction. However, (Yan and Cao, 2024b) studied the single-agent case, while this work considers a more complex model with decentralized noncooperative players and partially observed information about competitors' strategies. Additionally, this paper contributes to the evaluation of equilibria, whereas such analysis has not been involved in (Yan and Cao, 2024b).

## B  Existence and Uniqueness of Performative Stable Equilibrium

From the definition of the mapping $\mathcal{T}(\boldsymbol{\theta})$, we have that

$$\boldsymbol{\theta}_i' = \mathcal{T}_i(\boldsymbol{\theta}) = \underset{\boldsymbol{u}_i \in \boldsymbol{\Omega}_i}{\arg\min} \quad \mathbb{E}_{\boldsymbol{\xi}_i \sim \mathcal{D}_i(\boldsymbol{\theta})} J_i\left(\boldsymbol{\xi}_i; \boldsymbol{u}_i, \boldsymbol{\theta}_{-i}'\right) \quad \text{s.t.} \quad \boldsymbol{g}_i(\boldsymbol{u}_i) + \sum_{j \neq i} \boldsymbol{g}_j\left(\boldsymbol{\theta}_j'\right) \leq \boldsymbol{0}, \quad \forall i \in [n],$$

$$\boldsymbol{\delta}_i' = \mathcal{T}_i(\boldsymbol{\delta}) = \underset{\boldsymbol{u}_i \in \boldsymbol{\Omega}_i}{\arg\min} \quad \mathbb{E}_{\boldsymbol{\xi}_i \sim \mathcal{D}_i(\boldsymbol{\delta})} J_i\left(\boldsymbol{\xi}_i; \boldsymbol{u}_i, \boldsymbol{\delta}_{-i}'\right) \quad \text{s.t.} \quad \boldsymbol{g}_i(\boldsymbol{u}_i) + \sum_{j \neq i} \boldsymbol{g}_j\left(\boldsymbol{\delta}_j'\right) \leq \boldsymbol{0}, \quad \forall i \in [n].$$

Define $\mathbb{E}_{\boldsymbol{\xi}_i \sim \mathcal{D}_i(\boldsymbol{\theta})} \nabla_{\boldsymbol{\theta}_i} J_i\left(\boldsymbol{\xi}_i; \boldsymbol{\theta}_i', \boldsymbol{\theta}_{-i}'\right) := G_{\boldsymbol{\theta}}^{(i)}(\boldsymbol{\theta}_i', \boldsymbol{\theta}_{-i}')$. From the optimality condition of constrained optimization, we have

$$\left\langle G_{\boldsymbol{\theta}}^{(i)}\left(\boldsymbol{\theta}'\right), \boldsymbol{\theta}_i' - \boldsymbol{\delta}_i'\right\rangle \leq 0, \quad \forall i \in [n].$$

Define a vector $G_{\boldsymbol{\theta}}(\boldsymbol{\theta}') := \text{col}\left(G_{\boldsymbol{\theta}}^{(1)}(\boldsymbol{\theta}'), \cdots, G_{\boldsymbol{\theta}}^{(n)}(\boldsymbol{\theta}')\right)$ that concatenates all the $G_{\boldsymbol{\theta}}^{(i)}(\boldsymbol{\theta}')$, $i \in [n]$. Then, we have

$$\left\langle G_{\boldsymbol{\theta}}\left(\boldsymbol{\theta}'\right), \boldsymbol{\theta}' - \boldsymbol{\delta}'\right\rangle \leq 0. \tag{A1}$$

Similarly, we have

$$\left\langle G_{\boldsymbol{\delta}}\left(\boldsymbol{\delta}'\right), \boldsymbol{\theta}' - \boldsymbol{\delta}'\right\rangle \geq 0. \tag{A2}$$

Further, from the monotoniticy of the gradient mapping $\nabla J\left(\boldsymbol{\xi}; \boldsymbol{\theta}\right)$ in Assumption 2.1, we have

$$\left\langle G_{\boldsymbol{\theta}}(\boldsymbol{\theta}') - G_{\boldsymbol{\theta}}(\boldsymbol{\delta}'), \boldsymbol{\theta}' - \boldsymbol{\delta}'\right\rangle = \mathbb{E}_{\boldsymbol{\xi} \sim \mathcal{D}(\boldsymbol{\theta})} \left\langle \nabla J\left(\boldsymbol{\xi}; \boldsymbol{\theta}'\right) - \nabla J\left(\boldsymbol{\xi}; \boldsymbol{\delta}'\right), \boldsymbol{\theta}' - \boldsymbol{\delta}'\right\rangle \geq \mu \|\boldsymbol{\theta}' - \boldsymbol{\delta}'\|_2^2, \tag{A3}$$

where $\mathcal{D}(\boldsymbol{\theta}) := \mathcal{D}_1(\boldsymbol{\theta}) \times \cdots \times \mathcal{D}_n(\boldsymbol{\theta})$. Plugging (A1) and (A2) into (A3) gives

$$\begin{aligned}
\mu\|\boldsymbol{\theta}' - \boldsymbol{\delta}'\|_2^2 &\leq \left\langle -G_{\boldsymbol{\theta}}\left(\boldsymbol{\delta}'\right), \boldsymbol{\theta}' - \boldsymbol{\delta}'\right\rangle \\
&\leq \left\langle G_{\boldsymbol{\delta}}\left(\boldsymbol{\delta}'\right) - G_{\boldsymbol{\theta}}\left(\boldsymbol{\delta}'\right), \boldsymbol{\theta}' - \boldsymbol{\delta}'\right\rangle \\
&\leq \left\| G_{\boldsymbol{\delta}}\left(\boldsymbol{\delta}'\right) - G_{\boldsymbol{\theta}}\left(\boldsymbol{\delta}'\right) \right\|_2 \left\| \boldsymbol{\theta}' - \boldsymbol{\delta}' \right\|_2.
\end{aligned} \tag{A4}$$

From Assumption 2.2, $\mathcal{W}_1\left(\mathcal{D}_i\left(\boldsymbol{\theta}\right), \mathcal{D}_i\left(\boldsymbol{\theta}'\right)\right) \leq \varepsilon_i \sqrt{\sum_{j=1}^n p_{ij} \left\|\boldsymbol{\theta}_j - \boldsymbol{\theta}_j'\right\|_2^2}$. Along with Assumption 2.4, we have that

$$\begin{aligned}
\left\| G_{\boldsymbol{\delta}}\left(\boldsymbol{\delta}'\right) - G_{\boldsymbol{\theta}}\left(\boldsymbol{\delta}'\right) \right\|_2^2 &\leq \sum_{i=1}^n \sum_{j=1}^n L_i^2 \varepsilon_i^2 p_{ij} \left\|\boldsymbol{\delta}_j - \boldsymbol{\theta}_j\right\|_2^2 \\
&\leq \sum_{i=1}^n L_i^2 \varepsilon_i^2 \max_{j \in [n]} p_{ij} \left\|\boldsymbol{\delta} - \boldsymbol{\theta}\right\|_2^2.
\end{aligned}$$

Plugging the above result into (A4) yields

$$\|\boldsymbol{\theta}' - \boldsymbol{\delta}'\|_2 \leq \frac{1}{\mu} \sqrt{\sum_{i=1}^n L_i^2 \varepsilon_i^2 \max_{j \in [n]} p_{ij}} \left\|\boldsymbol{\delta} - \boldsymbol{\theta}\right\|_2.$$

From the RRM procedure, we know that $\boldsymbol{\theta}^{t+1} = \mathcal{T}(\boldsymbol{\theta}^t)$ and the PSE satisfies $\boldsymbol{\theta}^{\text{pse}} = \mathcal{T}(\boldsymbol{\theta}^{\text{pse}})$. Then, we have

$$\left\|\boldsymbol{\theta}^{t+1} - \boldsymbol{\theta}^{\text{pse}}\right\|_2 \leq \frac{1}{\mu}\sqrt{\sum_{i=1}^{n} L_i^2 \varepsilon_i^2 \max_{j \in [n]} p_{ij}} \left\|\boldsymbol{\theta}^t - \boldsymbol{\theta}^{\text{pse}}\right\|_2$$

$$\leq \left(\frac{1}{\mu}\sqrt{\sum_{i=1}^{n} L_i^2 \varepsilon_i^2 \max_{j \in [n]} p_{ij}}\right)^t \left\|\boldsymbol{\theta}^1 - \boldsymbol{\theta}^{\text{pse}}\right\|_2.$$

Further, if for any player $i$, its distribution $\mathcal{D}_i$ depends only on its own decision $\boldsymbol{\theta}_i$, i.e., $p_{ij} = 0$ and $p_{ii} = 1$ for all $i, j \in [n]$ and $j \neq i$, then, we have

$$\left\|\left(G_{\boldsymbol{\delta}}\left(\boldsymbol{\delta}'\right) - G_{\boldsymbol{\theta}}\left(\boldsymbol{\delta}'\right)\right)\right\|_2 \leq \sqrt{\sum_{i=1}^{n} L_i^2 \varepsilon_i^2 \left\|\boldsymbol{\delta}_i - \boldsymbol{\theta}_i\right\|_2^2} \leq \max_{i \in [n]} L_i \varepsilon_i \left\|\boldsymbol{\delta} - \boldsymbol{\theta}\right\|_2. \tag{A5}$$

Plugging (A5) into (A4) yields

$$\left\|\boldsymbol{\theta}' - \boldsymbol{\delta}'\right\|_2 \leq \frac{1}{\mu} \max_{i \in [n]} L_i \varepsilon_i \left\|\boldsymbol{\delta} - \boldsymbol{\theta}\right\|_2.$$

Correspondingly, we have

$$\left\|\boldsymbol{\theta}^{t+1} - \boldsymbol{\theta}^{\text{pse}}\right\|_2 \leq \left(\frac{1}{\mu} \max_{i \in [n]} L_i \varepsilon_i\right)^t \left\|\boldsymbol{\theta}^1 - \boldsymbol{\theta}^{\text{pse}}\right\|_2.$$

## C   Existence and Uniqueness of Nash Equilibrium

Based on the results in Facchinei and Pang (2003, Theorem 2.3.3(b)), to show the existence and uniqueness of NE, we need to prove that the gradient mapping $\nabla\text{PR}(\boldsymbol{\theta})$ of the performative game (1) is strongly monotone, i.e., there exists a $\alpha > 0$ such that $\langle\nabla\text{PR}(\boldsymbol{\theta}) - \nabla\text{PR}(\boldsymbol{\theta}), \boldsymbol{\theta} - \boldsymbol{\delta}\rangle \geq \alpha \|\boldsymbol{\theta} - \boldsymbol{\delta}\|_2^2$, where $\alpha$ denotes the strongly-monotone parameter. Since $\nabla\text{PR}(\boldsymbol{\theta}) = G_{\boldsymbol{\theta}}(\boldsymbol{\theta}) + H_{\boldsymbol{\theta}}(\boldsymbol{\theta})$, we have

$$\langle\nabla\text{PR}(\boldsymbol{\theta}) - \nabla\text{PR}(\boldsymbol{\delta}), \boldsymbol{\theta} - \boldsymbol{\delta}\rangle = \langle G_{\boldsymbol{\theta}}(\boldsymbol{\theta}) - G_{\boldsymbol{\delta}}(\boldsymbol{\delta}), \boldsymbol{\theta} - \boldsymbol{\delta}\rangle + \langle H_{\boldsymbol{\theta}}(\boldsymbol{\theta}) - H_{\boldsymbol{\delta}}(\boldsymbol{\delta}), \boldsymbol{\theta} - \boldsymbol{\delta}\rangle.$$

From Assumption 2.2, we have

$$\langle G_{\boldsymbol{\theta}}(\boldsymbol{\theta}) - G_{\boldsymbol{\delta}}(\boldsymbol{\theta}), \boldsymbol{\theta} - \boldsymbol{\delta}\rangle \geq -\sum_{i=1}^{n} L_i \varepsilon_i \max_{j \in [n]} \sqrt{p_{ij}} \|\boldsymbol{\theta} - \boldsymbol{\delta}\|_2^2.$$

Moreover, from the monotonicity of the gradient mapping $\nabla J(\boldsymbol{\xi}; \boldsymbol{\theta})$ in Assumption 2.1, we have

$$\langle G_{\boldsymbol{\delta}}(\boldsymbol{\theta}) - G_{\boldsymbol{\delta}}(\boldsymbol{\delta}), \boldsymbol{\theta} - \boldsymbol{\delta}\rangle = \mathbb{E}_{\boldsymbol{\xi}\sim\mathcal{D}(\boldsymbol{\delta})} \langle\nabla J(\boldsymbol{\xi}; \boldsymbol{\theta}) - \nabla J(\boldsymbol{\xi}; \boldsymbol{\delta}), \boldsymbol{\theta} - \boldsymbol{\delta}\rangle \geq \mu \|\boldsymbol{\theta} - \boldsymbol{\delta}\|_2^2.$$

Combining the above two inequalities yields

$$\langle G_{\boldsymbol{\theta}}(\boldsymbol{\theta}) - G_{\boldsymbol{\delta}}(\boldsymbol{\delta}), \boldsymbol{\theta} - \boldsymbol{\delta}\rangle = \langle G_{\boldsymbol{\theta}}(\boldsymbol{\theta}) - G_{\boldsymbol{\delta}}(\boldsymbol{\theta}), \boldsymbol{\theta} - \boldsymbol{\delta}\rangle + \langle G_{\boldsymbol{\delta}}(\boldsymbol{\theta}) - G_{\boldsymbol{\delta}}(\boldsymbol{\delta}), \boldsymbol{\theta} - \boldsymbol{\delta}\rangle$$

$$\geq \left(\mu - \sum_{i=1}^{n} L_i \varepsilon_i \max_{j \in [n]} \sqrt{p_{ij}}\right) \|\boldsymbol{\theta} - \boldsymbol{\delta}\|_2^2. \tag{A6}$$

Further, let $\gamma(s) = \boldsymbol{\theta}' + s\left(\boldsymbol{\theta} - \boldsymbol{\theta}'\right)$ for $s \in (0, 1)$. Then, we have

$$J_i\left(\boldsymbol{\xi}_i; \boldsymbol{\theta}\right) - J_i\left(\boldsymbol{\xi}_i; \boldsymbol{\theta}'\right) = \int_0^1 \left\langle\nabla J_i\left(\boldsymbol{\xi}_i; \boldsymbol{\theta}' + s\left(\boldsymbol{\theta} - \boldsymbol{\theta}'\right)\right), \boldsymbol{\theta} - \boldsymbol{\theta}'\right\rangle \mathrm{d}s$$

$$= \int_0^1 \left\langle\nabla J_i\left(\boldsymbol{\xi}_i; \gamma(s)\right), \boldsymbol{\theta} - \boldsymbol{\theta}'\right\rangle \mathrm{d}s. \tag{A7}$$

From the definition of $H_{\boldsymbol{\theta}}^{(i)}(\boldsymbol{\delta})$ that $H_{\boldsymbol{\theta}}^{(i)}(\boldsymbol{\delta}) := \nabla_{\boldsymbol{u}_i} \mathbb{E}_{\boldsymbol{\xi}_i \sim \mathcal{D}_i(\boldsymbol{u}_i, \boldsymbol{\theta}_{-i})} \left[ J_i\left( \boldsymbol{\xi}_i; \boldsymbol{\delta} \right) \right]\big|_{\boldsymbol{u}_i = \boldsymbol{\theta}_i}$, we have that

$$
\begin{aligned}
H_{\boldsymbol{\theta}}^{(i)}(\boldsymbol{\theta}) - H_{\boldsymbol{\theta}}^{(i)}(\boldsymbol{\theta}') &= \nabla_{\boldsymbol{u}_i} \mathbb{E}_{\boldsymbol{\xi}_i \sim \mathcal{D}_i(\boldsymbol{u}_i, \boldsymbol{\theta}_{-i})} \left[ \int_0^1 \left\langle \nabla J_i\left( \boldsymbol{\xi}_i; \gamma(s) \right), \boldsymbol{\theta} - \boldsymbol{\theta}' \right\rangle \mathrm{d}s \right]\Bigg|_{\boldsymbol{u}_i = \boldsymbol{\theta}_i} \\
&= \int_0^1 \nabla_{\boldsymbol{u}_i} \mathbb{E}_{\boldsymbol{\xi}_i \sim \mathcal{D}_i(\boldsymbol{u}_i, \boldsymbol{\theta}_{-i})} \left\langle \nabla J_i\left( \boldsymbol{\xi}_i; \gamma(s) \right), \boldsymbol{\theta} - \boldsymbol{\theta}' \right\rangle\Bigg|_{\boldsymbol{u}_i = \boldsymbol{\theta}_i} \mathrm{d}s. \quad \text{(A8)}
\end{aligned}
$$

From Assumption 2.4, we have

$$
\left\| \mathbb{E}_{\boldsymbol{\xi}_i \sim \mathcal{D}_i} \nabla J_i\left( \boldsymbol{\xi}_i; \boldsymbol{\theta} \right) - \mathbb{E}_{\boldsymbol{\xi}_i' \sim \mathcal{D}_i'} \nabla J_i\left( \boldsymbol{\xi}_i'; \boldsymbol{\theta} \right) \right\|_2 \le L_i \mathcal{W}_1(\mathcal{D}_i, \mathcal{D}_i').
$$

Along with Assumption 2.2, we know that the function $\mathbb{E}_{\boldsymbol{\xi}_i \sim \mathcal{D}_i(\boldsymbol{\theta}_i, \boldsymbol{\theta}_{-i})} \nabla J_i\left( \boldsymbol{\xi}_i; \boldsymbol{\theta}' \right)$ is $L_i \varepsilon_i p_{ii}$-Lipschitz continuous w.r.t $\boldsymbol{\theta}_i$, and thus its gradient satisfies

$$
\left\| \nabla_{\boldsymbol{u}_i} \mathbb{E}_{\boldsymbol{\xi}_i \sim \mathcal{D}_i(\boldsymbol{u}_i, \boldsymbol{\theta}_{-i})} \left[ \nabla J_i\left( \boldsymbol{\xi}_i; \gamma(s) \right) \right]\big|_{\boldsymbol{u}_i = \boldsymbol{\theta}_i} \right\|_2 \le L_i \varepsilon_i p_{ii}. \quad \text{(A9)}
$$

Combing (A8) and (A9) gives

$$
\begin{aligned}
\left\| H_{\boldsymbol{\theta}}^{(i)}(\boldsymbol{\theta}) - H_{\boldsymbol{\theta}}^{(i)}(\boldsymbol{\theta}') \right\|_2 &\le \int_0^1 \left\| \nabla_{\boldsymbol{u}_i} \mathbb{E}_{\boldsymbol{\xi}_i \sim \mathcal{D}_i(\boldsymbol{u}_i, \boldsymbol{\theta}_{-i})} \left[ \nabla J_i\left( \boldsymbol{\xi}_i; \gamma(s) \right) \right]\big|_{\boldsymbol{u}_i = \boldsymbol{\theta}_i} \right\|_2 \left\| \boldsymbol{\theta} - \boldsymbol{\theta}' \right\|_2 \mathrm{d}s \\
&\le L_i \varepsilon_i p_{ii} \left\| \boldsymbol{\theta} - \boldsymbol{\theta}' \right\|_2,
\end{aligned}
$$

where the first inequality holds due to the Cauchy-Schwartz inequality. This further implies that

$$
\begin{aligned}
\left\| H_{\boldsymbol{\theta}}(\boldsymbol{\theta}) - H_{\boldsymbol{\theta}}(\boldsymbol{\theta}') \right\|_2 &= \sqrt{\sum_{i=1}^n \left\| H_{\boldsymbol{\theta}}^{(i)}(\boldsymbol{\theta}) - H_{\boldsymbol{\theta}}^{(i)}(\boldsymbol{\theta}') \right\|_2^2} \\
&\le \sqrt{\sum_{i=1}^n L_i^2 \varepsilon_i^2 p_{ii}} \left\| \boldsymbol{\theta} - \boldsymbol{\theta}' \right\|_2.
\end{aligned}
$$

Following prior work (Narang et al., 2023) and (Wang et al., 2023) on performative games, we assume that the mapping $H_{\boldsymbol{\delta}}(\boldsymbol{\theta})$ is monotone w.r.t $\boldsymbol{\delta}$, i.e., $\langle H_{\boldsymbol{\theta}}(\boldsymbol{\theta}) - H_{\boldsymbol{\delta}}(\boldsymbol{\theta}), \boldsymbol{\theta} - \boldsymbol{\delta} \rangle \ge 0$. Then, we have that

$$
\begin{aligned}
\langle \nabla \mathrm{PR}(\boldsymbol{\theta}) - \nabla \mathrm{PR}(\boldsymbol{\delta}), \boldsymbol{\theta} - \boldsymbol{\delta} \rangle &= \langle G_{\boldsymbol{\theta}}(\boldsymbol{\theta}) - G_{\boldsymbol{\delta}}(\boldsymbol{\delta}), \boldsymbol{\theta} - \boldsymbol{\delta} \rangle + \langle H_{\boldsymbol{\theta}}(\boldsymbol{\theta}) - H_{\boldsymbol{\delta}}(\boldsymbol{\delta}), \boldsymbol{\theta} - \boldsymbol{\delta} \rangle \\
&= \langle G_{\boldsymbol{\theta}}(\boldsymbol{\theta}) - G_{\boldsymbol{\delta}}(\boldsymbol{\theta}), \boldsymbol{\theta} - \boldsymbol{\delta} \rangle + \langle H_{\boldsymbol{\theta}}(\boldsymbol{\theta}) - H_{\boldsymbol{\theta}}(\boldsymbol{\delta}), \boldsymbol{\theta} - \boldsymbol{\delta} \rangle \\
&\quad + \langle G_{\boldsymbol{\delta}}(\boldsymbol{\theta}) - G_{\boldsymbol{\delta}}(\boldsymbol{\delta}), \boldsymbol{\theta} - \boldsymbol{\delta} \rangle + \langle H_{\boldsymbol{\delta}}(\boldsymbol{\theta}) - H_{\boldsymbol{\delta}}(\boldsymbol{\delta}), \boldsymbol{\theta} - \boldsymbol{\delta} \rangle \\
&\ge \left( \mu - \sum_{i=1}^n L_i \varepsilon_i \max_{j \in [n]} \sqrt{p_{ij}} - \sqrt{\sum_{i=1}^n L_i^2 \varepsilon_i^2 p_{ii}} \right) \left\| \boldsymbol{\theta} - \boldsymbol{\delta} \right\|_2^2.
\end{aligned}
$$

Based on the classical result that a strongly monotone game over a non-empty, closed, and convex set admits a unique NE Facchinei and Pang (2003, Theorem 2.3.3(b)), we have the E&U condition for the NE of the game (1) as given in theorem 3.4.

## D  Distance Between PSE and NE

The computation on the distance between the PSE and NE of the game (1) is based on the strong duality (Boyd and Vandenberghe, 2004; Facchinei and Pang, 2010). Recall the definitions in Section 4.1 that

$$
\mathcal{L}_{\boldsymbol{\delta}}^{(i)}(\boldsymbol{\theta}_i, \boldsymbol{\theta}_{-i}, \boldsymbol{\lambda}) := \mathbb{E}_{\boldsymbol{\xi}_i \sim \mathcal{D}_i(\boldsymbol{\delta})} J_i\left( \boldsymbol{\xi}_i; \boldsymbol{\theta}_i, \boldsymbol{\theta}_{-i} \right) + \left\langle \boldsymbol{\lambda}, \boldsymbol{g}_i(\boldsymbol{\theta}_i) + \sum_{j \ne i} \boldsymbol{g}_j\left( \boldsymbol{\theta}_j \right) \right\rangle.
$$

Moreover, define a gradient mapping $\phi_i(\boldsymbol{\xi}_i; \boldsymbol{\theta}, \boldsymbol{\lambda}) := \nabla_{\boldsymbol{\theta}_i} J_i\left( \boldsymbol{\xi}_i; \boldsymbol{\theta} \right) + \nabla \boldsymbol{g}_i(\boldsymbol{\theta}_i)^\top \boldsymbol{\lambda}$ and a concatenation vector $\boldsymbol{\phi} := [\phi_1, \cdots, \phi_n]^\top$. For any $i \in [n]$, since $(\boldsymbol{\theta}_i^{\mathrm{pse}}, \boldsymbol{\lambda}^{\mathrm{pse}})$ is a saddle point of the Lagrangian $\mathcal{L}_{\boldsymbol{\theta}^{\mathrm{pse}}}^{(i)}(\boldsymbol{\theta}_i, \boldsymbol{\theta}_{-i}^{\mathrm{pse}}, \boldsymbol{\lambda})$ under $\boldsymbol{\xi}_i \sim \mathcal{D}_i(\boldsymbol{\theta}^{\mathrm{pse}})$, we have that

$$
\mathcal{L}_{\boldsymbol{\theta}^{\mathrm{pse}}}^{(i)}\left( \boldsymbol{\theta}_i^{\mathrm{pse}}, \boldsymbol{\theta}_{-i}^{\mathrm{pse}}, \boldsymbol{\lambda} \right) \le \mathcal{L}_{\boldsymbol{\theta}^{\mathrm{pse}}}^{(i)}\left( \boldsymbol{\theta}_i^{\mathrm{pse}}, \boldsymbol{\theta}_{-i}^{\mathrm{pse}}, \boldsymbol{\lambda}^{\mathrm{pse}} \right) \le \mathcal{L}_{\boldsymbol{\theta}^{\mathrm{pse}}}^{(i)}\left( \boldsymbol{\theta}_i, \boldsymbol{\theta}_{-i}^{\mathrm{pse}}, \boldsymbol{\lambda}^{\mathrm{pse}} \right) \quad \forall \boldsymbol{\theta}_i \in \boldsymbol{\Omega}_i, \boldsymbol{\lambda} \in \mathbb{R}_+^m.
$$

Similarly, for any $i \in [n]$, $(\boldsymbol{\theta}_i^{\text{ne}}, \boldsymbol{\lambda}^{\text{ne}})$ the saddle point of the regularized Lagrangian $\mathcal{L}_{\boldsymbol{\theta}_i, \boldsymbol{\theta}_{-i}^{\text{ne}}}^{(i)}(\boldsymbol{\theta}_i, \boldsymbol{\theta}_{-i}^{\text{ne}}, \boldsymbol{\lambda})$ with decision-dependent distribution $\boldsymbol{\xi}_i \sim \mathcal{D}_i(\boldsymbol{\theta}_i, \boldsymbol{\theta}_{-i}^{\text{ne}})$. Setting $\boldsymbol{\lambda} = \boldsymbol{\lambda}^{\text{ne}}$ in the first part of the proceeding inequality, we obtain

$$0 \leq \mathcal{L}_{\boldsymbol{\theta}^{\text{pse}}}^{(i)}\left(\boldsymbol{\theta}_i^{\text{pse}}, \boldsymbol{\theta}_{-i}^{\text{pse}}, \boldsymbol{\lambda}^{\text{pse}}\right) - \mathcal{L}_{\boldsymbol{\theta}^{\text{pse}}}^{(i)}\left(\boldsymbol{\theta}_i^{\text{pse}}, \boldsymbol{\theta}_{-i}^{\text{pse}}, \boldsymbol{\lambda}^{\text{ne}}\right) = (\boldsymbol{\lambda}^{\text{pse}} - \boldsymbol{\lambda}^{\text{ne}})^\top \boldsymbol{g}\left(\boldsymbol{\theta}^{\text{pse}}\right), \forall i \in [n],$$

where $(\boldsymbol{\lambda}^{\text{pse}} - \boldsymbol{\lambda}^{\text{ne}})^\top \boldsymbol{g}\left(\boldsymbol{\theta}^{\text{pse}}\right) = \sum_{j=1}^m \left(\lambda_j^{\text{pse}} - \lambda_j^{\text{ne}}\right)\left(\sum_{i=1}^n g_{ji}\left(\boldsymbol{\theta}_i^{\text{pse}}\right)\right)$. By the convexity of $g_{ji}(\cdot)$ for all $j \in [m], i \in [n]$, we have that

$$\sum_{i=1}^n g_{ji}\left(\boldsymbol{\theta}_i^{\text{pse}}\right) \leq \sum_{i=1}^n \left(g_{ji}\left(\boldsymbol{\theta}_i^{\text{ne}}\right) + \left\langle \nabla g_{ji}\left(\boldsymbol{\theta}_i^{\text{pse}}\right), \boldsymbol{\theta}_i^{\text{pse}} - \boldsymbol{\theta}_i^{\text{ne}}\right\rangle\right)$$

$$\leq \sum_{i=1}^n \left\langle \nabla g_{ji}\left(\boldsymbol{\theta}_i^{\text{pse}}\right), \boldsymbol{\theta}_i^{\text{pse}} - \boldsymbol{\theta}_i^{\text{ne}}\right\rangle, \forall j \in [m],$$

where the last inequality follows from that $g_j(\boldsymbol{\theta}^{\text{ne}}) = \sum_{i=1}^n g_{ji}\left(\boldsymbol{\theta}_i^{\text{ne}}\right) \leq 0$. Multiplying the preceding inequality with $\lambda_j^{\text{pse}}$ and adding over all $j \in [m]$, we obtain

$$\sum_{j=1}^m \sum_{i=1}^n \lambda_j^{\text{pse}} g_{ji}\left(\boldsymbol{\theta}_i^{\text{pse}}\right) = (\boldsymbol{\lambda}^{\text{pse}})^\top \boldsymbol{g}\left(\boldsymbol{\theta}^{\text{pse}}\right) \leq \sum_{i=1}^n \left\langle \sum_{j=1}^m \lambda_j^{\text{pse}} \nabla g_{ji}\left(\boldsymbol{\theta}_i^{\text{pse}}\right), \boldsymbol{\theta}_i^{\text{pse}} - \boldsymbol{\theta}_i^{\text{ne}}\right\rangle$$

$$= \sum_{i=1}^n \left\langle \nabla \boldsymbol{g}_i(\boldsymbol{\theta}_i^{\text{pse}})^\top \boldsymbol{\lambda}^{\text{pse}}, \boldsymbol{\theta}_i^{\text{pse}} - \boldsymbol{\theta}_i^{\text{ne}}\right\rangle. \quad \text{(A10)}$$

By the definition of the mapping $\phi_i(\cdot)$, for any $\boldsymbol{\xi}_i \in \boldsymbol{\Xi}_i$, we have that,

$$\nabla \boldsymbol{g}_i(\boldsymbol{\theta}_i^{\text{pse}})^\top \boldsymbol{\lambda}^{\text{pse}} = \phi_i(\boldsymbol{\xi}_i; \boldsymbol{\theta}^{\text{pse}}, \boldsymbol{\lambda}^{\text{pse}}) - \nabla_{\boldsymbol{\theta}_i} J_i\left(\boldsymbol{\xi}_i; \boldsymbol{\theta}^{\text{pse}}\right), \forall i \in [n]. \quad \text{(A11)}$$

Plugging (A11) into (A10) gives

$$(\boldsymbol{\lambda}^{\text{pse}})^\top \boldsymbol{g}\left(\boldsymbol{\theta}^{\text{pse}}\right) \leq \sum_{i=1}^n \left\langle \phi_i(\boldsymbol{\xi}_i; \boldsymbol{\theta}^{\text{pse}}, \boldsymbol{\lambda}^{\text{pse}}) - \nabla_{\boldsymbol{\theta}_i} J_i\left(\boldsymbol{\xi}_i; \boldsymbol{\theta}^{\text{pse}}\right), \boldsymbol{\theta}_i^{\text{pse}} - \boldsymbol{\theta}_i^{\text{ne}}\right\rangle, \forall i \in [n]. \quad \text{(A12)}$$

Likewise, we have the following inequality based on the convexity of the functions $\{g_{ji}(\cdot)\}$:

$$g_{ji}\left(\boldsymbol{\theta}_i^{\text{pse}}\right) \geq g_{ji}\left(\boldsymbol{\theta}_i^{\text{ne}}\right) + \left\langle \nabla g_{ji}\left(\boldsymbol{\theta}_i^{\text{ne}}\right), \boldsymbol{\theta}_i^{\text{pse}} - \boldsymbol{\theta}_i^{\text{ne}}\right\rangle, \forall j \in [m], i \in [n].$$

Multiplying the preceding inequality with $-\lambda_j^{\text{ne}}$ and summing over $j \in [m]$, we obtain

$$-\sum_{j=1}^m \lambda_i^{\text{ne}} \sum_{i=1}^n g_{ji}\left(\boldsymbol{\theta}_i^{\text{pse}}\right) \leq -\sum_{j=1}^m \lambda_j^{\text{ne}} \sum_{i=1}^n g_{ji}\left(\boldsymbol{\theta}_i^{\text{ne}}\right) - \sum_{i=1}^n \left\langle \sum_{j=1}^m \lambda_j^{\text{ne}} \nabla g_{ji}\left(\boldsymbol{\theta}_i^{\text{ne}}\right), \boldsymbol{\theta}_i^{\text{pse}} - \boldsymbol{\theta}_i^{\text{ne}}\right\rangle$$

$$= \sum_{i=1}^n \left\langle \nabla \boldsymbol{g}_i\left(\boldsymbol{\theta}_i^{\text{ne}}\right)^\top \boldsymbol{\lambda}^{\text{ne}}, \boldsymbol{\theta}_i^{\text{ne}} - \boldsymbol{\theta}_i^{\text{pse}}\right\rangle,$$

where the equality follows from that $\sum_{j=1}^m \lambda_j^{\text{ne}} \sum_{i=1}^n g_{ji}\left(\boldsymbol{\theta}_i^{\text{ne}}\right) = (\boldsymbol{\lambda}^{\text{ne}})^\top \boldsymbol{g}\left(\boldsymbol{\theta}^{\text{ne}}\right) = 0$, which holds by the complementary slackness condition of the Lagrangian $\mathcal{L}_{\boldsymbol{\theta}_i, \boldsymbol{\theta}_{-i}^{\text{ne}}}^{(i)}(\boldsymbol{\theta}_i, \boldsymbol{\theta}_{-i}^{\text{ne}}, \boldsymbol{\lambda})$ for all $i \in [n]$. Similar to (A12), we have

$$-(\boldsymbol{\lambda}^{\text{ne}})^\top \boldsymbol{g}\left(\boldsymbol{\theta}^{\text{pse}}\right) \leq \sum_{i=1}^n \left\langle \phi_i(\boldsymbol{\xi}_i; \boldsymbol{\theta}^{\text{ne}}, \boldsymbol{\lambda}^{\text{ne}}) - \nabla_{\boldsymbol{\theta}_i} J_i\left(\boldsymbol{\xi}_i; \boldsymbol{\theta}^{\text{ne}}\right), \boldsymbol{\theta}_i^{\text{ne}} - \boldsymbol{\theta}_i^{\text{pse}}\right\rangle. \quad \text{(A13)}$$

Combining (A12) and (A13) yields

$$(\boldsymbol{\lambda}^{\text{pse}} - \boldsymbol{\lambda}^{\text{ne}})^\top \boldsymbol{g}\left(\boldsymbol{\theta}^{\text{pse}}\right) \leq \sum_{i=1}^n \left\langle \phi_i(\boldsymbol{\xi}_i; \boldsymbol{\theta}^{\text{pse}}, \boldsymbol{\lambda}^{\text{pse}}) - \phi_i(\boldsymbol{\xi}_i; \boldsymbol{\theta}^{\text{ne}}, \boldsymbol{\lambda}^{\text{ne}}), \boldsymbol{\theta}_i^{\text{pse}} - \boldsymbol{\theta}_i^{\text{ne}}\right\rangle$$

$$- \sum_{i=1}^n \left\langle \nabla_{\boldsymbol{\theta}_i} J_i\left(\boldsymbol{\xi}_i; \boldsymbol{\theta}^{\text{pse}}\right) - \nabla_{\boldsymbol{\theta}_i} J_i\left(\boldsymbol{\xi}_i; \boldsymbol{\theta}^{\text{ne}}\right), \boldsymbol{\theta}_i^{\text{pse}} - \boldsymbol{\theta}_i^{\text{ne}}\right\rangle.$$

Taking expectation on both sides of the above inequality over the distribution $\mathcal{D}_i(\boldsymbol{\theta}^{\mathrm{pse}})$ for all $i \in [n]$ gives

$$(\boldsymbol{\lambda}^{\mathrm{pse}} - \boldsymbol{\lambda}^{\mathrm{ne}})^\top \boldsymbol{g}\left(\boldsymbol{\theta}^{\mathrm{pse}}\right) \leq \sum_{i=1}^{n} \mathbb{E}_{\boldsymbol{\xi}_i \sim \mathcal{D}_i(\boldsymbol{\theta}^{\mathrm{pse}})} \left\langle \phi_i(\boldsymbol{\xi}_i; \boldsymbol{\theta}^{\mathrm{pse}}, \boldsymbol{\lambda}^{\mathrm{pse}}) - \phi_i(\boldsymbol{\xi}_i; \boldsymbol{\theta}^{\mathrm{ne}}, \boldsymbol{\lambda}^{\mathrm{ne}}), \boldsymbol{\theta}_i^{\mathrm{pse}} - \boldsymbol{\theta}_i^{\mathrm{ne}} \right\rangle$$
$$- \sum_{i=1}^{n} \left\langle G_{\boldsymbol{\theta}^{\mathrm{pse}}}^{(i)}\left(\boldsymbol{\theta}^{\mathrm{pse}}\right) - G_{\boldsymbol{\theta}^{\mathrm{pse}}}^{(i)}\left(\boldsymbol{\theta}^{\mathrm{ne}}\right), \boldsymbol{\theta}_i^{\mathrm{pse}} - \boldsymbol{\theta}_i^{\mathrm{ne}} \right\rangle. \tag{A14}$$

Since $(\boldsymbol{\theta}_i^{\mathrm{pse}}, \boldsymbol{\lambda}^{\mathrm{pse}})$ is a saddle point of the Lagrangian $\mathcal{L}_{\boldsymbol{\theta}^{\mathrm{pse}}}^{(i)}(\boldsymbol{\theta}^{\mathrm{pse}}, \boldsymbol{\lambda}^{\mathrm{pse}})$ given $\boldsymbol{\xi}_i \sim \mathcal{D}_i(\boldsymbol{\theta}^{\mathrm{pse}})$, we have that

$$\mathbb{E}_{\boldsymbol{\xi}_i \sim \mathcal{D}_i(\boldsymbol{\theta}^{\mathrm{pse}})} \left\langle \phi_i(\boldsymbol{\xi}_i; \boldsymbol{\theta}^{\mathrm{pse}}, \boldsymbol{\lambda}^{\mathrm{pse}}), \boldsymbol{\theta}_i^{\mathrm{pse}} - \boldsymbol{\theta}_i^{\mathrm{ne}} \right\rangle \leq 0, \forall i \in [n]. \tag{A15}$$

Furthermore, for any $i \in [n]$, we have

$$\mathbb{E}_{\boldsymbol{\xi}_i \sim \mathcal{D}_i(\boldsymbol{\theta}^{\mathrm{pse}})} \phi_i(\boldsymbol{\xi}_i; \boldsymbol{\theta}^{\mathrm{ne}}, \boldsymbol{\lambda}^{\mathrm{ne}}) = G_{\boldsymbol{\theta}^{\mathrm{pse}}}^{(i)}\left(\boldsymbol{\theta}^{\mathrm{ne}}\right) + \nabla \boldsymbol{g}_i(\boldsymbol{\theta}_i^{\mathrm{ne}})^\top \boldsymbol{\lambda}^{\mathrm{ne}}$$
$$+ \nabla_{\boldsymbol{\theta}_i} \mathrm{PR}_i(\boldsymbol{\theta}_i^{\mathrm{ne}}, \boldsymbol{\theta}_{-i}^{\mathrm{ne}}) - \nabla_{\boldsymbol{\theta}_i} \mathrm{PR}_i(\boldsymbol{\theta}_i^{\mathrm{ne}}, \boldsymbol{\theta}_{-i}^{\mathrm{ne}}). \tag{A16}$$

Since $(\boldsymbol{\theta}_i^{\mathrm{ne}}, \boldsymbol{\lambda}^{\mathrm{ne}})$ is a saddle point of the Lagrangian $\mathcal{L}_{\boldsymbol{\theta}_i, \boldsymbol{\theta}_{-i}^{\mathrm{ne}}}^{(i)}(\boldsymbol{\theta}_i, \boldsymbol{\theta}_{-i}^{\mathrm{ne}}, \boldsymbol{\lambda}^{\mathrm{ne}})$ with decision-dependent distribution $\mathcal{D}_i(\boldsymbol{\theta}_i, \boldsymbol{\theta}_{-i}^{\mathrm{ne}})$, we have that

$$-\left\langle \nabla_{\boldsymbol{\theta}_i} \mathrm{PR}_i(\boldsymbol{\theta}_i^{\mathrm{ne}}, \boldsymbol{\theta}_{-i}^{\mathrm{ne}}) + \nabla \boldsymbol{g}_i(\boldsymbol{\theta}_i^{\mathrm{ne}})^\top \boldsymbol{\lambda}^{\mathrm{ne}}, \boldsymbol{\theta}_i^{\mathrm{pse}} - \boldsymbol{\theta}_i^{\mathrm{ne}} \right\rangle \leq 0, \forall i \in [n]. \tag{A17}$$

Plugging (A15), (A16), and (A17) into (A14) yields

$$0 \leq (\boldsymbol{\lambda}^{\mathrm{pse}} - \boldsymbol{\lambda}^{\mathrm{ne}})^\top \boldsymbol{g}\left(\boldsymbol{\theta}^{\mathrm{pse}}\right)$$
$$\leq \sum_{i=1}^{n} \left\langle \nabla_i \mathrm{PR}_i(\boldsymbol{\theta}_i^{\mathrm{ne}}, \boldsymbol{\theta}_{-i}^{\mathrm{ne}}) - G_{\boldsymbol{\theta}^{\mathrm{pse}}}^{(i)}\left(\boldsymbol{\theta}^{\mathrm{ne}}\right), \boldsymbol{\theta}_i^{\mathrm{pse}} - \boldsymbol{\theta}_i^{\mathrm{ne}} \right\rangle$$
$$- \sum_{i=1}^{n} \left\langle G_{\boldsymbol{\theta}^{\mathrm{pse}}}^{(i)}\left(\boldsymbol{\theta}^{\mathrm{pse}}\right) - G_{\boldsymbol{\theta}^{\mathrm{pse}}}^{(i)}\left(\boldsymbol{\theta}^{\mathrm{ne}}\right), \boldsymbol{\theta}_i^{\mathrm{pse}} - \boldsymbol{\theta}_i^{\mathrm{ne}} \right\rangle$$
$$= \sum_{i=1}^{n} \left\langle H_{\boldsymbol{\theta}^{\mathrm{ne}}}^{(i)}\left(\boldsymbol{\theta}^{\mathrm{ne}}\right) + G_{\boldsymbol{\theta}^{\mathrm{ne}}}^{(i)}\left(\boldsymbol{\theta}^{\mathrm{ne}}\right) - G_{\boldsymbol{\theta}^{\mathrm{pse}}}^{(i)}\left(\boldsymbol{\theta}^{\mathrm{pse}}\right), \boldsymbol{\theta}_i^{\mathrm{pse}} - \boldsymbol{\theta}_i^{\mathrm{ne}} \right\rangle.$$

Then, we have

$$\left\langle G_{\boldsymbol{\theta}^{\mathrm{pse}}}\left(\boldsymbol{\theta}^{\mathrm{pse}}\right) - G_{\boldsymbol{\theta}^{\mathrm{ne}}}\left(\boldsymbol{\theta}^{\mathrm{ne}}\right), \boldsymbol{\theta}^{\mathrm{pse}} - \boldsymbol{\theta}^{\mathrm{ne}} \right\rangle \leq \left\langle H_{\boldsymbol{\theta}^{\mathrm{ne}}}(\boldsymbol{\theta}^{\mathrm{ne}}), \boldsymbol{\theta}^{\mathrm{pse}} - \boldsymbol{\theta}^{\mathrm{ne}} \right\rangle.$$

From the result in (A6) and the Cauchy-Schwarz inequality, we have

$$\left(\mu - \sum_{i=1}^{n} L_i \varepsilon_i \max_{j \in [n]} \sqrt{p_{ij}}\right) \|\boldsymbol{\theta}^{\mathrm{pse}} - \boldsymbol{\theta}^{\mathrm{ne}}\|_2^2 \leq \|H_{\boldsymbol{\theta}^{\mathrm{ne}}}(\boldsymbol{\theta}^{\mathrm{ne}})\|_2 \|\boldsymbol{\theta}^{\mathrm{pse}} - \boldsymbol{\theta}^{\mathrm{ne}}\|_2.$$

Since the cost function $J_i(\cdot)$ is $G_i$ Lipschitz for any $i \in [n]$, along with Assumption 2.2, we have

$$\|H_{\boldsymbol{\theta}^{\mathrm{ne}}}(\boldsymbol{\theta}^{\mathrm{ne}})\|_2 = \sqrt{\sum_{i=1}^{n} \|H_{\boldsymbol{\theta}_i^{\mathrm{ne}}, \boldsymbol{\theta}_{-i}^{\mathrm{ne}}}^{(i)}(\boldsymbol{\theta}_i^{\mathrm{ne}}, \boldsymbol{\theta}_{-i}^{\mathrm{ne}})\|_2^2} \leq \sqrt{\sum_{i=1}^{n} G_i^2 \varepsilon_i^2 p_{ii}}.$$

Combining the above results yields

$$\|\boldsymbol{\theta}^{\mathrm{pse}} - \boldsymbol{\theta}^{\mathrm{ne}}\|_2 \leq \frac{\sqrt{\sum_{i=1}^{n} G_i^2 \varepsilon_i^2 p_{ii}}}{\mu - \sum_{i=1}^{n} L_i \varepsilon_i \max_{j \in [n]} \sqrt{p_{ij}}}.$$

Further, from Assumption 2.2, we have

$$|\mathrm{PR}_i(\boldsymbol{\theta}^{\mathrm{pse}}) - \mathrm{PR}_i(\boldsymbol{\theta}^{\mathrm{ne}})| \leq G_i \|\boldsymbol{\theta}^{\mathrm{pse}} - \boldsymbol{\theta}^{\mathrm{ne}}\|_2 + G_i \varepsilon_i \sqrt{\sum_{j=1}^{n} p_{ij} \left\|\boldsymbol{\theta}_j^{\mathrm{pse}} - \boldsymbol{\theta}_j^{\mathrm{ne}}\right\|_2^2}$$
$$\leq G_i \left(1 + \varepsilon_i \max_{j \in [n]} \sqrt{p_{ij}}\right) \|\boldsymbol{\theta}^{\mathrm{pse}} - \boldsymbol{\theta}^{\mathrm{ne}}\|_2.$$

Then, we have

$$
\begin{aligned}
|\mathrm{PR}(\boldsymbol{\theta}^{\mathrm{pse}}) - \mathrm{PR}(\boldsymbol{\theta}^{\mathrm{ne}})| &= \sum_{i=1}^{n} |\mathrm{PR}_i(\boldsymbol{\theta}^{\mathrm{pse}}) - \mathrm{PR}_i(\boldsymbol{\theta}^{\mathrm{ne}})| \\
&\leq \left( \sum_{i=1}^{n} G_i \left( 1 + \varepsilon_i \max_{j \in [n]} \sqrt{p_{ij}} \right) \right) \frac{\sqrt{\sum_{i=1}^{n} G_i^2 \varepsilon_i^2 p_{ii}}}{\mu - \sum_{i=1}^{n} L_i \varepsilon_i \max_{j \in [n]} \sqrt{p_{ij}}}.
\end{aligned}
$$

# E  Convergence of the Decentralized Stochastic Primal-Dual Algorithm

The proof of this section utilizes the following supporting lemmas.

**Lemma E.1.** *Based on the update rule of the dual variable $\boldsymbol{\lambda}$ in Algorithm 1, for any $\gamma_t \geq 0$, $\boldsymbol{\lambda}_i^t \in \mathbb{R}_+^m$, $i \in [n]$, and $t \in [T]$, we have that $\sum_{i=1}^{n} \|\gamma_t \boldsymbol{\lambda}_i^t\|_2^2 \leq nB^2$.*

**Lemma E.2.** *Define $\overline{\boldsymbol{\lambda}}^t := \frac{1}{n} \sum_{i=1}^{n} \boldsymbol{\lambda}_i^t$ the average of the dual variable over all players at the $t$th iteration. Then, for any $\gamma_t \geq 0$ and $t \in [T]$, we have the following relationship:*

$$
\begin{aligned}
-\sum_{t=1}^{T} \sum_{i=1}^{n} \gamma_t (\boldsymbol{\lambda}_i^t)^\top \boldsymbol{g}_i(\boldsymbol{\theta}_i^t) \leq {}& -\sum_{t=1}^{T} \sum_{i=1}^{n} \gamma_t \boldsymbol{\lambda}^\top \boldsymbol{g}_i\left(\boldsymbol{\theta}_i^t\right) + \frac{n}{2} \left( 1 + \sum_{t=1}^{T} \gamma_t^2 \right) \|\boldsymbol{\lambda}\|_2^2 + \frac{9}{2} \sum_{t=1}^{T} \sum_{i=1}^{n} \left\| \boldsymbol{\lambda}_i^t - \overline{\boldsymbol{\lambda}}^t \right\|_2^2 \\
&+ 2(1 + \sqrt{n}) B \sum_{t=1}^{T} \gamma_t \sum_{i=1}^{n} \left\| \boldsymbol{\lambda}_i^t - \overline{\boldsymbol{\lambda}}^t \right\|_2 + 4nB^2 \sum_{t=1}^{T} \gamma_t^2.
\end{aligned}
$$

Moreover, we require the following Lemma on the weight matrix $\mathbf{A}$.

**Lemma E.3.** *Let $\sigma_2(\mathbf{A})$ denote the second-largest eigenvalue of the weight matrix $\mathbf{A}$. Since $\mathbf{A}$ is assumed to be doubly stochastic, it holds that $\sigma_2(\mathbf{A}) < 1$ (Horn and Johnson, 2012). Furthermore, for any $i \in [n]$, we construct a weight matrix $\mathbf{A}_i^-$ by removing the $i$th row and column of $\mathbf{A}$. Let $\beta$ represent the maximum eigenvalue of $\mathbf{A}_i^-$ for all $i \in [n]$. It has been established in Hong et al. (2006, Lemma 3) that $\beta < 1$.*

With Lemma E.3, we have the following results.

**Lemma E.4.** *Define $e_{ih}^t := \widehat{\boldsymbol{\theta}}_{ih}^t - \boldsymbol{\theta}_h^t$ the estimation error of player $i$ on the decision of player $h$ at the $t$th iteration, for all $i, h \in [n]$ and $t \in [T]$. Let $e_h^t$ denote the concatenation of $e_{ih}^t$ that $e_h^t := \mathrm{col}\left( e_{1h}^t, \cdots, e_{(h-1)h}^t, e_{(h+1)h}^t, \cdots, e_{nh}^t \right)$. Then, the sum of $\|e_h^t\|_2$ over $h \in [n]$ and $t \in [T]$ satisfies*

$$
\sum_{t=1}^{T} \sum_{h=1}^{n} \mathbb{E}\|e_h^t\|_2 \leq \frac{nC}{1-\beta} + \frac{n\sqrt{n-1}(G + \sqrt{n}BG_g)}{1-\beta} \sum_{t=1}^{T} \gamma_t = \mathcal{O}\left( \sum_{t=1}^{T} \gamma_t \right).
$$

*Moreover, the sum of $\|e_{ih}^t\|_2^2$ over $h \in [n]$ and $t \in [T]$ satisfies*

$$
\sum_{t=1}^{T} \sum_{h=1}^{n} \mathbb{E}\|e_h^t\|_2^2 \leq \frac{2nC^2}{1-\beta} + \frac{2n(n-1)(G + \sqrt{n}BG_g)^2}{(1-\beta)^2} \sum_{t=1}^{T} \gamma_t = \mathcal{O}\left( \sum_{t=1}^{T} \gamma_t \right).
$$

**Lemma E.5.** *With the definition $\overline{\boldsymbol{\lambda}}^t := \frac{1}{n} \sum_{i=1}^{n} \boldsymbol{\lambda}_i^t$, we have the following relationship on the consensus error of the dual variable $\boldsymbol{\lambda}_i^t$, given by $\boldsymbol{\lambda}_i^t - \overline{\boldsymbol{\lambda}}^t$, for all $i \in [n]$ and $t \in [T]$:*

$$
\begin{aligned}
\sum_{t=1}^{T} \sum_{i=1}^{n} \left\| \boldsymbol{\lambda}_i^t - \overline{\boldsymbol{\lambda}}^t \right\|_2 &\leq \frac{2(n + \sqrt{n})B}{1 - \sigma_2(\mathbf{A})} \sum_{t=1}^{T} \gamma_t = \mathcal{O}\left( \sum_{t=1}^{T} \gamma_t \right), \\
\sum_{t=1}^{T} \sum_{i=1}^{n} \left\| \boldsymbol{\lambda}_i^t - \overline{\boldsymbol{\lambda}}^t \right\|_2^2 &\leq \frac{4(n + \sqrt{n})^2 B^2}{(1 - \sigma_2(\mathbf{A}))^2} \sum_{t=1}^{T} \gamma_t = \mathcal{O}\left( \sum_{t=1}^{T} \gamma_t \right).
\end{aligned}
$$

Next, we start the proof of Theorem 4.2. For ease of proposition, we define the following gradient mappings: for any $t \in [T]$, $\phi_i^t(\boldsymbol{\xi}_i; \boldsymbol{\theta}_i, \boldsymbol{\theta}_{-i}, \boldsymbol{\lambda}) := \nabla_i J_i(\boldsymbol{\xi}_i; \boldsymbol{\theta}_i, \boldsymbol{\theta}_{-i}, \boldsymbol{\theta}) + \gamma_t \nabla g_i(\boldsymbol{\theta}_i)^\top \boldsymbol{\lambda}$, $\phi^t(\cdot) := [\phi_1^t(\cdot), \cdots, \phi_n^t(\cdot)]^\top$, $\Phi_{\boldsymbol{\delta}}^{i,t}(\boldsymbol{\theta}, \boldsymbol{\lambda}) := G_{\boldsymbol{\delta}}^{(i)}(\boldsymbol{\theta}) + \gamma_t \nabla g_i(\boldsymbol{\theta}_i)^\top \boldsymbol{\lambda}$, and $\Phi_{\boldsymbol{\delta}}^t(\boldsymbol{\theta}, \boldsymbol{\lambda}) := \left[\Phi_{\boldsymbol{\delta}}^{1,t}(\boldsymbol{\theta}, \boldsymbol{\lambda}), \cdots, \Phi_{\boldsymbol{\delta}}^{n,t}(\boldsymbol{\theta}, \boldsymbol{\lambda})\right]^\top$. Then, we have

$$
\mathbb{E}\left\|\boldsymbol{\theta}^{t+1} - \boldsymbol{\theta}^{\mathrm{pse}}\right\|_2^2 = \sum_{i=1}^n \mathbb{E}\left\|P_{\boldsymbol{\Omega}_i}\left[\boldsymbol{\theta}_i^t - \gamma_t \phi_i^t\left(\boldsymbol{\xi}_i^t; \boldsymbol{\theta}_i^t, \widehat{\boldsymbol{\theta}}_i^t, \boldsymbol{\lambda}_i^t\right)\right] - P_{\boldsymbol{\Omega}_i}\left[\boldsymbol{\theta}_i^{\mathrm{pse}} - \gamma_t \Phi_{\boldsymbol{\theta}^{\mathrm{pse}}}^{i,t}(\boldsymbol{\theta}^{\mathrm{pse}}, \boldsymbol{\lambda}^{\mathrm{pse}})\right]\right\|_2^2
$$

$$
\leq \mathbb{E}\left\|\boldsymbol{\theta}^t - \boldsymbol{\theta}^{\mathrm{pse}}\right\|_2^2 + \gamma_t^2 \sum_{i=1}^n \mathbb{E}\left\|\phi_i^t\left(\boldsymbol{\xi}_i^t; \boldsymbol{\theta}_i^t, \widehat{\boldsymbol{\theta}}_i^t, \boldsymbol{\lambda}_i^t\right) - \Phi_{\boldsymbol{\theta}^{\mathrm{pse}}}^{i,t}(\boldsymbol{\theta}^{\mathrm{pse}}, \boldsymbol{\lambda}^{\mathrm{pse}})\right\|_2^2
$$

$$
- 2\gamma_t \sum_{i=1}^n \mathbb{E}\left\langle \boldsymbol{\theta}_i^t - \boldsymbol{\theta}_i^{\mathrm{pse}}, \phi_i^t\left(\boldsymbol{\xi}_i^t; \boldsymbol{\theta}_i^t, \widehat{\boldsymbol{\theta}}_i^t, \boldsymbol{\lambda}_i^t\right) - \Phi_{\boldsymbol{\theta}^{\mathrm{pse}}}^{i,t}(\boldsymbol{\theta}^{\mathrm{pse}}, \boldsymbol{\lambda}^{\mathrm{pse}})\right\rangle. \quad \text{(A18)}
$$

The second term on the right side of (A18) is handled as follows.

$$
\gamma_t^2 \sum_{i=1}^n \mathbb{E}\left\|\phi_i^t\left(\boldsymbol{\xi}_i^t; \boldsymbol{\theta}_i^t, \widehat{\boldsymbol{\theta}}_i^t, \boldsymbol{\lambda}_i^t\right) - \Phi_{\boldsymbol{\theta}^{\mathrm{pse}}}^{i,t}(\boldsymbol{\theta}^{\mathrm{pse}}, \boldsymbol{\lambda}^{\mathrm{pse}})\right\|_2^2
$$

$$
= \gamma_t^2 \sum_{i=1}^n \mathbb{E}\left\|\phi_i^t\left(\boldsymbol{\xi}_i^t; \boldsymbol{\theta}_i^t, \widehat{\boldsymbol{\theta}}_i^t, \boldsymbol{\lambda}_i^t\right) - \Phi_{\boldsymbol{\theta}^t}^{i,t}\left(\boldsymbol{\theta}_i^t, \widehat{\boldsymbol{\theta}}_i^t, \boldsymbol{\lambda}_i^t\right) + \Phi_{\boldsymbol{\theta}^t}^{i,t}\left(\boldsymbol{\theta}_i^t, \widehat{\boldsymbol{\theta}}_i^t, \boldsymbol{\lambda}_i^t\right) - \Phi_{\boldsymbol{\theta}^{\mathrm{pse}}}^{i,t}(\boldsymbol{\theta}^{\mathrm{pse}}, \boldsymbol{\lambda}^{\mathrm{pse}})\right\|_2^2
$$

$$
\leq \underbrace{3\gamma_t^2 \sum_{i=1}^n \mathbb{E}\left\|\phi_i^t\left(\boldsymbol{\xi}_i^t; \boldsymbol{\theta}_i^t, \widehat{\boldsymbol{\theta}}_i^t, \boldsymbol{\lambda}_i^t\right) - \Phi_{\boldsymbol{\theta}^t}^{i,t}\left(\boldsymbol{\theta}_i^t, \widehat{\boldsymbol{\theta}}_i^t, \boldsymbol{\lambda}_i^t\right)\right\|_2^2}_{(a)} + \underbrace{3\gamma_t^2 \sum_{i=1}^n \mathbb{E}\left\|G_{\boldsymbol{\theta}^t}^{(i)}\left(\boldsymbol{\theta}_i^t, \widehat{\boldsymbol{\theta}}_i^t\right) - G_{\boldsymbol{\theta}^{\mathrm{pse}}}^{(i)}(\boldsymbol{\theta}^{\mathrm{pse}})\right\|_2^2}_{(b)}
$$

$$
+ \underbrace{3\gamma_t^4 \sum_{i=1}^n \mathbb{E}\left\|\nabla g_i\left(\boldsymbol{\theta}_i^t\right)^\top \boldsymbol{\lambda}_i^t - \nabla g_i(\boldsymbol{\theta}_i^{\mathrm{pse}})^\top \boldsymbol{\lambda}^{\mathrm{pse}}\right\|_2^2}_{(c)}. \quad \text{(A19)}
$$

We have the following results on these three terms in the last inequality of (A19).

$$
(a) = 3\gamma_t^2 \sum_{i=1}^n \mathbb{E}\left\|\nabla_{\boldsymbol{\theta}_i} J_i\left(\boldsymbol{\xi}_i^t; \boldsymbol{\theta}_i^t, \widehat{\boldsymbol{\theta}}_i^t\right) - G_{\boldsymbol{\theta}^t}^{(i)}\left(\boldsymbol{\theta}_i^t, \widehat{\boldsymbol{\theta}}_i^t\right)\right\|_2^2
$$

$$
\leq 3\gamma_t^2\left(\sigma_0^2 + \sigma_1^2 \mathbb{E}\left\|\boldsymbol{\theta}^t - \boldsymbol{\theta}^{\mathrm{pse}}\right\|_2^2\right).
$$

$$
(b) = 3\gamma_t^2 \sum_{i=1}^n \mathbb{E}\left\|G_{\boldsymbol{\theta}^t}^{(i)}\left(\boldsymbol{\theta}_i^t, \widehat{\boldsymbol{\theta}}_i^t\right) - G_{\boldsymbol{\theta}^t}^{(i)}\left(\boldsymbol{\theta}^t\right) + G_{\boldsymbol{\theta}^t}^{(i)}\left(\boldsymbol{\theta}^t\right) - G_{\boldsymbol{\theta}^t}^{(i)}(\boldsymbol{\theta}^{\mathrm{pse}}) + G_{\boldsymbol{\theta}^t}^{(i)}(\boldsymbol{\theta}^{\mathrm{pse}}) - G_{\boldsymbol{\theta}^{\mathrm{pse}}}^{(i)}(\boldsymbol{\theta}^{\mathrm{pse}})\right\|_2^2
$$

$$
\leq 9\gamma_t^2 \sum_{i=1}^n \mathbb{E}\left(\left\|G_{\boldsymbol{\theta}^t}^{(i)}\left(\boldsymbol{\theta}_i^t, \widehat{\boldsymbol{\theta}}_i^t\right) - G_{\boldsymbol{\theta}^t}^{(i)}\left(\boldsymbol{\theta}^t\right)\right\|_2^2 + \left\|G_{\boldsymbol{\theta}^t}^{(i)}\left(\boldsymbol{\theta}^t\right) - G_{\boldsymbol{\theta}^t}^{(i)}(\boldsymbol{\theta}^{\mathrm{pse}})\right\|_2^2 + \left\|G_{\boldsymbol{\theta}^t}^{(i)}(\boldsymbol{\theta}^{\mathrm{pse}}) - G_{\boldsymbol{\theta}^{\mathrm{pse}}}^{(i)}(\boldsymbol{\theta}^{\mathrm{pse}})\right\|_2^2\right)
$$

$$
\leq 9\gamma_t^2 \sum_{i=1}^n \mathbb{E}\left(L_i^2\left\|\widehat{\boldsymbol{\theta}}_i^t - \boldsymbol{\theta}_{-i}^t\right\|_2^2 + L_i^2\left\|\boldsymbol{\theta}^t - \boldsymbol{\theta}^{\mathrm{pse}}\right\|_2^2 + L_i^2 \varepsilon_i^2 \max_{j \in [n]} p_{ij}\left\|\boldsymbol{\theta}^t - \boldsymbol{\theta}^{\mathrm{pse}}\right\|_2^2\right),
$$

where the last inequality is based on Assumptions 2.2 and 2.4. Further, since the constriant function $g_i(\cdot)$ is $G_g$ Lipschitz for all $i \in [n]$, we have that

$$
(c) \leq 6\gamma_t^4 \sum_{i=1}^n \mathbb{E}\left\|\nabla g_i\left(\boldsymbol{\theta}_i^t\right)^\top \boldsymbol{\lambda}_i^t\right\|_2^2 + 6\gamma_t^4 \sum_{i=1}^n \mathbb{E}\left\|\nabla g_i(\boldsymbol{\theta}_i^{\mathrm{pse}})^\top \boldsymbol{\lambda}^{\mathrm{pse}}\right\|_2^2
$$

$$
\leq 6\gamma_t^2 G_g^2 \sum_{i=1}^n \mathbb{E}\|\gamma_t \boldsymbol{\lambda}_i^t\|_2^2 + 6\gamma_t^4 n G_g^2\|\boldsymbol{\lambda}^{\mathrm{pse}}\|_2^2
$$

$$
\leq 6\gamma_t^2 n B^2 G_g^2 + 6\gamma_t^4 n G_g^2\|\boldsymbol{\lambda}^{\mathrm{pse}}\|_2^2,
$$

where the last inequality is based on Lemma E.1.

Plugging the results of $(a)$, $(b)$, and $(c)$ into (A19) gives

$$\gamma_t^2 \sum_{i=1}^n \mathbb{E} \left\| \phi_i^t \left( \boldsymbol{\xi}_i^t; \boldsymbol{\theta}_i^t, \widehat{\boldsymbol{\theta}}_i^t, \boldsymbol{\lambda}_i^t \right) - \Phi_{\boldsymbol{\theta}^{\mathrm{pse}}}^{i,t} (\boldsymbol{\theta}^{\mathrm{pse}}, \boldsymbol{\lambda}^{\mathrm{pse}}) \right\|_2^2$$

$$\leq 3\gamma_t^2 \sigma_0^2 + 3\gamma_t^2 \left( \sigma_1^2 + 3 \sum_{i=1}^n L_i^2 \left( 1 + \varepsilon_i^2 \max_{j \in [n]} p_{ij} \right) \right) \mathbb{E} \left\| \boldsymbol{\theta}^t - \boldsymbol{\theta}^{\mathrm{pse}} \right\|_2^2$$

$$+ 9\gamma_t^2 \sum_{i=1}^n L_i^2 \mathbb{E} \left\| \widehat{\boldsymbol{\theta}}_i^t - \boldsymbol{\theta}_{-i}^t \right\|_2^2 + 6\gamma_t^2 n B^2 G_g^2 + 6\gamma_t^4 n G_g^2 \| \boldsymbol{\lambda}^{\mathrm{pse}} \|_2^2. \tag{A20}$$

Next, we deal with the last term on the right side of (A18). First, we have the following inequality:

$$\mathbb{E} \left[ \phi_i^t \left( \boldsymbol{\xi}_i^t; \boldsymbol{\theta}_i^t, \widehat{\boldsymbol{\theta}}_i^t, \boldsymbol{\lambda}_i^t \right) - \Phi_{\boldsymbol{\theta}^{\mathrm{pse}}}^{i,t} (\boldsymbol{\theta}^{\mathrm{pse}}, \boldsymbol{\lambda}^{\mathrm{pse}}) \right]$$

$$= \mathbb{E} \left[ G_{\boldsymbol{\theta}^t}^{(i)} \left( \boldsymbol{\theta}_i^t, \widehat{\boldsymbol{\theta}}_i^t \right) - G_{\boldsymbol{\theta}^{\mathrm{pse}}}^{(i)} (\boldsymbol{\theta}^{\mathrm{pse}}) \right] + \gamma_t \mathbb{E} \left[ \nabla \boldsymbol{g}_i \left( \boldsymbol{\theta}_i^t \right)^\top \boldsymbol{\lambda}_i^t - \nabla \boldsymbol{g}_i (\boldsymbol{\theta}_i^{\mathrm{pse}})^\top \boldsymbol{\lambda}^{\mathrm{pse}} \right].$$

Moreover, we have

$$-2\gamma_t \sum_{i=1}^n \mathbb{E} \left\langle \boldsymbol{\theta}_i^t - \boldsymbol{\theta}^{\mathrm{pse}}, G_{\boldsymbol{\theta}^t}^{(i)} \left( \boldsymbol{\theta}_i^t, \widehat{\boldsymbol{\theta}}_i^t \right) - G_{\boldsymbol{\theta}^{\mathrm{pse}}}^{(i)} (\boldsymbol{\theta}^{\mathrm{pse}}) \right\rangle$$

$$= -2\gamma_t \sum_{i=1}^n \mathbb{E} \left\langle \boldsymbol{\theta}_i^t - \boldsymbol{\theta}^{\mathrm{pse}}, G_{\boldsymbol{\theta}^t}^{(i)} \left( \boldsymbol{\theta}_i^t, \widehat{\boldsymbol{\theta}}_i^t \right) - G_{\boldsymbol{\theta}^t}^{(i)} (\boldsymbol{\theta}^t) \right\rangle - 2\gamma_t \mathbb{E} \left\langle \boldsymbol{\theta}^t - \boldsymbol{\theta}^{\mathrm{pse}}, G_{\boldsymbol{\theta}^t} (\boldsymbol{\theta}^t) - G_{\boldsymbol{\theta}^t} (\boldsymbol{\theta}^{\mathrm{pse}}) \right\rangle$$

$$- 2\gamma_t \mathbb{E} \left\langle \boldsymbol{\theta}^t - \boldsymbol{\theta}^{\mathrm{pse}}, G_{\boldsymbol{\theta}^t} (\boldsymbol{\theta}^{\mathrm{pse}}) - G_{\boldsymbol{\theta}^{\mathrm{pse}}} (\boldsymbol{\theta}^{\mathrm{pse}}) \right\rangle$$

$$\leq 4C\gamma_t \sum_{i=1}^n L_i \mathbb{E} \left\| \widehat{\boldsymbol{\theta}}_i^t - \boldsymbol{\theta}_{-i}^t \right\|_2 - 2\mu\gamma_t \mathbb{E} \left\| \boldsymbol{\theta}^t - \boldsymbol{\theta}^{\mathrm{pse}} \right\|_2^2 + 2\gamma_t \sum_{i=1}^n L_i \varepsilon_i \max_{j \in [n]} \sqrt{p_{ij}} \mathbb{E} \left\| \boldsymbol{\theta}^t - \boldsymbol{\theta}^{\mathrm{pse}} \right\|_2^2, \tag{A21}$$

where the last inequality is from Assumptions 2.2, 2.3, 2.4 and the Cauchy-Schwarz inequality.
Further, we have

$$-2\gamma_t^2 \sum_{i=1}^n \mathbb{E} \left\langle \boldsymbol{\theta}_i^t - \boldsymbol{\theta}_i^{\mathrm{pse}}, \nabla \boldsymbol{g}_i \left( \boldsymbol{\theta}_i^t \right)^\top \boldsymbol{\lambda}_i^t - \nabla \boldsymbol{g}_i (\boldsymbol{\theta}_i^{\mathrm{pse}})^\top \boldsymbol{\lambda}^{\mathrm{pse}} \right\rangle$$

$$\leq 2\gamma_t^2 \sum_{i=1}^n \mathbb{E} \left\langle \boldsymbol{\theta}_i^{\mathrm{pse}} - \boldsymbol{\theta}_i^t, \nabla \boldsymbol{g}_i \left( \boldsymbol{\theta}_i^t \right)^\top \boldsymbol{\lambda}_i^t \right\rangle + 4\gamma_t^2 C G_g \| \boldsymbol{\lambda}^{\mathrm{pse}} \|_2$$

$$\leq 2\gamma_t^2 \sum_{i=1}^n \mathbb{E} \left[ \left( \boldsymbol{g}_i (\boldsymbol{\theta}_i^{\mathrm{pse}}) - \boldsymbol{g}_i \left( \boldsymbol{\theta}_i^t \right) \right)^\top \boldsymbol{\lambda}_i^t \right] + 4\gamma_t^2 C G_g \| \boldsymbol{\lambda}^{\mathrm{pse}} \|_2$$

$$\leq 2\gamma_t^2 \mathbb{E} \left[ \sum_{i=1}^n \boldsymbol{g}_i (\boldsymbol{\theta}_i^{\mathrm{pse}})^\top \left( \boldsymbol{\lambda}_i^t - \overline{\boldsymbol{\lambda}}^t \right) + \boldsymbol{g} (\boldsymbol{\theta}^{\mathrm{pse}})^\top \overline{\boldsymbol{\lambda}}^t - \sum_{i=1}^n \boldsymbol{g}_i \left( \boldsymbol{\theta}_i^t \right)^\top \boldsymbol{\lambda}_i^t \right] + 4\gamma_t^2 C G_g \| \boldsymbol{\lambda}^{\mathrm{pse}} \|_2$$

$$\leq 2\gamma_t^2 \sum_{i=1}^n \mathbb{E} \left[ \| \boldsymbol{g}_i (\boldsymbol{\theta}_i^{\mathrm{pse}}) \|_2 \left\| \boldsymbol{\lambda}_i^t - \overline{\boldsymbol{\lambda}}^t \right\|_2 - \boldsymbol{g}_i \left( \boldsymbol{\theta}_i^t \right)^\top \boldsymbol{\lambda}_i^t \right] + 4\gamma_t^2 C G_g \| \boldsymbol{\lambda}^{\mathrm{pse}} \|_2, \tag{A22}$$

where the last inequality uses the fact that $\boldsymbol{g} (\boldsymbol{\theta}^{\mathrm{pse}})^\top \overline{\boldsymbol{\lambda}}^t \leq 0$.

Define $\widetilde{\mu} := \mu - \sum_{i=1}^n L_i \varepsilon_i \max_{j \in [n]} \sqrt{p_{ij}}$, $\nu := 3 \left( \sigma_1^2 + 3 \sum_{i=1}^n L_i^2 \left( 1 + \varepsilon_i^2 \max_{j \in [n]} p_{ij} \right) \right)$, and $\pi := 3\sigma_0^2 + 6n B^2 G_g^2 + 6n G_g^2 \| \boldsymbol{\lambda}^{\mathrm{pse}} \|_2^2 + 4C G_g \| \boldsymbol{\lambda}^{\mathrm{pse}} \|$. Plugging the results in (A20), (A21), and

(A22) into (A18) yields

$$\mathbb{E}\left\|\boldsymbol{\theta}^{t+1} - \boldsymbol{\theta}^{\mathrm{pse}}\right\|_2^2$$

$$\leq \left(1 - 2\gamma_t \widetilde{\mu} + \nu\gamma_t^2\right) \mathbb{E}\left\|\boldsymbol{\theta}^t - \boldsymbol{\theta}^{\mathrm{pse}}\right\|_2^2 + 4C\gamma_t \sum_{i=1}^n L_i \mathbb{E}\left\|\widehat{\boldsymbol{\theta}}_i^t - \boldsymbol{\theta}_{-i}^t\right\|_2 + 9\gamma_t^2 \sum_{i=1}^n L_i^2 \mathbb{E}\left\|\widehat{\boldsymbol{\theta}}_i^t - \boldsymbol{\theta}_{-i}^t\right\|_2^2$$

$$+ 2\gamma_t^2 \sum_{i=1}^n \mathbb{E}\left[B\left\|\boldsymbol{\lambda}_i^t - \overline{\boldsymbol{\lambda}}^t\right\|_2 - \boldsymbol{g}_i\left(\boldsymbol{\theta}_i^t\right)^\top \boldsymbol{\lambda}_i^t\right] + \pi\gamma_t^2. \tag{A23}$$

Let $\sup_{t\geq 1} \gamma_t \leq \frac{\widetilde{\mu}}{\nu}$, then $1 - 2\widetilde{\mu}\gamma_t + \nu\gamma_t^2 \leq 1 - \widetilde{\mu}\gamma_t$. Thus, we have

$$\mathbb{E}\left\|\boldsymbol{\theta}^t - \boldsymbol{\theta}^{\mathrm{pse}}\right\|_2^2$$

$$\leq \frac{1}{\widetilde{\mu}\gamma_t}\left(\mathbb{E}\left\|\boldsymbol{\theta}^t - \boldsymbol{\theta}^{\mathrm{pse}}\right\|_2^2 - \mathbb{E}\left\|\boldsymbol{\theta}^{t+1} - \boldsymbol{\theta}^{\mathrm{pse}}\right\|_2^2\right) + \frac{4C}{\widetilde{\mu}} \sum_{i=1}^n L_i \mathbb{E}\left\|\widehat{\boldsymbol{\theta}}_i^t - \boldsymbol{\theta}_{-i}^t\right\|_2$$

$$+ \frac{9\gamma_t}{\widetilde{\mu}} \sum_{i=1}^n L_i^2 \mathbb{E}\left\|\widehat{\boldsymbol{\theta}}_i^t - \boldsymbol{\theta}_{-i}^t\right\|_2^2 + \frac{2\gamma_t}{\widetilde{\mu}} \sum_{i=1}^n \mathbb{E}\left[B\left\|\boldsymbol{\lambda}_i^t - \overline{\boldsymbol{\lambda}}^t\right\|_2 - \boldsymbol{g}_i\left(\boldsymbol{\theta}_i^t\right)^\top \boldsymbol{\lambda}_i^t\right] + \frac{\pi\gamma_t}{\widetilde{\mu}}.$$

Summing the above inequality over $t \in [T]$ and plugging into the result of Lemma E.2 yields

$$\sum_{t=1}^T \mathbb{E}\left\|\boldsymbol{\theta}^t - \boldsymbol{\theta}^{\mathrm{pse}}\right\|_2^2 \leq \sum_{t=1}^T \frac{1}{\widetilde{\mu}\gamma_t}\left(\mathbb{E}\left\|\boldsymbol{\theta}^t - \boldsymbol{\theta}^{\mathrm{pse}}\right\|_2^2 - \mathbb{E}\left\|\boldsymbol{\theta}^{t+1} - \boldsymbol{\theta}^{\mathrm{pse}}\right\|_2^2\right) + \frac{4C}{\widetilde{\mu}} \sum_{t=1}^T \sum_{i=1}^n L_i \mathbb{E}\left\|\widehat{\boldsymbol{\theta}}_i^t - \boldsymbol{\theta}_{-i}^t\right\|_2$$

$$+ \frac{9}{\widetilde{\mu}} \sum_{t=1}^T \gamma_t \sum_{i=1}^n L_i^2 \mathbb{E}\left\|\widehat{\boldsymbol{\theta}}_i^t - \boldsymbol{\theta}_{-i}^t\right\|_2^2 + \frac{2}{\widetilde{\mu}}\left(3 + 2\sqrt{n}\right) B \sum_{t=1}^T \gamma_t \sum_{i=1}^n \mathbb{E}\left\|\boldsymbol{\lambda}_i^t - \overline{\boldsymbol{\lambda}}^t\right\|_2$$

$$+ \frac{9}{\widetilde{\mu}} \sum_{t=1}^T \sum_{i=1}^n \left\|\boldsymbol{\lambda}_i^t - \overline{\boldsymbol{\lambda}}^t\right\|_2^2 + \frac{\pi}{\widetilde{\mu}} \sum_{t=1}^T \gamma_t + \frac{8nB^2}{\widetilde{\mu}} \sum_{t=1}^T \gamma_t^2$$

$$- \frac{2}{\widetilde{\mu}} \sum_{t=1}^T \sum_{i=1}^n \gamma_t \boldsymbol{\lambda}^\top \boldsymbol{g}_i(\boldsymbol{\theta}_i^t) + \frac{n}{\widetilde{\mu}}\left(1 + \sum_{t=1}^T \gamma_t^2\right)\|\boldsymbol{\lambda}\|_2^2. \tag{A24}$$

Since $\left\|\boldsymbol{\theta}^t - \boldsymbol{\theta}^{\mathrm{pse}}\right\|_2^2 \leq 4C^2$, we have that

$$\sum_{t=1}^T \frac{1}{\gamma_t}\left(\mathbb{E}\left\|\boldsymbol{\theta}^t - \boldsymbol{\theta}^{\mathrm{pse}}\right\|_2^2 - \mathbb{E}\left\|\boldsymbol{\theta}^{t+1} - \boldsymbol{\theta}^{\mathrm{pse}}\right\|_2^2\right)$$

$$= \frac{1}{\gamma_1}\mathbb{E}\left\|\boldsymbol{\theta}^1 - \boldsymbol{\theta}^{\mathrm{pse}}\right\|_2^2 - \frac{1}{\gamma_T}\mathbb{E}\left\|\boldsymbol{\theta}^{T+1} - \boldsymbol{\theta}^{\mathrm{pse}}\right\|_2^2 + \sum_{t=2}^T \left(\frac{1}{\gamma_t} - \frac{1}{\gamma_{t-1}}\right)\mathbb{E}\left\|\boldsymbol{\theta}^t - \boldsymbol{\theta}^{\mathrm{pse}}\right\|_2^2$$

$$\leq \frac{1}{\gamma_1}4C^2 + \sum_{t=2}^T \left(\frac{1}{\gamma_t} - \frac{1}{\gamma_{t-1}}\right)4C^2 \leq \frac{4C^2}{\gamma_T}, \tag{A25}$$

where in the last inequality is based on the fact that $\frac{1}{\gamma_t} - \frac{1}{\gamma_{t-1}} \geq 0$ because $\gamma_t$ is a non-increasing sequence. Further, we have the following relations:

$$\sum_{i=1}^n L_i^2 \mathbb{E}\left\|\widehat{\boldsymbol{\theta}}_i^t - \boldsymbol{\theta}_{-i}^t\right\|_2^2 \leq \max_i L_i \sum_{i=1}^n \sum_{h\neq i} \left\|\widehat{\boldsymbol{\theta}}_{ih}^t - \boldsymbol{\theta}_h^t\right\|_2^2 = \max_i L_i \sum_{h=1}^n \|\boldsymbol{e}_h^t\|_2^2, \tag{A26}$$

$$\sum_{i=1}^n L_i \mathbb{E}\left\|\widehat{\boldsymbol{\theta}}_i^t - \boldsymbol{\theta}_{-i}^t\right\|_2 \leq \max_i L_i \sum_{i=1}^n \sqrt{\sum_{h\neq i}\left\|\widehat{\boldsymbol{\theta}}_{ih}^t - \boldsymbol{\theta}_h^t\right\|_2^2}$$

$$\leq \max_i L_i \sqrt{n \sum_{i=1}^n \sum_{h\neq i}\left\|\widehat{\boldsymbol{\theta}}_{ih}^t - \boldsymbol{\theta}_h^t\right\|_2^2}$$

$$\leq \max_i L_i \sqrt{n} \sum_{h=1}^n \|\boldsymbol{e}_h^t\|_2, \tag{A27}$$

where the last inequality is based on the fact that $\sqrt{a+b+c} \le \sqrt{a}+\sqrt{b}+\sqrt{c}$ for any $a, b, c \ge 0$. Plugging (A25), (A26) and (A27) into (A24) and utilizing the results in Lemmas E.4 and E.5, we have that

$$\sum_{t=1}^{T} \mathbb{E} \left\| \boldsymbol{\theta}^t - \boldsymbol{\theta}^{\text{pse}} \right\|_2^2 + \frac{2}{\widetilde{\mu}} \sum_{t=1}^{T} \sum_{i=1}^{n} \gamma_t \boldsymbol{\lambda}^\top \boldsymbol{g}_i \left( \boldsymbol{\theta}_i^t \right) - \frac{n}{\widetilde{\mu}} \left( 1 + \sum_{t=1}^{T} \gamma_t^2 \right) \| \boldsymbol{\lambda} \|_2^2$$
$$\le \mathcal{O} \left( \frac{1}{\widetilde{\mu} \gamma_T} + \frac{1}{\widetilde{\mu}} \sum_{t=1}^{T} \gamma_t \right). \tag{A28}$$

Since any $\boldsymbol{\lambda} \in \mathbb{R}_+^m$ satisfies the above inequality, by setting $\boldsymbol{\lambda} = \frac{\left[ \sum_{t=1}^{T} \gamma_t \sum_{i=1}^{n} \boldsymbol{g}_i \left( \boldsymbol{\theta}_i^t \right) \right]_+}{n \left( 1 + \sum_{t=1}^{T} \gamma_t^2 \right)}$, we have that

$$\frac{2}{\widetilde{\mu}} \boldsymbol{\lambda}^\top \left( \sum_{t=1}^{T} \gamma_t \sum_{i=1}^{n} \boldsymbol{g}_i \left( \boldsymbol{\theta}_i^t \right) \right) - \frac{n}{\widetilde{\mu}} \left( 1 + \sum_{t=1}^{T} \gamma_t^2 \right) \| \boldsymbol{\lambda} \|_2^2 = \frac{\left\| \left[ \sum_{t=1}^{T} \gamma_t \sum_{i=1}^{n} \boldsymbol{g}_i \left( \boldsymbol{\theta}_i^t \right) \right]_+ \right\|_2^2}{\widetilde{\mu} n \left( 1 + \sum_{t=1}^{T} \gamma_t^2 \right)}. \tag{A29}$$

As the terms in (A29) is non-negative, omitting it in (A28) gives

$$\sum_{t=1}^{T} \mathbb{E} \left\| \boldsymbol{\theta}^t - \boldsymbol{\theta}^{\text{pse}} \right\|_2^2 \le \mathcal{O} \left( \frac{1}{\widetilde{\mu} \gamma_T} + \frac{1}{\widetilde{\mu}} \sum_{t=1}^{T} \gamma_t \right).$$

Furthermore, since $\mathbb{E}_{\boldsymbol{\xi}_i \sim \mathcal{D}(\boldsymbol{\theta}^{\text{pse}})} | J_i(\boldsymbol{\xi}_i; \boldsymbol{\theta}_i^t, \boldsymbol{\theta}_{-i}^{\text{pse}}) - J_i(\boldsymbol{\xi}_i; \boldsymbol{\theta}^{\text{pse}}) | \le G_i \left\| \boldsymbol{\theta}_i^t - \boldsymbol{\theta}_i^{\text{pse}} \right\|_2$, for any $i \in [n]$, we have that

$$\mathcal{R}_i(T) = \sum_{t=1}^{T} \left( \mathbb{E}_{\boldsymbol{\xi}_i \sim \mathcal{D}(\boldsymbol{\theta}^{\text{pse}})} \left[ J \left( \boldsymbol{\xi}_i; \boldsymbol{\theta}_i^t, \boldsymbol{\theta}_{-i}^{\text{pse}} \right) - J \left( \boldsymbol{\xi}_i; \boldsymbol{\theta}^{\text{pse}} \right) \right] \right)$$
$$\le G_i \sum_{t=1}^{T} \left\| \boldsymbol{\theta}_i^t - \boldsymbol{\theta}_i^{\text{pse}} \right\|_2$$
$$\le G_i \sqrt{T \sum_{t=1}^{T} \left\| \boldsymbol{\theta}_i^t - \boldsymbol{\theta}_i^{\text{pse}} \right\|_2^2}$$
$$\le \mathcal{O} \left( \sqrt{\frac{T}{\widetilde{\mu}} \left( \frac{1}{\gamma_T} + \sum_{t=1}^{T} \gamma_t \right)} \right), \forall i \in [n].$$

On the other hand, plugging (A29) into (A28) and omitting the non-negtive term $\sum_{t=1}^{T} \mathbb{E} \left\| \boldsymbol{\theta}^t - \boldsymbol{\theta}^{\text{pse}} \right\|_2^2$, we have

$$\frac{\left\| \left[ \sum_{t=1}^{T} \gamma_t \sum_{i=1}^{n} \boldsymbol{g}_i \left( \boldsymbol{\theta}_i^t \right) \right]_+ \right\|_2^2}{\widetilde{\mu} n \left( 1 + \sum_{t=1}^{T} \gamma_t^2 \right)} \le \mathcal{O} \left( \frac{1}{\widetilde{\mu} \gamma_T} + \frac{1}{\widetilde{\mu}} \sum_{t=1}^{T} \gamma_t \right).$$

$$\left\| \left[ \sum_{t=1}^{T} \gamma_t \sum_{i=1}^{n} \boldsymbol{g}_i \left( \boldsymbol{\theta}_i^t \right) \right]_+ \right\|_2 \le \mathcal{O} \left( \sqrt{\left( \frac{1}{\gamma_T} + \sum_{t=1}^{T} \gamma_t \right) \left( 1 + \sum_{t=1}^{T} \gamma_t^2 \right)} \right).$$

Then, we prove that

$$\mathcal{R}_g(T) \le \mathcal{O} \left( \frac{1}{\gamma_T} \sqrt{\left( \frac{1}{\gamma_T} + \sum_{t=1}^{T} \gamma_t \right) \left( 1 + \sum_{t=1}^{T} \gamma_t^2 \right)} \right).$$

## E.1 Proof of Lemma E.1

From the update rule of the dual variables, for any $\boldsymbol{\lambda}_i^t \in \mathbb{R}_+^m$, $i \in [n]$, and $t \in [T]$, we have that

$$\sum_{i=1}^n \left\| \boldsymbol{\lambda}_i^{t+1} \right\|_2^2 \leq \sum_{i=1}^n \left\| \sum_{j=1}^n a_{ij} \left[ \left(1 - \gamma_t^2\right) \boldsymbol{\lambda}_j^t + \gamma_t \boldsymbol{g}_i(\boldsymbol{\theta}_i^t) \right] \right\|_2^2$$

$$\leq \sum_{i=1}^n \sum_{j=1}^n a_{ij} \left\| (1 - \gamma_t^2) \boldsymbol{\lambda}_j^t + \gamma_t^2 \frac{\boldsymbol{g}_i(\boldsymbol{\theta}_i^t)}{\gamma_t} \right\|_2^2$$

$$\leq \sum_{i=1}^n \sum_{j=1}^n a_{ij} \left[ (1 - \gamma_t^2) \left\| \boldsymbol{\lambda}_j^t \right\|_2^2 + \left\| \boldsymbol{g}_i(\boldsymbol{\theta}_i^t) \right\|_2^2 \right]$$

$$\leq (1 - \gamma_t^2) \sum_{i=1}^n \left\| \boldsymbol{\lambda}_i^t \right\|_2^2 + \sum_{i=1}^n \left\| \boldsymbol{g}_i(\boldsymbol{\theta}_i^t) \right\|_2^2$$

$$\leq (1 - \gamma_t^2) \sum_{i=1}^n \left\| \boldsymbol{\lambda}_i^t \right\|_2^2 + nB^2.$$

We next bound $\sum_{i=1}^n \left\| \boldsymbol{\lambda}_i^t \right\|_2^2$, $\forall t \in [T]$ by deduction. First, since $\boldsymbol{\lambda}_i^1 = \boldsymbol{0}$, $\gamma_1 \leq 1$, and $\left\| \boldsymbol{g}_i(\boldsymbol{\theta}_i^1) \right\|_2^2 \leq B^2$, $\forall i \in [n]$, we have that $\sum_{i=1}^n \left\| \boldsymbol{\lambda}_i^2 \right\|_2^2 \leq \sum_{i=1}^n \left\| \boldsymbol{g}_i(\boldsymbol{\theta}_i^1) \right\|_2^2 \leq nB^2 \leq \frac{nB^2}{\gamma_1^2}$. Assume that $\sum_{i=1}^n \left\| \boldsymbol{\lambda}_i^t \right\|_2^2 \leq \frac{nB^2}{\gamma_{t-1}^2}$. Since $\{\gamma_t\}_{t \in [T]}$ is a non-incerasing sequence, $\sum_{i=1}^n \left\| \boldsymbol{\lambda}_i^t \right\|_2^2 \leq \frac{nB^2}{\gamma_{t-1}^2} \leq \frac{nB^2}{\gamma_t^2}$ and thus $\sum_{i=1}^n \left\| \boldsymbol{\lambda}_i^{t+1} \right\|_2^2 \leq (1 - \gamma_t^2) \frac{nB^2}{\gamma_t^2} + nB^2 = \frac{nB^2}{\gamma_t^2}$. Therefore, for any $t \in [T]$, we have $\sum_{i=1}^n \left\| \boldsymbol{\lambda}_i^{t+1} \right\|_2^2 \leq \frac{nB^2}{\gamma_t^2} \leq \frac{nB^2}{\gamma_{t+1}^2}$, i.e., $\sum_{i=1}^n \left\| \gamma_t \boldsymbol{\lambda}_i^t \right\|_2^2 \leq nB^2$, which completes the proof.

## E.2 Proof of Lemma E.2

From the update rule of the dual variables $\boldsymbol{\lambda}_i$, for any $\boldsymbol{\lambda} \in \mathbb{R}_+^m$, we have that

$$\sum_{i=1}^n \left\| \boldsymbol{\lambda}_i^{t+1} - \boldsymbol{\lambda} \right\|_2^2 = \sum_{i=1}^n \left\| \left[ (1 - \gamma_t^2) \sum_{j \in \mathcal{N}_i} a_{ij} \boldsymbol{\lambda}_j^t + \gamma_t \boldsymbol{g}_i\left(\boldsymbol{\theta}_i^t\right) \right]_+ - \boldsymbol{\lambda} \right\|_2^2$$

$$\leq \sum_{i=1}^n \left\| (1 - \gamma_t^2) \sum_{j \in \mathcal{N}_i} a_{ij} \left( \boldsymbol{\lambda}_j^t - \boldsymbol{\lambda}_i^t \right) + \left( \boldsymbol{\lambda}_i^t - \boldsymbol{\lambda} \right) + \gamma_t \left( \boldsymbol{g}_i\left(\boldsymbol{\theta}_i^t\right) - \gamma_t \boldsymbol{\lambda}_i^t \right) \right\|_2^2$$

$$\leq \sum_{i=1}^n \left( \sum_{j \in \mathcal{N}_i} a_{ij} \left\| \boldsymbol{\lambda}_j^t - \boldsymbol{\lambda}_i^t \right\|_2^2 + \left\| \boldsymbol{\lambda}_i^t - \boldsymbol{\lambda} \right\|_2^2 + \gamma_t^2 \left\| \boldsymbol{g}_i\left(\boldsymbol{\theta}_i^t\right) - \gamma_t \boldsymbol{\lambda}_i^t \right\|_2^2 \right.$$

$$+ 2 \sum_{j \in \mathcal{N}_i} a_{ij} \left\langle \boldsymbol{\lambda}_j^t - \boldsymbol{\lambda}_i^t, \boldsymbol{\lambda}_i^t - \boldsymbol{\lambda} \right\rangle + 2\gamma_t \left\langle \boldsymbol{\lambda}_i^t - \boldsymbol{\lambda}, \boldsymbol{g}_i\left(\boldsymbol{\theta}_i^t\right) - \gamma_t \boldsymbol{\lambda}_i^t \right\rangle$$

$$\left. + 2\gamma_t \sum_{j \in \mathcal{N}_i} a_{ij} \left\| \boldsymbol{\lambda}_j^t - \boldsymbol{\lambda}_i^t \right\|_2 \left\| \boldsymbol{g}_i\left(\boldsymbol{\theta}_i^t\right) - \gamma_t \boldsymbol{\lambda}_i^t \right\|_2 \right), \tag{A30}$$

where we use the fact $1 - \gamma_t^2 \leq 1$ in (A30). Next, we simplify the terms in (A30). First, based on the inequality $(a - b)^2 \leq 2 \left( a^2 + b^2 \right)$ for any $a, b \geq 0$, we have that

$$\sum_{i=1}^n \sum_{j=1}^n a_{ij} \left\| \boldsymbol{\lambda}_j^t - \boldsymbol{\lambda}_i^t \right\|_2^2 = \sum_{i=1}^n \sum_{j=1}^n a_{ij} \left( \left\| \left( \boldsymbol{\lambda}_j^t - \overline{\boldsymbol{\lambda}}^t \right) - \left( \boldsymbol{\lambda}_i^t - \overline{\boldsymbol{\lambda}}^t \right) \right\|_2^2 \right) \leq 4 \sum_{i=1}^n \left\| \boldsymbol{\lambda}_i^t - \overline{\boldsymbol{\lambda}}^t \right\|_2^2.$$

In addition, with the result in Lemma E.1, we know that

$$\sum_{i=1}^{n} \left\| \boldsymbol{g}_i \left( \boldsymbol{\theta}_i^t \right) - \gamma_t \boldsymbol{\lambda}_i^t \right\|_2^2 \leq 2 \sum_{i=1}^{n} \left\| \boldsymbol{g}_i \left( \boldsymbol{\theta}_i^t \right) \right\|_2^2 + 2 \sum_{i=1}^{n} \left\| \gamma_t \boldsymbol{\lambda}_i^t \right\|_2^2 \leq 2nB^2 + 2nB^2 = 4nB^2,$$

$$\left\| \boldsymbol{g}_i \left( \boldsymbol{\theta}_i^t \right) - \gamma_t \boldsymbol{\lambda}_i^t \right\|_2 \leq \left\| \boldsymbol{g}_i \left( \boldsymbol{\theta}_i^t \right) \right\|_2 + \left\| \gamma_t \boldsymbol{\lambda}_i^t \right\|_2 \leq B + \sqrt{n}B = (1 + \sqrt{n})B.$$

Moreover, based on the fact that $\sum_{i=1}^{n} \sum_{j=1}^{n} a_{ij} \left\langle \boldsymbol{\lambda}_j^t - \boldsymbol{\lambda}_i^t, \boldsymbol{z} \right\rangle = 0$ for any $\boldsymbol{z} \in \mathbb{R}^m$, we have that

$$\sum_{i=1}^{n} \sum_{j=1}^{n} a_{ij} \left\langle \boldsymbol{\lambda}_j^t - \boldsymbol{\lambda}_i^t, \boldsymbol{\lambda}_i^t - \boldsymbol{\lambda} \right\rangle = \sum_{i=1}^{n} \sum_{j=1}^{n} a_{ij} \left\langle \boldsymbol{\lambda}_j^t - \boldsymbol{\lambda}_i^t, \boldsymbol{\lambda}_i^t - \overline{\boldsymbol{\lambda}}^t \right\rangle$$

$$\leq \frac{1}{2} \sum_{i=1}^{n} \sum_{j=1}^{n} a_{ij} \left( \left\| \boldsymbol{\lambda}_j^t - \boldsymbol{\lambda}_i^t \right\|_2^2 + \left\| \boldsymbol{\lambda}_i^t - \overline{\boldsymbol{\lambda}}^t \right\|_2^2 \right)$$

$$\leq \frac{5}{2} \sum_{i=1}^{n} \left\| \boldsymbol{\lambda}_i^t - \overline{\boldsymbol{\lambda}}^t \right\|_2^2.$$

Furthermore, notice that

$$\left\langle \boldsymbol{\lambda}_i^t - \boldsymbol{\lambda}, \boldsymbol{g}_i \left( \boldsymbol{\theta}_i^t \right) - \gamma_t \boldsymbol{\lambda}_i^t \right\rangle = \left\langle \boldsymbol{\lambda}_i^t - \boldsymbol{\lambda}, \boldsymbol{g}_i \left( \boldsymbol{\theta}_i^t \right) \right\rangle - \gamma_t \| \boldsymbol{\lambda}_i^t \|_2^2 + \gamma_t \boldsymbol{\lambda}^\top \boldsymbol{\lambda}_i^t$$

$$\leq \left\langle \boldsymbol{\lambda}_i^t - \boldsymbol{\lambda}, \boldsymbol{g}_i \left( \boldsymbol{\theta}_i^t \right) \right\rangle + \frac{\gamma_t}{2} \left( \| \boldsymbol{\lambda} \|_2^2 - \| \boldsymbol{\lambda}_i^t \|_2^2 \right),$$

where the last inequality follows that $\boldsymbol{\lambda}^\top \boldsymbol{\lambda}_i^t = \frac{1}{2} (\| \boldsymbol{\lambda} \|_2^2 + \| \boldsymbol{\lambda}_i^t \|_2^2)$. We also have

$$\sum_{i=1}^{n} \sum_{j=1}^{n} a_{ij} \left\| \boldsymbol{\lambda}_j^t - \boldsymbol{\lambda}_i^t \right\|_2 \leq 2 \sum_{i=1}^{n} \left\| \boldsymbol{\lambda}_i^t - \overline{\boldsymbol{\lambda}}^t \right\|_2.$$

Plugging all the above results into (A30), we obtain

$$\sum_{i=1}^{n} \left\| \boldsymbol{\lambda}_i^{t+1} - \boldsymbol{\lambda} \right\|_2^2 \leq \sum_{i=1}^{n} \left( \left\| \boldsymbol{\lambda}_i^t - \boldsymbol{\lambda} \right\|_2^2 + 9 \left\| \boldsymbol{\lambda}_i^t - \overline{\boldsymbol{\lambda}}^t \right\|_2^2 \right.$$

$$+ 2\gamma_t \left\langle \boldsymbol{\lambda}_i^t - \boldsymbol{\lambda}, \boldsymbol{g}_i \left( \boldsymbol{\theta}_i^t \right) \right\rangle + \gamma_t^2 \left( \| \boldsymbol{\lambda} \|_2^2 - \| \boldsymbol{\lambda}_i^t \|_2^2 \right)$$

$$\left. + 4\gamma_t (1 + \sqrt{n})B \left\| \boldsymbol{\lambda}_i^t - \overline{\boldsymbol{\lambda}}^t \right\|_2 \right) + 4nB^2 \gamma_t^2.$$

Rearranging the terms in the above inequality and summing over $t \in [T]$ gives

$$\sum_{t=1}^{T} \sum_{i=1}^{n} \gamma_t \left\langle \boldsymbol{\lambda}_i^t - \boldsymbol{\lambda}, \boldsymbol{g}_i \left( \boldsymbol{\theta}_i^t \right) \right\rangle + \sum_{t=1}^{T} \frac{n\gamma_t^2}{2} \| \boldsymbol{\lambda} \|_2^2$$

$$\geq \frac{1}{2} \sum_{t=1}^{T} \sum_{i=1}^{n} \left( \left\| \boldsymbol{\lambda}_i^{t+1} - \boldsymbol{\lambda} \right\|_2^2 - \left\| \boldsymbol{\lambda}_i^t - \boldsymbol{\lambda} \right\|_2^2 \right)$$

$$- \frac{9}{2} \sum_{t=1}^{T} \sum_{i=1}^{n} \left\| \boldsymbol{\lambda}_i^t - \overline{\boldsymbol{\lambda}}^t \right\|_2^2 - 4nB^2 \sum_{t=1}^{T} \gamma_t^2$$

$$- 2(1 + \sqrt{n})B \sum_{t=1}^{T} \gamma_t \sum_{i=1}^{n} \left\| \boldsymbol{\lambda}_i^t - \overline{\boldsymbol{\lambda}}^t \right\|_2 + \sum_{t=1}^{T} \sum_{i=1}^{n} \frac{\gamma_t^2}{2} \| \boldsymbol{\lambda}_i^t \|_2^2.$$

The last term on the right side of the above inequality is non-negative and can be omitted. Besides, since $\boldsymbol{\lambda}_i^1 = \boldsymbol{0}$ for all $i \in [n]$, then $\sum_{t=1}^{T} \left( \left\| \boldsymbol{\lambda}_i^{t+1} - \boldsymbol{\lambda} \right\|_2^2 - \left\| \boldsymbol{\lambda}_i^t - \boldsymbol{\lambda} \right\|_2^2 \right) \geq -\| \boldsymbol{\lambda} \|_2^2$ for any $\boldsymbol{\lambda}_i^{T+1} \in$

$\mathbb{R}^m_+$. Thus, we have

$$\sum_{t=1}^{T}\sum_{i=1}^{n}\gamma_t\left((\boldsymbol{\lambda}_i^t)^\top \boldsymbol{g}_i(\boldsymbol{\theta}_i^t) - \boldsymbol{\lambda}^\top \boldsymbol{g}_i(\boldsymbol{\theta}_i^t)\right)$$

$$\geq -\frac{n}{2}\left(1+\sum_{t=1}^{T}\gamma_t^2\right)\|\boldsymbol{\lambda}\|_2^2 - \frac{9}{2}\sum_{t=1}^{T}\sum_{i=1}^{n}\left\|\boldsymbol{\lambda}_i^t - \overline{\boldsymbol{\lambda}}^t\right\|_2^2$$

$$- 2(1+\sqrt{n})B\sum_{t=1}^{T}\gamma_t\sum_{i=1}^{n}\left\|\boldsymbol{\lambda}_i^t - \overline{\boldsymbol{\lambda}}^t\right\|_2 - 4nB^2\sum_{t=1}^{T}\gamma_t^2.$$

Rearranging the terms in the above inequality yields

$$-\sum_{t=1}^{T}\sum_{i=1}^{n}\gamma_t(\boldsymbol{\lambda}_i^t)^\top \boldsymbol{g}_i(\boldsymbol{\theta}_i^t) \leq -\sum_{t=1}^{T}\sum_{i=1}^{n}\gamma_t\boldsymbol{\lambda}^\top \boldsymbol{g}_i(\boldsymbol{\theta}_i^t) + \frac{n}{2}\left(1+\sum_{t=1}^{T}\gamma_t^2\right)\|\boldsymbol{\lambda}\|_2^2 + \frac{9}{2}\sum_{t=1}^{T}\sum_{i=1}^{n}\left\|\boldsymbol{\lambda}_i^t - \overline{\boldsymbol{\lambda}}^t\right\|_2^2$$

$$+ 2(1+\sqrt{n})B\sum_{t=1}^{T}\gamma_t\sum_{i=1}^{n}\left\|\boldsymbol{\lambda}_i^t - \overline{\boldsymbol{\lambda}}^t\right\|_2 + 4nB^2\sum_{t=1}^{T}\gamma_t^2,$$

which completes the proof.

### E.3 Proof of Lemma E.4

Based on the update rule of $\widehat{\boldsymbol{\theta}}_{ih}^t$ that $\widehat{\boldsymbol{\theta}}_{ih}^{t+1} = \sum_{k\neq h}a_{ik}\widehat{\boldsymbol{\theta}}_{kh}^t + a_{ih}\boldsymbol{\theta}_h^t, \forall h \neq i$ and $i, h \in [n]$, we have that

$$\boldsymbol{e}_{ih}^{t+1} := \widehat{\boldsymbol{\theta}}_{ih}^{t+1} - \boldsymbol{\theta}_h^{t+1} = \sum_{k\neq h}a_{ik}\widehat{\boldsymbol{\theta}}_{kh}^t + a_{ih}\boldsymbol{\theta}_h^t - \boldsymbol{\theta}_h^{t+1} + \boldsymbol{\theta}_h^t - \boldsymbol{\theta}_h^t$$

$$= \sum_{k\neq h}a_{ik}\boldsymbol{e}_{kh}^t - \left(\boldsymbol{\theta}_h^{t+1} - \boldsymbol{\theta}_h^t\right).$$

Recall that $\mathbf{A}_h^-$ is the weight matrix formed by removing the $h$th row and $h$th column of the weight matrix $\mathbf{A}$ for any $h \in [n]$, and $\boldsymbol{e}_h^t := \mathrm{col}\left(\boldsymbol{e}_{1h}^t, \cdots, \boldsymbol{e}_{(h-1)h}^t, \boldsymbol{e}_{(h+1)h}^t, \cdots, \boldsymbol{e}_{nh}^t\right)$. Then,

$$\boldsymbol{e}_h^{t+1} = (\mathbf{A}_h^- \otimes \mathbf{I}_d)\boldsymbol{e}_h^t + \mathbf{1}_{n-1} \otimes \left(\boldsymbol{\theta}_h^{t+1} - \boldsymbol{\theta}_h^t\right).$$

Since $\beta$ is the maximum eigenvalue of $\mathbf{A}_h^-$ for all $h \in [n]$, we have that

$$\mathbb{E}\|\boldsymbol{e}_h^{t+1}\|_2 \leq \mathbb{E}\left\|(\mathbf{A}_h^- \otimes \mathbf{I}_d)\boldsymbol{e}_h^t\right\|_2 + \mathbb{E}\left\|\mathbf{1}_{n-1} \otimes \left(\boldsymbol{\theta}_h^{t+1} - \boldsymbol{\theta}_h^t\right)\right\|_2$$

$$\leq \beta\mathbb{E}\|\boldsymbol{e}_h^t\|_2 + \sqrt{n-1}\gamma_t\mathbb{E}\left\|\phi_h^t\left(\boldsymbol{\xi}_h^t; \boldsymbol{\theta}_h^t, \widehat{\boldsymbol{\theta}}_h^t, \boldsymbol{\lambda}_h^t\right)\right\|_2$$

$$\leq \beta\mathbb{E}\|\boldsymbol{e}_h^t\|_2 + \sqrt{n-1}\gamma_t\mathbb{E}\left\|\nabla_{\boldsymbol{\theta}_h}J_h\left(\boldsymbol{\xi}_h^t; \boldsymbol{\theta}_h^t, \widehat{\boldsymbol{\theta}}_h^t\right)\right\|_2 + \sqrt{n-1}\gamma_t^2\mathbb{E}\left\|\nabla \boldsymbol{g}_h(\boldsymbol{\theta}_h)^\top \boldsymbol{\lambda}_h^t\right\|_2$$

$$\leq \beta\mathbb{E}\|\boldsymbol{e}_h^t\|_2 + \sqrt{n-1}\gamma_t(G + G_g\mathbb{E}\|\gamma_t\boldsymbol{\lambda}_h^t\|_2)$$

$$\leq \beta^t\mathbb{E}\|\boldsymbol{e}_h^1\|_2 + \sqrt{n-1}\sum_{k=0}^{t-1}\beta^k\gamma_{t-k}(G + \sqrt{n}BG_g). \tag{A31}$$

Further, since $\boldsymbol{\theta}_{ih}^1 = \mathbf{0}$ for any $i, h \in [n]$, then, from Assumption 2.3, $\mathbb{E}\|\boldsymbol{e}_{ih}^1\|_2 = \|\boldsymbol{\theta}_h^1\|_2 \leq C$. Summing the above inequality over $t \in [T]$ and $h \in [n]$, we obtain

$$\sum_{t=1}^{T}\sum_{h=1}^{n}\mathbb{E}\|\boldsymbol{e}_h^t\|_2 \leq nC\sum_{t=1}^{T}\beta^{t-1} + n\sqrt{n-1}(G + \sqrt{n}BG_g)\sum_{t=1}^{T}\sum_{k=0}^{t-2}\beta^k\gamma_{t-k-1}$$

$$\leq \frac{nC}{1-\beta} + n\sqrt{n-1}(G + \sqrt{n}BG_g)\sum_{k=1}^{T}\sum_{t=k+1}^{T}\beta^{t-k-1}\gamma_k$$

$$\leq \frac{nC}{1-\beta} + \frac{n\sqrt{n-1}(G + \sqrt{n}BG_g)}{1-\beta}\sum_{k=1}^{T}\gamma_k.$$

On the other hand, taking square on both sides of (A31), we have

$$\mathbb{E}\|e_h^t\|_2^2 \le 2\beta^t \mathbb{E}\left\|e_h^1\right\|_2^2 + 2(n-1)(G+\sqrt{n}BG_g)^2 \left(\sum_{k=0}^{t-2} \beta^k \gamma_{t-k-1}\right)^2. \tag{A32}$$

Using the Cauchy-Schwarz inequality yields

$$\left(\sum_{k=0}^{t-2} \beta^k \gamma_{t-k-1}\right)^2 \le \left(\sum_{k=0}^{t-2} \beta^k\right)\left(\sum_{k=0}^{t-2} \beta^k \gamma_{t-k-1}^2\right) \le \frac{\sum_{k=0}^{t-2} \beta^k \gamma_{t-k-1}}{1-\beta}. \tag{A33}$$

Plugging (A33) into (A32) and summing over $t \in [T]$, we have that

$$\sum_{t=1}^{T}\sum_{h=1}^{n} \mathbb{E}\|e_h^t\|_2^2 \le 2nC^2 \sum_{t=1}^{T} \beta^t + \frac{2n(n-1)(G+\sqrt{n}BG_g)^2}{1-\beta}\left(\sum_{t=1}^{T}\sum_{k=0}^{t-2} \beta^k \gamma_{t-k-1}\right)$$

$$\le \frac{2nC^2}{1-\beta} + \frac{2n(n-1)(G+\sqrt{n}BG_g)^2}{(1-\beta)^2}\sum_{k=1}^{T} \gamma_k.$$

## E.4 Proof of Lemma E.5

Let $\omega_i^t := \left[\left(1-\gamma_t^2\right)\sum_{j\in\mathcal{N}_i} a_{ij}\lambda_j^t + \gamma_t g_i\left(\theta_i^t\right)\right]_+ - \sum_{j\in\mathcal{N}_i} a_{ij}\lambda_j^t$. Then, for any $i \in [n]$, we have that

$$
\begin{aligned}
\left\|\omega_i^t\right\|_2 &= \left\|\left[\left(1-\gamma_t^2\right)\sum_{j\in\mathcal{N}_i} a_{ij}\lambda_j^t + \gamma_t g_i\left(\theta_i^t\right)\right]_+ - \sum_{j\in\mathcal{N}_i} a_{ij}\lambda_j^t\right\|_2 \\
&\le \left\|-\gamma_t \sum_{j\in\mathcal{N}_i} a_{ij}\gamma_t\lambda_j^t + \gamma_t g_i\left(\theta_i^t\right)\right\|_2 \\
&\le \gamma_t \sum_{j\in\mathcal{N}_i} a_{ij}\left\|\gamma_t\lambda_j^t\right\|_2 + \gamma_t \left\|g_i\left(\theta_i^t\right)\right\|_2 \\
&\le \gamma_t(\sqrt{n}+1)B. \tag{A34}
\end{aligned}
$$

The first inequality in (A34) results from the nonexpansive property of projection, and the third inequality holds by using Lemma E.1. By the update rule of $\lambda_i$ for any $i \in [n]$, we have that

$$\lambda_i^{t+1} = \sum_{j\in\mathcal{N}_i} a_{ij}\lambda_j^t + \omega_i^t.$$

Define concatenation vectors $\lambda_o^t = \mathrm{col}\left(\lambda_1^t, \cdots, \lambda_n^t\right)$ and $\omega_o^t = \mathrm{col}\left(\omega_1^t, \cdots, \omega_n^t\right)$. Then, for any $t \in [T]$, we have

$$\lambda_o^{t+1} = (\mathbf{A} \otimes \mathbf{I}_m)\lambda_o^t + \omega_o^t. \tag{A35}$$

Since $\overline{\lambda}^t = \frac{1}{n}\sum_{i=1}^{n} \lambda_i^t$, we have that

$$\Delta^t := \lambda_o^t - (\mathbf{1}_n \otimes \mathbf{I}_m)\overline{\lambda}^t = \left(\left(\mathbf{I}_n - \frac{\mathbf{1}_n\mathbf{1}_n^T}{n}\right) \otimes \mathbf{I}_m\right)\lambda_o^t, \forall t \in [T]. \tag{A36}$$

Combining (A35) and (A36) yields

$$\Delta^{t+1} = (\mathbf{A} \otimes \mathbf{I}_m)\Delta^t + \left(\left(\mathbf{I} - \frac{\mathbf{1}\mathbf{1}^T}{n}\right) \otimes \mathbf{I}_m\right)\omega_o^t, \forall t \in [T].$$

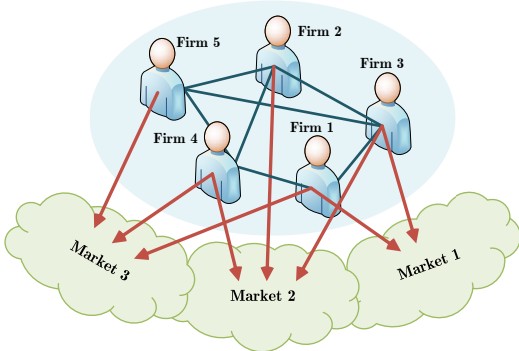

Figure 3: A networked Cournot game with five firms and three markets.

Since $\boldsymbol{\lambda}_i^1 = \mathbf{0}$ for all $i \in [n]$, then $\boldsymbol{\Delta}^1 = \mathbf{0}$. Based on the fact that $\left\| \mathbf{I} - \frac{\mathbf{11}^T}{n} \right\|_{\mathrm{F}} \leq 2$, we have that

$$\sum_{i=1}^{n} \left\| \boldsymbol{\lambda}_i^{t+1} - \overline{\boldsymbol{\lambda}}^{t+1} \right\|_2 = \left\| \boldsymbol{\Delta}^{t+1} \right\|_2 = \left\| (\mathbf{A} \otimes \mathbf{I}_m) \boldsymbol{\Delta}^t + \left( \left( \mathbf{I} - \frac{\mathbf{11}^T}{n} \right) \otimes \mathbf{I}_m \right) \boldsymbol{\omega}_o^t \right\|_2$$

$$\leq \left\| (\mathbf{A} \otimes \mathbf{I}_m) \boldsymbol{\Delta}^t \right\|_2 + \left\| \left( \left( \mathbf{I} - \frac{\mathbf{11}^T}{n} \right) \otimes \mathbf{I}_m \right) \boldsymbol{\omega}_o^t \right\|_2$$

$$\leq \sigma_2(\mathbf{A}) \left\| \boldsymbol{\Delta}^t \right\|_2 + 2 \left\| \boldsymbol{\omega}_o^t \right\|_2$$

$$\leq 2 \sum_{k=0}^{t-1} \sigma_2(\mathbf{A})^k \left\| \boldsymbol{\omega}_o^{t-k} \right\|_2$$

$$\leq 2(n + \sqrt{n}) B \sum_{k=0}^{t-1} \sigma_2(\mathbf{A})^k \gamma_{t-k},$$

where the last inequality is based on the result in (A34). Summing the above inequality over $t \in [T]$ yields

$$\sum_{t=1}^{T} \sum_{i=1}^{n} \left\| \boldsymbol{\lambda}_i^t - \overline{\boldsymbol{\lambda}}^t \right\|_2 \leq 2(n + \sqrt{n}) B \sum_{t=1}^{T} \sum_{k=0}^{t-2} \sigma_2(\mathbf{A})^k \gamma_{t-1-k}$$

$$\leq \frac{2(n + \sqrt{n}) B}{1 - \sigma_2(\mathbf{A})} \sum_{k=1}^{T} \gamma_k.$$

Similarly to the calculation of $\sum_{t=1}^{T} \sum_{h=1}^{n} \| e_h^t \|_2^2$ in Section E.3, we have that

$$\sum_{t=1}^{T} \sum_{i=1}^{n} \left\| \boldsymbol{\lambda}_i^t - \overline{\boldsymbol{\lambda}}^t \right\|_2^2 \leq \frac{4(n + \sqrt{n})^2 B^2}{(1 - \sigma_2(\mathbf{A}))^2} \sum_{k=1}^{T} \gamma_k.$$

## F  Simulation Details

### F.1  Networked Cournot Game

The Cournot game is a foundational model in economic theory (Allaz and Vila, 1993) for analyzing oligopolistic competition, where a limited number of firms dominate a specific market. In Cournot games, all firms sell a homogeneous commodity and aim to maximize their individual profits by independently and simultaneously determining optimal production quantities. The total quantity produced by all firms is constrained by factors such as market capacity, raw material availability, and environmental considerations. The profit of each firm depends not only on its own production quantity but also on the quantities chosen by its competitors, as they influence the demand price determined

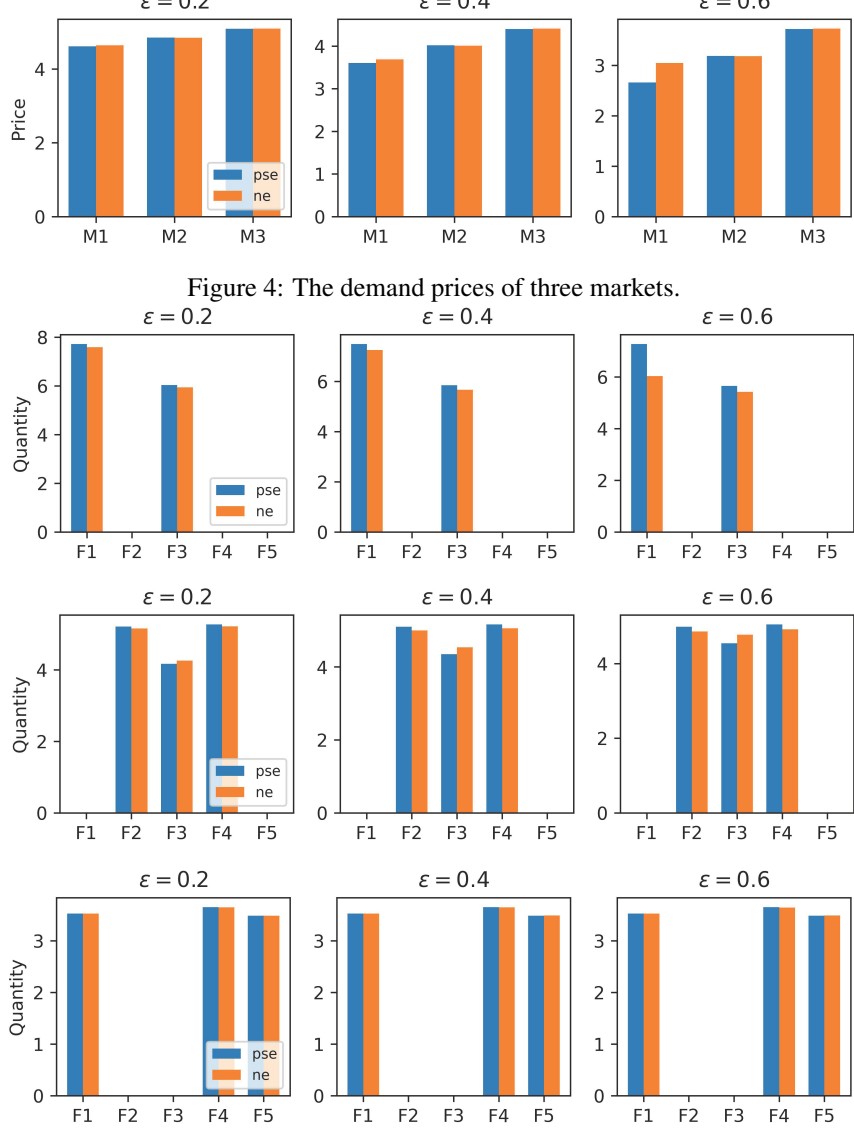

Figure 4: The demand prices of three markets.

Figure 5: The serving quantities of five firms to three markets.

by the market's demand curve and the total production quantity. There are strategic interactions between firms and markets in the Cournot game. According to the law of supply and demand, an increased production quantity drives down the demand price, and vice versa. The Cournot game model has diverse applications in various fields, including supply chain management, electricity market competition, natural resource extraction, online advertising auctions, and the telecommunications industry.

In this experiment, we consider a networked Cournot game comprising $n$ firms selling a single commodity across $m$ markets, as illustrated in Fig. 3. Each firm $i \in [n]$ determines its output quantity $\boldsymbol{\theta}_i = \mathrm{col}\left(\theta_{i1}, \cdots, \theta_{im}\right)$ subject to the constraint of its production capacity $Q_i$ that $\sum_{j=1}^{m} \theta_{ij} \leq Q_i$. Here, $\theta_{ij}$ denotes the quantity of player $i$ sold to the $j$th market. The total quantity allocated to market $j$ is limited by its market capacity $B_j$, satisfying the condition that $\sum_{i=1}^{n} \theta_{ij} \leq B_j \ \forall j \in [m]$. Thus, the local constraint of player $i$ associated with market $j$ is

$$g_{ij}(\boldsymbol{\theta}_i) = \theta_{ij} - B_j/n, \forall i \in [n], j \in [m].$$

Let $\boldsymbol{g}_i(\boldsymbol{\theta}_i) := \mathrm{col}\left(g_{i1}(\boldsymbol{\theta}_i), \cdots, g_{im}(\boldsymbol{\theta}_i)\right), \forall i \in [n]$.

The cost function of firm $i$ is defined as

$$J_i = \boldsymbol{d}_i^\top \boldsymbol{\theta}_i - \sum_{j=1}^m p_j \theta_{ij},$$

where $\boldsymbol{d}_i = \mathrm{col}\,(d_{i1}, \cdots, d_{im})$ and $d_{ij}$ represents the cost that firm $i$ sells a unit of its product to the $j$th market, $\forall i \in [n], j \in [m]$. $\boldsymbol{d}_i$ includes the cost of raw material, transportation, maintenance, etc. In $J_i$, the term $p_j$ denotes the unit demand price of market $j$ determined by its market demand curve and the total production quantity, given by

$$p_j = \xi_j + \Lambda_j \left( c_j + \frac{1}{d_j} \sum_{i=1}^n \theta_{ij} \right)^{-\frac{1}{\tau_j}}, \forall j \in [m], \tag{A37}$$

where $c_j, d_j \Lambda_j$, and $\tau_j > 0$ are constants, $\xi_j$ is a random variable. Due to the interaction between firms and markets, the demand price can fluctuate with production quantities, represented by $\xi_j \sim \mathcal{D}_j(\boldsymbol{\theta})$. Note that the quantity-dependent distributions $\mathcal{D}_j(\boldsymbol{\theta})$ for all $j \in [m]$ are unknown by players. For any $j \in [m]$, the variable $\xi_j$ is defined as

$$\xi_j = \xi_j^o + \varepsilon \frac{\alpha_j}{\sum_{j'=1}^m \alpha_{j'}} \left( \sum_{i=1}^n \theta_{ij} \right),$$

where $\xi_j^o$ is the random base component, $\varepsilon \geq 0$ represents the performative strength of markets, and $\alpha_j$ is the relative strength of market $j$ for any $j \in [m]$. According to the law of supply and demand, an increased production quantity generally decreases a market's demand price, which corresponds to the setup that $\alpha_j \leq 0$ for all $j \in [m]$. Thus, the objective of each play $i \in [n]$ in the network Cournot game is formulated by

$$\min_{\boldsymbol{\theta}_i \in \boldsymbol{\Omega}_i} \quad \mathbb{E}_{p_j \sim \mathcal{D}_j(\theta_{ij}, \forall i \in [n]), j \in [m]} \left[ \boldsymbol{d}_i^\top \boldsymbol{\theta}_i - \sum_{j=1}^m p_j \theta_{ij} \right]$$

$$\text{subject to} \quad \theta_{ij} + \sum_{i' \neq i} \theta_{i'j} \leq B_j, \forall j \in [m].$$

In the simulation, we set $n = 5$ and $m = 3$. The network structure is as depicted in Fig. 3. Each element of the communication weight matrix $A = (a_{ij})_{n \times n}$ is set to be $a_{ij} = \frac{1}{|\mathcal{N}_i|}$, and $|\mathcal{N}_i|$ is the cardinality of $\mathcal{N}_i$. The production capacity $Q_i$ is randomly and uniformly drawn from $[10, 12]$ for all $i \in [5]$, and the market's capacity $B_j$ is randomly and uniformly drawn from $[10, 15]$ for all $j \in [m]$. All entries in $\boldsymbol{d}_i, \forall i \in [n]$ are randomly and uniformly drawn from $[1, 1.5]$. The distribution of $\xi_j^o$ is set to $\min(\max(\mathcal{N}(2.5, 1), 2.5), 7.5)$. The performative power $\alpha_j$ is randomly and uniformly drawn from $(-1, 0]$, for all $j \in [3]$. Other settings are: $\Lambda_j = 10, c_j = 10, d_j = 5$ and $\tau_j = 2, \forall j \in [3]$.

Fig. 4 compares the demand prices of three markets at PSE and NE with performative strength $\varepsilon = 0.2, 0.4$, and $0.6$ and Fig. 5 compares the corresponding serving quantities of five firms to these three markets. The results suggest that, although a larger performative strength leads to a wider gap, the difference in these two indicators between the PSE and NE remains insignificant. This confirms the effectiveness of PSE solutions and our distance analysis between the PSE and NE as stated in Theorem 3.5.

### F.2 Ride-Share Market

We further examine an example of a ride-share market, where multiple platforms compete to maximize their individual revenue by offering shared rides in competitive areas, taking into account operational constraints and market demands. This experiment builds upon the semi-synthetic simulation conducted in (Narang et al., 2023), adapting it to our constrained noncooperative game setting.

Consider a ride-share market with $n$ platforms competing in $m$ areas. Each platform $i \in [n]$ aims to maximize its revenue by determining the quantities it offers at the $j$th area, denoted as $\theta_{ij}$, for all $j \in [m]$. Let $\boldsymbol{\theta}_i = [\theta_{i1}, \cdots, \theta_{im}]^\top$. The total number of rides provided by each platform $i$ cannot exceed a predefined limit $Q_i$, given by $\sum_{j=1}^m \theta_{ij} \leq Q_i, \forall i \in [n]$. Let $p_j$ denote the demand price

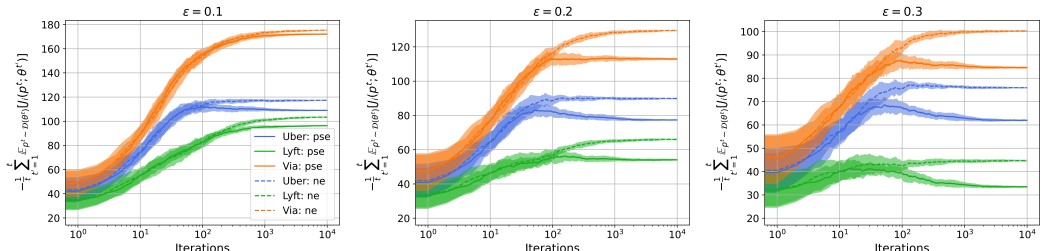

Figure 6: Convergence of the time-average revenues of three platforms.

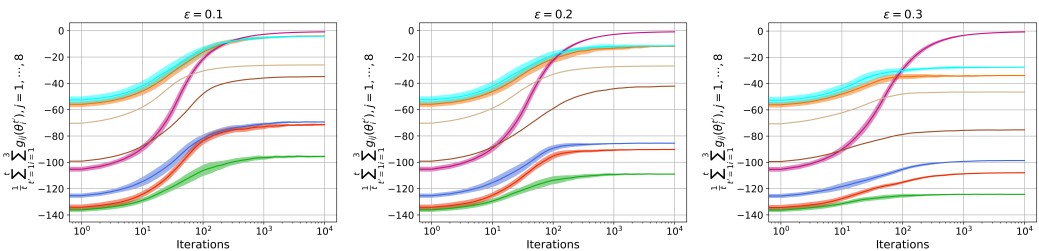

Figure 7: Convergence of the time-average constraint violations at eight areas.

at the $j$th location, which fluctuates with the total offered quantity at the area following the law of supply and demand. We adopt the same model for $\{p_j\}$ as in the network Cournot game, given by (A37). Additionally, the maintenance costs associated with platform operations may vary across locations due to factors such as distance or labor costs. Let $\boldsymbol{d}_i \in \mathbb{R}^m$ represent the cost vector of platform $i$ at all areas. Then, the inverse of the revenue function for each platform can be expressed as

$$J_i = -\sum_{j=1}^{m} p_j \theta_{ij} + \boldsymbol{d}_i^\top \boldsymbol{\theta}_i, \forall i \in [n].$$

Assume that each platform only offers one type of ride. Considering the diverse ride characteristics, such as shape and speed, we use $h_i$ to denote the spatial occupancy of each ride offered by platform $i$. The accommodated ride quantity at each location is constrained by $B_j$ due to parking availability and road conditions, such that $\sum_{i=1}^{n} h_i \theta_{ij} \leq B_j$. Then, the objective of each platform $i \in [n]$ in the ride-share market is formulated as

$$\min_{\boldsymbol{\theta}_i \in \boldsymbol{\Omega}_i} \quad \mathbb{E}_{p_j \sim \mathcal{D}_j(\theta_{ij}, \forall i \in [n]), \forall j \in [m]} \left[ -\sum_{j=1}^{m} p_j \theta_{ij} + \boldsymbol{d}_i^\top \boldsymbol{\theta}_i \right] \tag{A38}$$
$$\text{subject to} \quad h_i \theta_{ij} + \sum_{i' \neq i} h_{i'} \theta_{i'j} \leq B_j, \forall j \in [m].$$

The simulation setup is based on dataset from a prior Kaggle competition.[2] Our study focuses on three ride-share platforms (Uber, Lyft, and Via) and eight competing areas within New York. We randomly and uniformly assign the total number of rides, $Q_i$, from the range $[200, 400]$ for each platform $i \in [3]$. Similarly, the accommodated capacity, $B_j$, is randomly and uniformly drawn from $[50, 150]$ for all $j \in [8]$. All entries in $\boldsymbol{d}_i, \forall i \in [n]$ are randomly and uniformly drawn from $[0.2, 2.2]$. The distribution of $\xi_j^o$ is set as $\min(\max(\mathcal{N}(1, 1), 1), 5)$. Additionally, we set the following values for all areas $j \in [8]$: $\Lambda_j = 5, c_j = 5, d_j = 5$, and $\tau_j = 2$.

Fig. 6 compares the convergence of the time-average revenues of these three platforms: Uber, Lyft, and Via, denoted by $-\frac{1}{t}\sum_{t'=1}^{t} \mathbb{E}_{\boldsymbol{p}^t \sim \mathcal{D}(\boldsymbol{\theta}^t)}[J_i(\boldsymbol{p}^t; \boldsymbol{\theta}^{t'})]$. We consider three performative strengths:

[2]The data is publicly available at https://www.kaggle.com/brllrb/uber-and-lyft-dataset-boston-ma

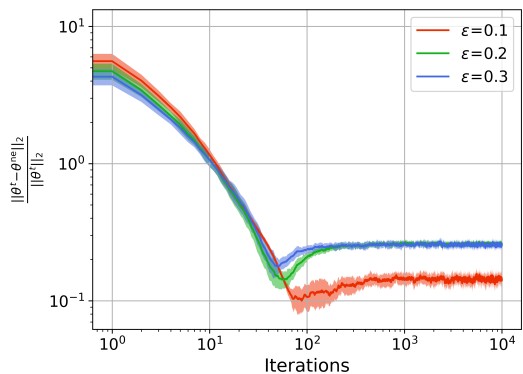

Figure 8: The normalized distance between $\boldsymbol{\theta}^t$ and $\boldsymbol{\theta}^{\mathrm{ne}}$.

$\varepsilon = 0.1, 0.2,$ and $0.3$. Similarly to Fig. 2 (b), we compare the performance of Algorithm 1, represented by "pse", and the performance of Algorithm 1 with perfect knowledge of data distributions $\mathcal{D}_j(\boldsymbol{\theta})$ for all $j \in [m]$. It is observed that, with a mild performative strength $\varepsilon$, the revenues achieved by the "pse" are close to those of the "ne" for all three platforms. However, as $\varepsilon$ increases, the gap between the two approaches widens, although it remains relatively small. This observation confirms the analytical result presented in Theorem 3.5.

Fig. 7 shows the convergence of the time-average constraint violations at eight areas by Algorithm 1, denoted by $\frac{1}{t} \sum_{t'=1}^{t} \sum_{i=1}^{3} g_{ij}(\boldsymbol{\theta}_i^{t'}), j = 1, \cdots, 8$, with performative strengths of $\varepsilon = 0.1, 0.2,$ and $0.3$. The constraints hold for all three performative strengths. However, as $\varepsilon$ increases, the platform tends to allocate fewer rides. This may be attributed to larger market fluctuations associated with a higher $\varepsilon$, leading to a more conservative allocation.

Fig. 8 compares the normalized distance between $\boldsymbol{\theta}^t$ and the NE point $\boldsymbol{\theta}^{\mathrm{ne}}$, denoted as $\|\boldsymbol{\theta}^t - \boldsymbol{\theta}^{\mathrm{ne}}\|_2 / \|\boldsymbol{\theta}^t\|_2$, with performative strengths: $\varepsilon = 0.1, 0.2,$ and $0.3$. The result is quantitatively analogous to the findings presented in Fig. 8. Firstly, $\boldsymbol{\theta}^t$ gradually approaches $\boldsymbol{\theta}^{\mathrm{ne}}$ with iterations. Secondly, a higher performative strength leads to a wider normalized distance between the convergent point of $\boldsymbol{\theta}^t$ and $\boldsymbol{\theta}^{\mathrm{ne}}$.

Fig. 9 compares the demand prices of eight areas and the ride quantities offered to them by three platforms at PSE and NE. We consider performative strengths $\varepsilon = 0.1$ and $\varepsilon = 0.3$. It is observed that the values of these indicators at the PSE and NE are close to each other when $\varepsilon = 0.1$. However, a noticeable discrepancy arises when $\varepsilon = 0.3$.

Additionally, we display the demand prices of eight areas in New York in Fig. 10, with different performative strengths: $\varepsilon = 0.1, 0.2,$ and $0.3$. It can be observed that, while prices vary by location, smaller values of $\varepsilon$ generally correspond to higher prices. The offered quantities of these three platforms to the eight locations are illustrated in Fig. 11. The results indicate a conservative allocation as the performative strength increases. Furthermore, with the cost of these three platforms at different locations in Fig. 12, we obtain the revenues of the platforms Uber, Lyft, and Via in different areas, as illustrated in Fig. 13. Clearly, performativity has an inverse effect on revenues, and the stronger the performative strength, the lower the revenues.

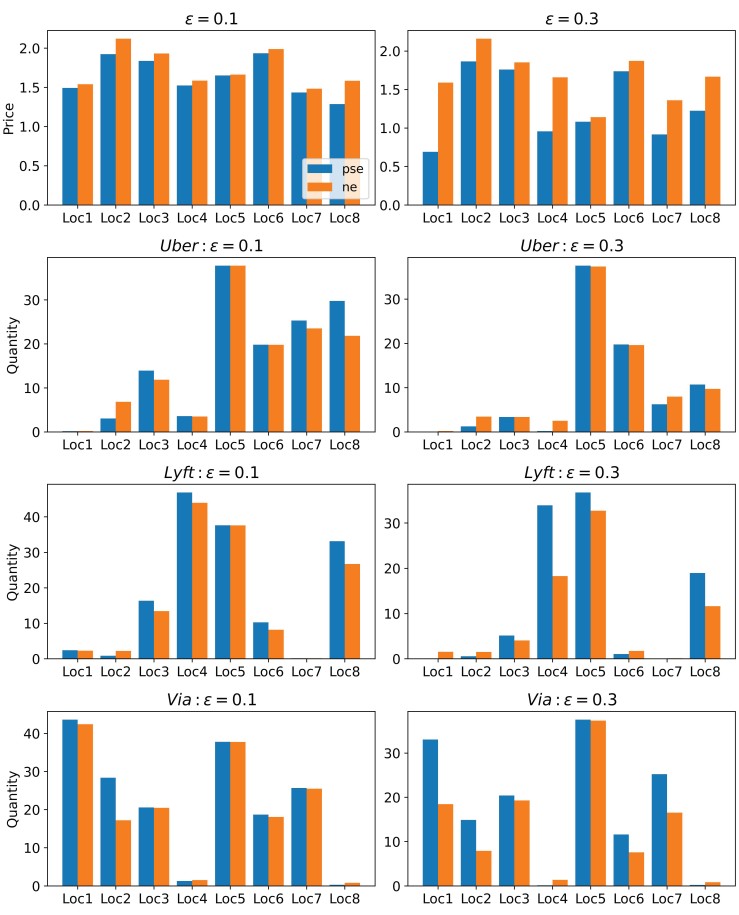

Figure 9: The demand prices of eight areas and the ride quantities offered to them by three platforms.

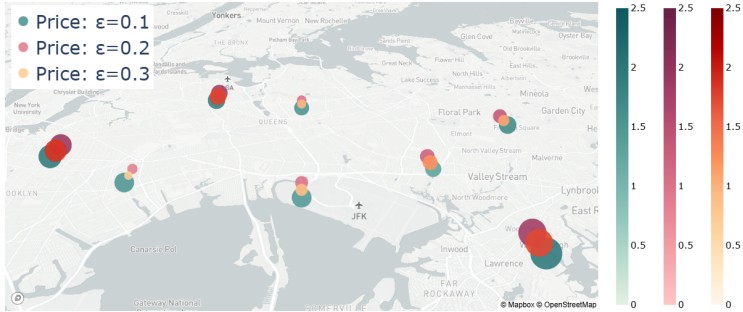

Figure 10: The demand prices of different areas.

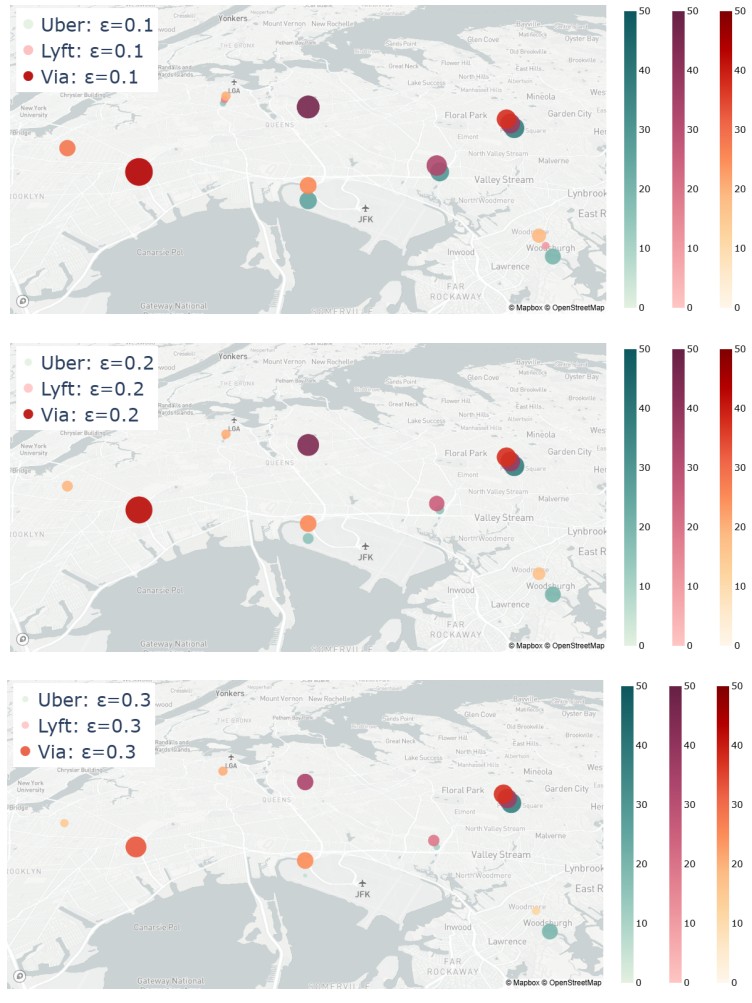

Figure 11: The quantities of platforms offered to different areas.

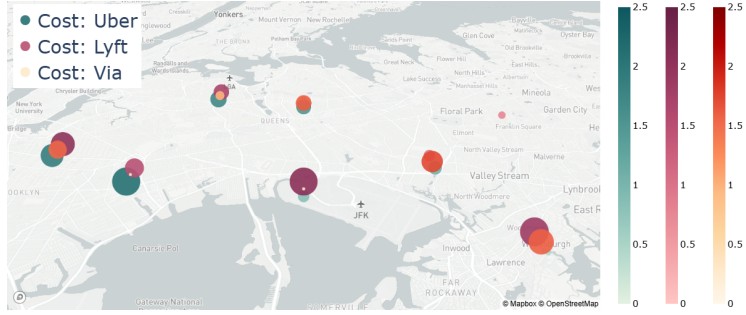

Figure 12: The cost of platforms in different areas.

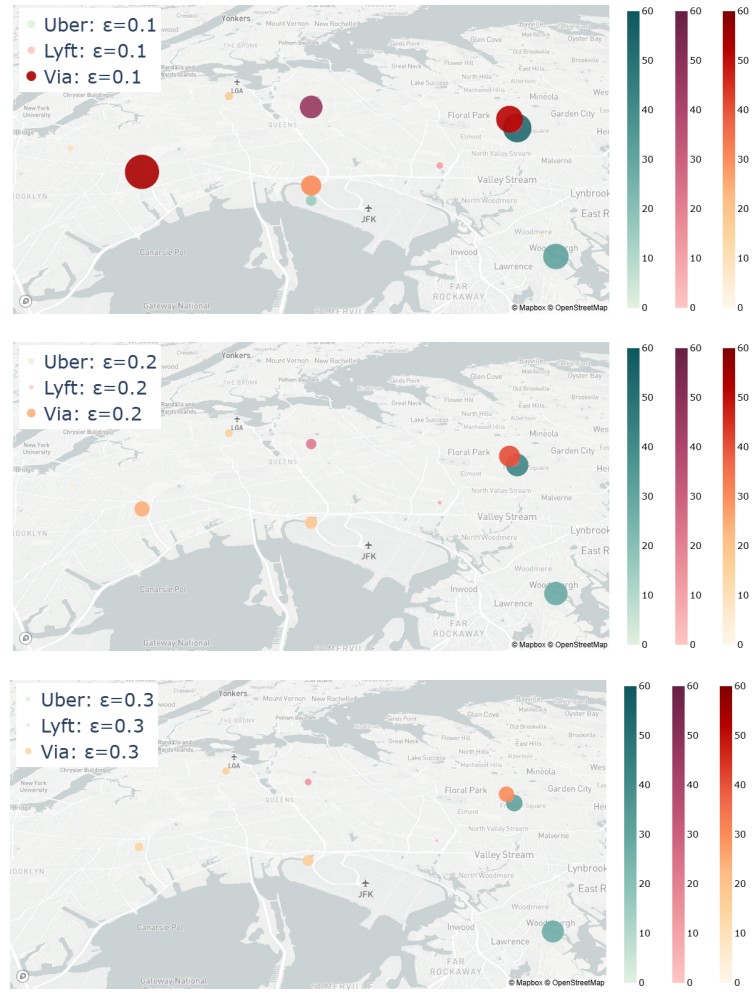

Figure 13: The revenues of platforms in different areas.

