# OpenReview forum: "Decentralized Noncooperative Games with Coupled Decision-Dependent Distributions"
_NeurIPS.cc/2024/Conference — NeurIPS 2024 poster_

### Official Review · Reviewer_pBFf · 2024-07-11

**Soundness:** 3
**Presentation:** 2
**Contribution:** 2
**Rating:** 6
**Confidence:** 2

**Summary:**

This paper examines endogenous distribution shifts in learning systems, where deployed models influence their environments, altering data distributions. The authors first prove a set of sufficient conditions for the existence and uniqueness of performative stable equilibrium (PSE) and Nash equilibrium (NE). The main contribution of the paper is to show that the distance between PSE and NE scales with the distributional shift parameter. They also provide a primal-dual algorithm for computing PSE, which achieves sublinear convergence rates for both performative regrets and constraint violations.

**Strengths:**

Although I did not check the proof, the results of the paper appear to be correct. I appreciate that the authors provide intuition for the underlying mechanism. I am not an expert in this particular field, so I cannot comment much on the contribution. My impression is that the theoretical contribution is fairly significant and the convergence rate shown in this paper is strong.

**Weaknesses:**

No

**Questions:**

1. Is Theorem 3.4 just a simple corollary of Theorem 3.3?

2. The paper provides a sufficient condition for the existence and uniqueness of NE and PSE, but it is not necessary. Can you comment on the potential necessity for the existence and uniqueness of NE and PSE as well?

**Limitations:**

The authors did discuss the limitation

---

> ### Author Rebuttal · Authors · 2024-07-31
>
> **We sincerely appreciate the reviewer for recognizing our contributions and for the constructive comments. Our point-to-point responses to concerns on Questions are given below.**
>
> **Reply to Question 1:** Theorem 3.3 pertains to the existence and uniqueness of the performative stable equilibrium (PSE). Its characterization is based on the contraction condition of the dynamics of repeated risk minimization (RRM), as defined in Line 208. Theorem 3.4 pertains to the existence and uniqueness of the Nash equilibrium (NE), with the proof established on the strong monotonicity of the performative game (1). These two theorems follow distinct analytical processes, and Theorem 3.4 should not be considered a corollary of Theorem 3.3.
>
> **Reply to Question 2:** The study of equilibrium is fundamental in game theory because it ensures predictability and stability in strategic interactions among rational players. The existence of an equilibrium guarantees that there is at least one outcome where all players' strategies are mutually optimal responses, providing a foundation for predicting behavior in competitive scenarios. Uniqueness further enhances this predictability by ensuring a single, definitive outcome, eliminating ambiguity, and facilitating more straightforward strategic decision-making. This evaluation is essential for practical applications in economics, political science, and beyond, where reliable and stable outcomes are necessary for effective planning and analysis.
>
> We thank the reviewer for pointing this out, and we have included a discussion on this point in our manuscript. Moreover, we would like to highlight one of our contributions on the equilibrium analysis.
>
>  This paper makes a significant technical contribution by presenting the first upper bound on the distance between PSE and NE, which is significant to understand the impact of data performativity in games. Such bounds have been a major focus in prior research on performative optimization but are challenging to quantify in games due to the competitive nature of players. We leverage relations provided by strong duality and derive a result comparable to the findings in performative optimization settings.
>
>
> **Thank you again for your review and for recognizing our contributions.** We appreciate the strengths you have identified and we would like to reiterate our key contributions, which have been positively acknowledged by other reviewers. For instance, Reviewer QNeF remarked that:
> > I believe that the paper makes a concrete contribution that fills a natural gap left in the literature. One can view most of the results derived in this paper as the natural counterparts of the results obtained by Perdomo et al. but in a more general non-cooperative game setting, which can be motivated in a number of applications. The results obtained in the paper take an important step toward characterizing the performative effect in multi-player non-cooperative settings. The theoretical results are technically non-trivial, and all claims appear to be sound.
>
> **Given our contributions to the field of performative prediction and noncooperative games, we respectfully request that you consider improving the rating scores to enhance the likelihood of acceptance for this work. We thank you for your time and thoughtful consideration.**

---

> > ### Comment · Reviewer_pBFf · 2024-08-11
> >
> > Many thanks for your response.

---

### Official Review · Reviewer_dWNU · 2024-07-12

**Soundness:** 3
**Presentation:** 3
**Contribution:** 2
**Rating:** 5
**Confidence:** 3

**Summary:**

This paper studies decentralized non-cooperative games with endogenous distribution shifts. In the model, each player aims to minimize its expected risk over a distribution that changes with their own and other players’ decisions, subject to a coupled constraint. The paper establishes conditions for the existence and uniqueness of Nash equilibria (NE) and and performative stable equilibria (PSE), and further bounds the distance between NE and PSE. The paper then proposes a primal-dual algorithm that computationally-efficiently solves the problem.

**Strengths:**

1. The paper makes solid contributions on formulating and solving decentralized non-cooperative games with endogenous distribution shifts. The model is richer than existing works.
2. The paper first bounds the distance between PSE and NE. This is an important result in the setting of endogenous distribution shift because it characterize how the distribution shift coefficients influence the connection between different equilibrium notions.
3. The paper proposes a primal-dual algorithm for finding the PSE that is both computationally efficient and decentralized. The regret rate of matches the decentralized noncooperative game.
4. The paper carries numerical experiments that validates the theoretical results.

**Weaknesses:**

1. Several existing works have studied noncooperative games under performativity. This work extend the existing formulation by considering coupled constraints and decentralized setting. However, the constraint condition is not the fundamental difficulty in optimizing non-cooperative games as is known in variational inequalities literatures. Furthermore, the proposed algorithm relies on the assumption that players can communicate through a graph A with irreducible and doubly-stochastic weight matrix. The assumption basically ensures that the decentralized information can be disseminated effectively and makes decentralized optimization trivial. In conclusion, the problem studied in this work seems incremental.
2. The proposed algorithm looks very similar to that in Lu et al., 2020.

**Questions:**

Why is the condition for $\mu$ different in E&U of PSE and NE?

In assumption 4.1, what does $\nabla_\theta J(\xi,\delta)$ mean? why is the right hand side independent on $\delta$?

Some minor issues:
Some citations should be changed to \citep instead of \citet

Line 224: $p_{ij} \propto $ should be $p_{ij} \leq$

**Limitations:**

The weakness section lists some limitations of this work.

---

> ### Author Rebuttal · Authors · 2024-07-31
>
> **We sincerely appreciate the reviewer for recognizing our contributions and for the constructive comments. Our point-to-point responses to concerns on Weaknesses and Questions are given below.**
>
> **Reply to Weakness 1:**
> 1. First, we would like to clarify that only two existing works, namely, Narang et al. (2023) and Wang et al. (2023), have explored performative games to the best of our knowledge, with Narang et al. (2023) considering a centralized case and Wang et al. (2023) focusing on a relatively restrictive model, both of which are constraint-free. In contrast, our work delves into a more practical setting with a mathematically richer model. Considering constraints is crucial since many performative games have inherent restrictions, such as safety and cost constraints in transportation, relevance and diversity constraints in advertising, and risk tolerance and portfolio constraints in financial trading.  This leads to a completely different algorithm design and convergence analysis. Our work is based on the primal-dual technique, while theirs focuses on gradient descent.
>
> 2. Second, although there are works on non-cooperative games with constraints, the presence of data performativity poses significant challenges:
> - The NE evaluation needs to quantify the performative effect that the studied performative game requires the condition $\mu - \sum_{i=1}^n L_i \varepsilon_i \max_{j\in[n]} \sqrt{p_{ij}} -  \sqrt{\sum_{i=1}^n L_i^2 \varepsilon_i^2p_{ii}}>0$ to guarantee monotonicity while the non-performative counterpart (Lu et al., 2020) only requires $\mu>0$, which is an assumption of the paper.
> - The PSE is a unique concept in performative games and has not been considered in (Lu et al., 2020).
> - The convergence analysis of our algorithm necessitates meticulous control of the performative effect to maintain the convergence order. Our results demonstrate how performativity influences convergence, slowing it down by a coefficient $\tilde\mu := \mu - \sum_{i=1}^n L_i \varepsilon_i \max_{j\in[n]} \sqrt{p_{ij}}$, which offers a theoretical foundation for designing decentralized performative game systems.
>
> 3. Furthermore, the vast majority of existing research on decentralized learning is based on the assumption of a doubly-stochastic weight matrix. Without this premise, consensus on decentralized systems cannot be guaranteed, rendering meaningful analysis challenging, if not impossible.
>
> 4. Last but most important, this paper presents the first upper bound on this distance between PSE and NE, which is significant to understand the impact of data performativity in games. Such bounds have been a major focus in prior research on performative optimization but are challenging to quantify in games due to the competitive nature of players.
>
> **Reply to Weakness 2:**  Some comparisons of our work with (Lu et al., 2020) have been provided in the Reply to Weakness 1. We would like to supplement that the date performativity does not cause significant difference in the algorithm development when calculating stable points. The primary challenge lies in the performance analysis that requires bounding the deviation induced by the performative effect. This is common in the performative prediction literature where predominate works focus on finding performative stable points. Our algorithm can be extended to calculate the NE by incorporating the gradient of $D_i(\theta_i;\theta_{-i})$ with respect to $\theta_i$ in Step 7. However, estimating $D_i$ for all $i$ is computationally prohibitive as $D_i$ is related to the decisions of all players. Our algorithm results in a PSE that is not far from the NE, as demonstrated in Theorem 3.5.
>
> **Reply to Question 1:** The E&U of NE requires a more stringent condition since the computation of NE needs to take into account the gradient of $D_i(\theta_i;\theta_{-i})$ with respect to $\theta_i$ for all $i$. More specifically, NE computes the gradient of performative risk, given by $\nabla{\rm PR}(\theta) = G_{\theta}(\theta) + H_{\theta}(\theta)$, while PSE only considers the first component $G_{\theta}(\theta)$.
>
> **Reply to Question 2:**  We appreciate the reviewer's observation. Assumption 4.1 pertains to the stochastic gradient variance of $\nabla_{\delta_i} J_i(\xi_i;\delta_i, \delta_{-i})$ and $G_{\theta}^{(i)}(\delta_i, \delta_{-i}) := E_{\xi_i\sim D_i(\theta)} \nabla_{\delta_i} J_i(\xi_i;\delta_i, \delta_{-i})$. To enhance clarity, we have revised Assumption 4.1 in our manuscript.
>
> **Reply to Question 3:** Thank you for pointing this out. We have carefully revised the citation style throughout our paper.
>
> **Reply to Question 4:** Thank you for the advise.  Since $p_{ij}$ is the normalized influence parameter of $D_i$ with respective to $\theta_j$, to make sure that $\sum_{j=1}^n p_{ij}=1$,  we take $p_{ij}\propto \mathcal{O}(1/n)$.
>
> **Thank you again for your review and for recognizing our contributions.** We appreciate the strengths you have identified, as they are crucial for the evaluation of this paper and align with the observations of other reviewers. For instance, reviewer QNeF provided the following positive remark:
> > I believe that the paper makes a concrete contribution that fills a natural gap left in the literature. One can view most of the results derived in this paper as the natural counterparts of the results obtained by Perdomo et al. but in a more general non-cooperative game setting, which can be motivated in a number of applications. The results obtained in the paper take an important step toward characterizing the performative effect in multi-player non-cooperative settings. The theoretical results are technically non-trivial, and all claims appear to be sound.
>
> **Given our contributions to the field of performative prediction and noncooperative games, we respectfully request that you consider improving the rating scores to enhance the likelihood of acceptance for this work. We thank you for your time and thoughtful consideration.**

---

> > ### Comment · Reviewer_dWNU · 2024-08-12
> >
> > Thank you for the responses.

---

### Official Review · Reviewer_ReB3 · 2024-07-13

**Soundness:** 3
**Presentation:** 3
**Contribution:** 2
**Rating:** 4
**Confidence:** 4

**Summary:**

In this paper, the authors introduce the framework of decentralized noncooperative games incorporating the performativity factor and investigate the existence and uniqueness of two equilibrium concepts: Nash equilibrium and performative stable equilibrium. The distance upper bound between NE and PSE is firstly provided. To compute the PSE point under the given problem settings, a decentralized stochastic primal-dual algorithm is proposed, and corresponding convergence analyses are conducted.

**Strengths:**

The paper is well-written and easy to follow, clearly presenting the authors’ logic. The proof of the theorem is solid.

**Weaknesses:**

- Since the authors proposed the primal-dual algorithm to find the PSE solution, how can the NE be found under the performativity framework? In the Simulation part, how do the authors find $\theta^{ne}$?

- The analytical techniques of the Decentralized Stochastic Primal-Dual algorithm appear to be a special case of the approach in Wood and Dall’Anese (2023), as the primal-dual algorithm can be transformed into a min-max problem. Are there any unique analytical techniques used in this paper? Could the authors provide a comparison between this work and Wood and Dall’Anese (2023)?

[a] Wood, Killian, and Emiliano Dall’Anese. ``Stochastic saddle point problems with decision-dependent distributions." SIAM Journal on Optimization 33.3 (2023): 1943-1967.

- In decentralized noncooperative games, each player is assumed to be selfish. Why would they want to communicate with their neighbors and share their information via a graph? In Line 4 of Algorithm 1, the estimator is constructed by a weighted average of the local parameter and neighbors' information. Is this reasonable in a competitive problem setting? I am confused by the logic of these game settings.

- In Line 6 of Algorithm 1, the gradient is given as $\nabla_{\theta_i} J_i(\xi_i^t; \theta_i^t, \hat{\theta}_i^t) + \underline{\gamma_t} \nabla g_i(\theta_i^t)^\top \lambda_i^t$. Is this $\gamma_t$ a typo? The same concern arises in the equation after line 278.

- Is Assumption 4.1 used in the proof of Theorem 4.2? Also, in Assumption 4.1, where is the $\delta$ in the right-hand side of the assumption inequality?

- In Line 347, the performative strengths are set as $\rho=0.2, 0.4, 0.6$. I suggest using $\varepsilon$ to denote performative strength to maintain contextual consistency. Additionally, does this setting imply that all ${\cal D}_i(\theta_i)$ admit the same shifting strength?

- How does the spectral gap of graph ${\cal G}(A)$, i.e., graph topology, affect the convergence results in Theorem 4.2?

- Since Theorem 3.5, the upper boudn of PSE and NE is one of the major contribution of this paper, is there any corresponding simualtion which can support the theoretical results?

**Questions:**

**Typo List**:

- Line 176, "there exists a constant **$C\leq 0$** such that ...", it should be $C\geq 0.$

- After Line 208, second line of optimization problem, $g_i(u_i)$ should be $g_i(\theta_i)$.

- In the problem after Line 727, symbol $\theta$ does not use same font. Same typo in equation (A37).

**Limitations:**

Please answer the questions in weakness part. I will reconsider the rating based on the author's response.

---

> ### Author Rebuttal · Authors · 2024-07-31
>
> **Thank you for recognizing our contributions and for the constructive comments. Our point-to-point responses are given below.**
>
> **Reply to Weakness 1:** Given the explicit expressions of the decision-dependent distributions $D_i(\theta)$ for all $i$, we can compute the exact gradient of the performative risk PR($\theta$). Consequently, the Nash Equilibrium (NE) can be asymptotically determined through best response dynamics. This approach is structurally similar to repeated risk minimization but crucially accounts for the decision-dependent nature of the distributions, rather than fixing them at the current deployment for each update. In our simulations, we employ this method to calculate the NE, which serves as our baseline for comparison purpose. We appreciate the reviewer's observation and have incorporated detailed explanations of the NE computation process in the simulation section of our manuscript.
>
> **Reply to Weakness 2:** We appreciate the reviewer for bringing this reference to our attention. We have incorporated a comprehensive comparison of our work with [a] in our paper, as summarized below:
> - [a] investigates centralized minimax optimization, whereas our work addresses a more complex model involving decentralized noncooperative players with partial information observation. Our analysis needs to account for consensus and competition among players. The NE of a game generally does not correspond to the optimal solution of its corresponding optimization problem.
> - [a] requires a more stringent assumption of strong-convexity-strong-concavity on the minimax problem. In contrast, our paper only necessitates the stochastic gradient mapping $\nabla J(\xi; \theta)$ to be monotone, which is a weaker condition than strong convexity.
> - We characterize the existence and uniqueness of the NE and PSE, which are concepts unique to game theory.
>
> **Reply to Weakness 3:** In noncooperative games, players are assumed to be both selfish and rational. Despite their self-interest, players have incentives to share information as it enables more effective optimization of their individual strategies, potentially leading to improved personal outcomes and enhanced overall game stability. The vast majority of existing research on decentralized games is predicated on this assumption of rational players. Without this premise, player behavior could become unpredictable, rendering meaningful analysis challenging, if not impossible.
>
> **Reply to Weakness 4:** The symbol $\gamma_t$ is not a typo; rather, it serves as a crucial coefficient for regulating the step size of the optimization process in Algorithm 1.
>
> **Reply to Weakness 5:** We appreciate the reviewer's observation. Assumption 4.1 pertains to the stochastic gradient variance of $\nabla_{\delta_i} J_i(\xi_i;\delta_i, \delta_{-i})$, which forms the foundation for Theorem 4.2. To enhance clarity, we have revised Assumption 4.1 and added it into Theorem 4.2 as a necessary condition.
>
> **Reply to Weakness 6:** Thank you for the valuable advice. We have substituted $\rho$ with $\varepsilon$ to denote the performative strength. You are right that this setting implies that all markets exhibit an equivalent shifting strength.
>
> **Reply to Weakness 7:** Let $\sigma_2(A)$ denote the spectral gap of the weight matrix $A$. The spectral gap affects the convergence results in Theorem 4.2 by a coefficient $\mathcal{O}\left(\frac{1}{1-\sigma_2(A)}\right)$, which is shown in Lemma E.5 of the Appendix. We omit this coefficient in Theorem 4.2 since this paper primarily concentrates on the impact of data performativity, which slows down convergence by a coefficient $\tilde\mu := \mu - \sum_{i=1}^n L_i \varepsilon_i \max_{j\in[n]} \sqrt{p_{ij}}$.
>
> **Reply to Weakness 8:** Thank you for the question.  In the simulation of the networked Cournot game, Fig. 2(b) compares the total revenue of all firms at the PSE and NE. Figure 4 contrasts the demand prices across five markets, while Fig. 5 illustrates the quantities served by five firms in different markets at both PSE and NE. For the simulation of the ride-share market, Fig. 6 compares the time-averaged revenues of three platforms at PSE and NE, whereas Fig. 9 depicts the demand prices in distinct areas and the corresponding ride quantities offered by various markets at both PSE and NE.
>
> **Reply to Questions:** Thank you for your careful review.
>
> 1. We have rectified this typo in our manuscript.
> 2. In Line 208, the optimization variable is $u_i$ instead of $\theta_i$, thereby the expression of $g_i(u_i)$.
> 3. In Line 727 and equation (A37), the bold font $\boldsymbol{\theta}$ represents a vector that $\boldsymbol\theta_i = [\theta_{i1},\cdots,\theta_{im} ]^{\top}$ for all $i$, where $\theta_{ij}$ denotes the product quantity that firm $i$ sells to the $j$th market and is represented in non-bold font as a scalar.
>
> **We sincerely appreciate the reviewer's meticulous review and useful comments**. We would like to reiterate our key contributions, which have been positively acknowledged by other reviewers. For instance, Reviewer QNeF remarked that:
> > I believe that the paper makes a concrete contribution that fills a natural gap left in the literature. One can view most of the results derived in this paper as the natural counterparts of the results obtained by Perdomo et al. but in a more general non-cooperative game setting, which can be motivated in a number of applications. The results obtained in the paper take an important step toward characterizing the performative effect in multi-player non-cooperative settings. The theoretical results are technically non-trivial, and all claims appear to be sound.
>
> Similar commendations can be found in the remarks of the remaining two reviewers.
>
> **Given our contributions to the field of performative prediction and noncooperative games, we respectfully request that you consider improving the rating scores. We thank you for your time and thoughtful consideration.**

---

> > ### Author Response · Authors · 2024-08-13
> > **Follow-Up: Request for NeurIPS Rebuttal Response**
> >
> > Dear Reviewer ReB3:
> >
> > Thank you again for taking the time to review our paper and provide your valuable feedback. As the rebuttal period comes to a close, we wanted to kindly check if our clarifications in the rebuttal have satisfactorily addressed the concerns raised in your initial review. Your feedback is invaluable to us.
> >
> > If so, we respectfully request you to consider updating your review score based on our responses. However, if you have any remaining questions or need further clarification from us, please do not hesitate to let us know.
> >
> > Once again, we sincerely appreciate your time and consideration. Your prompt response would be greatly appreciated.
> >
> > Sincerely,
> >
> > Authors of the paper

---

> ### Comment · Area_Chair_tDpu · 2024-08-13
>
> Dear Reviewer,
>
> I would be grateful if you could respond to the authors' rebuttal.
>
> Thanks,
> AC

---

> ### Comment · Reviewer_ReB3 · 2024-08-14
>
> I appreciate the authors’ detailed response. Regarding the novelty of the technique, I agree with Reviewer dWNU that the constraint condition is not the primary challenge in non-cooperative games, as the use of primal-dual algorithms to tackle Lagrangian form is a natural approach. Existing work [a] already provides a more general framework for min-max problems with performativity. Additionally, the convexity assumption on the constraint in Assumption 2.5 of the submitted paper appears to be somewhat strong.
>
> In Section 4, the convergence analysis in Theorem 4.2 indicates that performative shifts affect the regret bound by a constant factor, but this may not be an intrinsic challenge that performativity can bring in the analysis of primal-dual algorithm.
>
> Therefore, I would like to maintain my original rating of borderline reject.

---

### Official Review · Reviewer_QNeF · 2024-07-22

**Soundness:** 3
**Presentation:** 3
**Contribution:** 3
**Rating:** 6
**Confidence:** 3

**Summary:**

The paper studies the effect of endogenous distribution shifts stemming from the underlying interaction of the learning system, as formalized in the recent framework of performative prediction. In particular, the paper focuses on the performative effect in a non-cooperative game in which players endeavor to minimize individual cost functions while satisfying coupled constraints. They consider two natural equilibrium concepts, and provide sufficient conditions for their existence and uniqueness. Further, they provide a bound relating the distance between those two, and they develop a decentralized stochastic primal-dual algorithm for efficiently computing one equilibrium that achieves sublinear rate under various assumptions. Numerical experiments support the theoretical results.

**Strengths:**

Overall, I believe that the paper makes a concrete contribution that fills a natural gap left in the literature. One can view most of the results derived in this paper as the natural counterparts of the results obtained by Perdomo et al. but in a more general non-cooperative game setting, which can be motivated in a number of applications. The results obtained in the paper take an important step toward characterizing the performative effect in multi-player non-cooperative settings. The theoretical results are technically non-trivial, and all claims appear to be sound; I did not find any notable issue. The related work section is also quite thorough, and the most relevant papers have been discussed adequately to the most part. Finally, the presentation and the writing are of high quality, and the main body goes a great job at giving high-level sketches of the main ideas behind the proofs.

**Weaknesses:**

In terms of weaknesses, I believe that some of the assumptions necessary for the theoretical results (Assumptions 2.1-2.5) require further justification. In most cases, the authors simply claim that the assumption is quite standard in some other settings, but that itself is not sufficient justification. The authors need to argue that the assumptions are also aligned with the applications considered in their paper, otherwise they appear to be artificial. Perhaps an example could serve to make that point.

Besides the point above, an important caveat of the paper is that most of the results appear to follow from relatively standard techniques in the literature, and so the technical novelty is limited. Although the setting considered in the paper is, to my knowledge, novel, it seems that it can be readily reduced to well-understood problems in game theory.

**Questions:**

- The citation style is not used correctly; for example, "unseen data El Naqa and Murphy (2015)" should instead be "unseen data (El Naqa and Murphy, 2015)". That is, use parenthesis when the citation is not syntactically part of the sentence. This is done consistently thoughout the paper.
- I would like to see more discussion concerning the papers by Li et al. (2022) and Piliouras and Yu (2023). It is not quite clear how those results relate to those in the paper.

**Limitations:**

The authors have adequately addressed the limitations.

---

> ### Author Rebuttal · Authors · 2024-08-01
>
> **We sincerely appreciate the reviewer for recognizing our contributions and for the constructive comments. Our point-to-point responses to concerns on Weaknesses and Questions are given below.**
>
> **Reply to Weakness 1:** Thank you for the useful comment. We have provided justifications for the conditions outlined in Assumptions 2.1-2.5 through both simulation examples, specifically the networked Cournot game and the ride-share market. Below is an excerpt of the justification for the networked Cournot game:
>
> > Given the formulation of the networked Cournot game, it is evident that the stochastic gradient mapping of $J(\cdot)$ adheres to Assumption 2.1, as expressed by:
> >
> > $$\left\langle \nabla J\left(\xi;\theta\right) -\nabla J\left(\xi;\theta^{\prime}\right), \theta - \theta^{\prime} \right\rangle
>   \geq \mu ||\theta - \theta^{\prime}||_2^2$$
> >
> > with $\mu =2\varepsilon \min_j \frac{\alpha_j}{\sum_{j^{\prime}=1}^m\alpha_{j^{\prime}}} $, where $\varepsilon\geq 0$ denotes the performative strength of markets, and $\alpha_j$ represents the relative strength of market $j$ for any $j\in[m]$.
> >
> > Moreover, for any $i$, the quantity-dependent distribution $D_i(\theta)$ satisfies
> >
> > $$\mathcal{W}_1(D_i(\theta), D_i(\theta^{\prime})) \leq\varepsilon\sqrt{\sum_j\frac{\alpha_j}{\sum_j^\prime\alpha_j^\prime}||\theta_j-\theta_j^{\prime}||_2^2}$$
> >
> > in accordance with the condition in Assumption 2.
> >
> > Furthermore, each local cost function $J_i(\cdot)$ demonstrates smoothness with a parameter $L \geq \varepsilon \max_j\frac{\alpha_j}{\sum_{j^{\prime}=1}^m\alpha_{j^{\prime}}}$ and thus satisfies the condition in Assumption 4.
> >
> > Given that the output quantity of each firm is constrained by an upper bound $Q_i$, Assumption 3 is inherently fulfilled with $C>\max_iQ_i$.
> >
> > Lastly, the Lipschitz condition of constraints in Assumption 5 holds true for any $G_g\geq 1$, under the accommodating quantity constraints of markets.
> >
> > Overall, the formulation of the networked Cournot game satisfies all assumptions required in our study.
>
> **Reply to Weakness 2:** We would like to highlight the following technical contributions of this paper:
>
> 1. We evaluate the conditions for the existence and uniqueness of both NE and PSE. The NE evaluation needs to quantify the performative effect that the studied performative game requires the condition $\mu - \sum_{i=1}^n L_i \varepsilon_i \max_{j\in[n]} \sqrt{p_{ij}} -  \sqrt{\sum_{i=1}^n L_i^2 \varepsilon_i^2p_{ii}}>0$ to guarantee monotonicity while the non-performative counterpart (Lu et al., 2020) only requires $\mu>0$, which is an assumption of the paper. The PSE is a unique concept in performative games and has not been considered in conventional games without data performativity.
>
> 2. This paper makes a significant technical contribution by presenting the first upper bound on the distance between PSE and NE, which is significant to understand the impact of data performativity in games. Such bounds have been a major focus in prior research on performative optimization but are challenging to quantify in games due to the competitive nature of players. We leverage relations provided by strong duality and derive a result comparable to the findings in performative optimization settings.
>
> 3. The convergence analysis of our decentralized algorithm needs to carefully control the deviation caused by data performativity. The decentralized nature and the presence of constraints further complicate our analysis. Nevertheless, we derived a result that preserves the order of convergence as the case without data performativity (Lu et al., 2020). Our results demonstrate how performativity influences convergence, slowing it down by a coefficient $\tilde\mu := \mu - \sum_{i=1}^n L_i \varepsilon_i \max_{j\in[n]} \sqrt{p_{ij}}$, which offers a theoretical foundation for designing decentralized performative game systems.
>
> **Reply to Question 1:** Thank you for pointing this out. We have carefully revised the citation style throughout our paper.
>
> **Reply to Question 2:**  Thank you for the advice. We have added more discussions on the works Li et al. (2022) and Piliouras and Yu (2023) in our paper. Similarly to our work, Li et al. (2022) and Piliouras and Yu (2023) also studied multiagent systems with performativity but in constraint-free scenarios.  Specifically, Li et al. (2022)  focused on decentralized optimization with consensus-seeking agents, where the data distribution of each agent depends solely on its own decision. They investigated conditions for the existence and uniqueness of performative stable solutions and provided theoretical analysis on the convergence of gradient descent algorithms in multiagent performative prediction systems. In contrast, Piliouras and Yu (2023) studied multiagent systems in a centralized fashion with data distributions restricted to location-scale families. Under this setting, they explored the existence conditions of performative stable points and demonstrated the equivalence of performative stability and performative optimality.
>
> **We sincerely appreciate the reviewer's positive assessment. The strengths you have identified are crucial for evaluating this paper and align with observations from other reviewers.**
>
> **Given our contributions to the field of performative prediction and noncooperative games, we respectfully request that you consider improving the rating scores to enhance the likelihood of acceptance for this work. We thank you for your time and thoughtful consideration.**

---

> > ### Comment · Reviewer_QNeF · 2024-08-11
> >
> > I thank the authors for the detailed response. I have no further questions.

---

### Decision · Program_Chairs · 2024-09-25

**Decision:**

Accept (poster)

**Comment:**

This paper studies decentralized non-cooperative games where each player aims to minimize its expected risk over a distribution that changes with their own and other players’ decisions, subject to a coupled constraint. It establishes conditions for the existence and uniqueness of Nash equilibria (NE) and performative stable equilibria (PSE), and further bounds the distance between NE and PSE. The paper then proposes a primal-dual algorithm that solves the problem with provable convergence rates.


There are several highlighted weaknesses. For instance, one reviewer questioned the assumptions; for example, the work requires the stochastic gradient mapping to be strongly monotone, which is a stringent condition. Also, some reviewers questioned the difficulty of the analysis technique since they appear to follow relatively standard techniques in the literature extended to decentralized settings.
Despite these weaknesses, most reviewers find the paper well-written, with non-trivial and novel results. The results obtained in the paper take an important step toward characterizing the performative effect in multi-player non-cooperative settings. Given these strengths, and with most reviewers leaning toward acceptance, I recommend accepting this work.

In the final version, I highly encourage the authors to highlight the technical novelty of this work and to discuss the assumptions on $\varepsilon_i$ in equations (3) and (4), which seem a bit strong.